JCB Journal of Cell Biology

# Arf1-dependent LRBA recruitment to Rab4 endosomes is required for endolysosome homeostasis

Viktória Szentgyörgyi[1]*, Leon Maximilian Lueck[2]*, Daan Overwijn[1], Danilo Ritz[1], Nadja Zoeller[3], Alexander Schmidt[1], Maria Hondele[1], Anne Spang[1], and Shahrzad Bakhtiar[2]

**Deleterious mutations in the lipopolysaccharide responsive beige-like anchor protein (LRBA) gene cause severe childhood immune dysregulation. The complexity of the symptoms involving multiple organs and the broad range of unpredictable clinical manifestations of LRBA deficiency complicate the choice of therapeutic interventions. Although LRBA has been linked to Rab11-dependent trafficking of the immune checkpoint protein CTLA-4, its precise cellular role remains elusive. We show that LRBA, however, only slightly colocalizes with Rab11. Instead, LRBA is recruited by members of the small GTPase Arf protein family to the TGN and to Rab4+ endosomes, where it controls intracellular traffic. In patient-derived fibroblasts, loss of LRBA led to defects in the endosomal pathway promoting the accumulation of enlarged endolysosomes and lysosome secretion. Thus, LRBA appears to regulate flow through the endosomal system on Rab4+ endosomes. Our data strongly suggest functions of LRBA beyond CTLA-4 trafficking and provide a conceptual framework to develop new therapies for LRBA deficiency.**

## Introduction

LPS responsive beige-like anchor (LRBA) belongs to the beige and Chediak-Higashi (BEACH) domain–containing protein family (Wang et al., 2001). Biallelic mutations in LRBA are associated with severe immune deficiency and autoimmunity syndrome (Alangari et al., 2012; Lopez-Herrera et al., 2012). The average onset of the disease is at the age of two, and the only cure is allogeneic hematopoietic stem cell transplantation (alloHSCT) (Seidel et al., 2017; Bakhtiar et al., 2017; Tesch et al., 2020). The symptoms partially resemble those of CTLA-4 insufficiency (Lo et al., 2015). CTLA-4 is a plasma membrane receptor on regulatory T cells (Tregs) and suppresses autoimmune responses (Karandikar et al., 1996). In LRBA-deficient patients, the overall and surface CTLA-4 protein levels are decreased due to its increased lysosomal degradation, which is otherwise prevented by LRBA. Consequently, LRBA deficiency is accompanied by a reduced number and suppressive capacity of Tregs. Based on the role of LRBA in CTLA-4 trafficking and data on its *C. elegans* homolog SEL-2 (De Souza et al., 2007), LRBA has been linked to polarized endosomal recycling to the plasma membrane.

LRBA is a large, 319 kDa protein that contains an N-terminal concanavalin A–like domain, a domain of unknown function (DUF), a protein kinase A binding motif (AKAP), the conserved BEACH domain, a non-canonical PH domain, and C-terminal WD40 repeats, which are important for protein–protein interactions (Gebauer et al., 2004; Wang et al., 2001) (Fig. 1 A). LRBA is expressed in most human tissues and was reported to be associated with the Golgi apparatus and vesicular structures (Kurtenbach et al., 2017; Roussa et al., 2024). In support of this notion, the typical perinuclear localization of LRBA was lost when the Golgi was dispersed by brefeldin A (BFA) (Martinez-Jaramillo and Trujillo-Vargas, 2020; Kurtenbach et al., 2017). Since LRBA can coimmunoprecipitate with CTLA-4 and given that this interaction is abolished when the sorting motif in the CTLA-4 tail is mutated, LRBA has been proposed to be a cargo adaptor (Lo et al., 2015). Yet, how LRBA would exert its function in the endosomal pathway when the bulk of the protein is present on the Golgi remains elusive.

Here, we report that in patient-derived dermal fibroblasts, the loss of LRBA leads to defects in endosome maturation accompanied by the accumulation of enlarged endolysosomes. We show that LRBA is recruited by members of the small GTPase Arf protein family to the trans-Golgi network (TGN) and by Arf1 and Arf3 to Rab4+ endosomes, where it controls intracellular

---

[1]Biozentrum, University of Basel, Basel, Switzerland; [2]Department of Pediatrics, Goethe-University Frankfurt, Frankfurt, Germany; [3]Dermatology, Goethe University Frankfurt, Frankfurt, Germany.

*V. Szentgyörgyi and L.M. Lueck contributed equally to this paper. Correspondence to Shahrzad Bakhtiar: bakhtiar@med.uni-frankfurt.de; Anne Spang: anne.spang@unibas.ch.



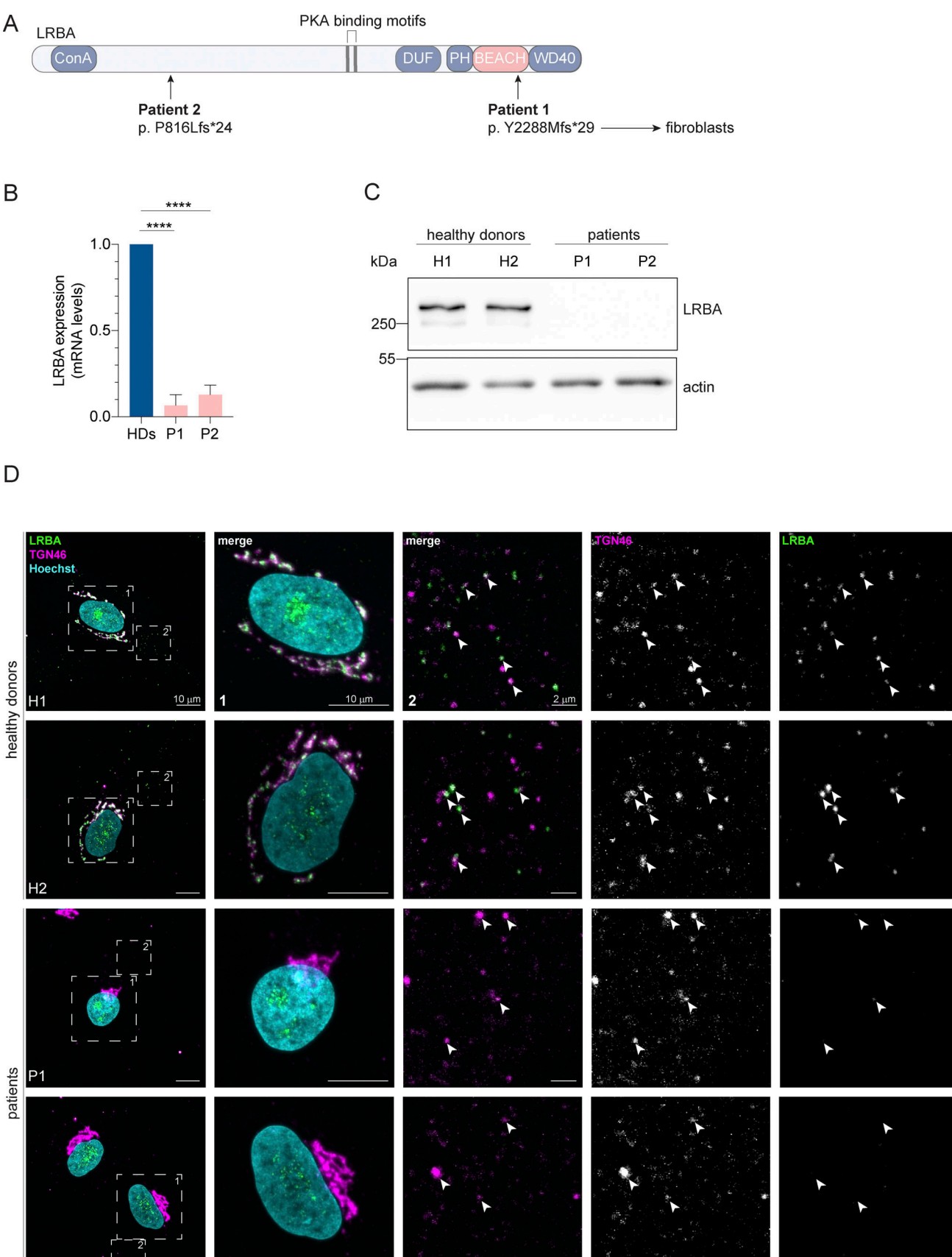

Figure 1.   **Distinct point mutations in the *LRBA* gene of two LRBA-deficient patients cause mRNA decay and loss of the protein. (A)** Schematic of LRBA protein structure with annotated domains. The genetic mutations carried by the two LRBA-deficient patients investigated in this study are shown. Dermal

fibroblasts obtained from these patients were used in this study. **(B)** LRBA mRNA levels in the two patient and two healthy donor (HD) fibroblast cell lines were determined by qRT-PCR. Mean and standard deviation are shown from $n$ = 4 biological replicates; one-way ANOVA using Dunnett's multiple comparison, ****$P < 0.0001$. **(C)** Immunoblot analysis of LRBA presence in fibroblasts of two HDs and two patient donors using polyclonal LRBA antibody and actin as a loading control. **(D)** Colocalization of LRBA and TGN46 in fibroblasts of HDs. LRBA is absent in patient-derived fibroblasts. Cells were fixed, immunostained with TGN46 and LRBA antibodies, and imaged using a confocal microscope. Squares show magnification of the perinuclear area (1) and the periphery (2). The labeling of the single channels represents the color of the channel on the merged image. H1: HD 1, H2: HD 2, P1: patient 1, P2: patient 2. Source data are available for this figure: SourceData F1.

trafficking and endolysosomal homeostasis. We propose that in LRBA deficiency the failure to correctly sort proteins for transport to the plasma membrane impedes endosome- and endolysosome maturation. These findings expand our understanding of the underlying pathomechanism in the LRBA deficiency syndrome.

## Results

### LRBA is absent in patients with LRBA deficiency and affects Golgi organization on a global scale

To gain a better understanding of the molecular function of LRBA in the endosomal system, we analyzed dermal fibroblasts of two LRBA-deficient patients. The two patients have mutations at amino acid position 816 and 2288, respectively (Fig. 1 A). We analyzed the expression of LRBA in these cells and detected strongly reduced LRBA expression on both the mRNA (Fig. 1 B) and the protein level (Fig. 1 C), confirming the LRBA deficiency. Within the LRBA locus, another gene, MAB21L2 is nested (Tsang et al., 2009). Our qRT-PCR analysis indicated that MAB21L2 transcripts are still present in both patients, although with somewhat reduced levels as compared with the controls (Fig. S1 A). Thus, LRBA deficiency in these patients does not chiefly affect MAB21L2 expression. Next, we analyzed LRBA localization in fibroblasts. In cells obtained from two healthy donors, LRBA localized mainly to the trans-Golgi network (TGN), as observed previously (Wang et al., 2001; Lo et al., 2015; Kurtenbach et al., 2017) (Fig. 1 D). We also detected LRBA in foci throughout the cytoplasm, which at times colocalized with TGN46. Consistent with the western blot results, LRBA was not detectable in patient-derived cells by immunofluorescence (Fig. 1 D). The TGN appeared more compacted in patient-derived cells as compared with the TGN in cells from healthy donors (Fig. 1 D and Fig. 2 A), indicating that LRBA may play a role in maintaining Golgi morphology.

To corroborate our findings, we determined the perinuclear distribution and the volume of the TGN (Fig. 2, B–D). While the volume of the TGN remained unchanged (Fig. 2 D), the TGN distribution around the nucleus was reduced in LRBA-deficient cells (Fig. 2, B and C), and in parallel, we observed an increase in the mean fluorescence intensity of TGN46 (Fig. 2 E). These observations are consistent with the role of LRBA in maintaining Golgi distribution in the perinuclear region. Despite the changes in Golgi distribution around the nucleus, the Golgi ribbon and cisternal organization per se appeared to be intact in LRBA-deficient cells as determined by electron microscopy (Fig. 2 F). Thus, loss of LRBA affects Golgi morphology, but the Golgi might still be functional.

### Traffic from the Golgi to the plasma membrane is only slightly impaired in the absence of LRBA

To assess whether the TGN compactness in LRBA-deficient cells impacted trafficking from and to the Golgi, we analyzed the localization of the mannose 6-phosphate receptor (M6PR) (Fig. 3, A–C and Fig. S1, B–D), which cycles between the TGN and endosomes in an adaptor protein complex-1 (AP1)- and retromer-dependent manner (Arighi et al., 2004). M6PR was not trapped in the TGN in patient-derived cells (Fig. 3, A and B), and the colocalization of the receptor with the TGN (Fig. 3, A and C) was not changed between patient-derived and healthy cells. The increase in the Mander's coefficient of TGN overlap with M6PR in patient-derived cells is most likely due to the more compact Golgi in those cells (Fig. 3 B). To confirm these findings, we analyzed the colocalization of M6PR with the retromer subunit Vps35 (Fig. S1, B–D). We only observed very minor differences, which we consider unlikely to reflect a biologically meaningful difference. Therefore, we conclude that the trafficking of M6PR is unaffected in LRBA-deficient fibroblasts. Furthermore, we tested the recruitment of AP1 to the TGN (Fig. S1 E). AP1-positive carriers, which recycle cargo back from the recycling endosomes to the Golgi network (Hirst et al., 2012; Robinson et al., 2024), were recruited normally in patient-derived cells.

To get a more global view, we determined the secretome and the surface proteome of patient-derived fibroblasts and compared them with those of healthy donors. To obtain the surface proteome, we biotinylated all plasma membrane proteins, followed by a pull-down with streptavidin beads and LC/MS analysis. We performed surface biotinylation for both patient-derived cell lines and pooled the data afterward to only detect proteins that changed plasma membrane localization in both patients. Our data revealed that a subset of proteins was reduced in the samples derived from patients' cells, among which were the transferrin receptor and EGFR (Fig. 3 D). Similar to the surface proteome, we observed the reduction of a subset of secreted proteins in the patient-derived fibroblasts (Fig. 3, E–G). Thus, our data are consistent with a slight defect in traffic from the Golgi to the plasma membrane and that only a few proteins are affected. Our data indicate that trafficking from and to the TGN is only mildly altered in LRBA deficiency. Moreover, Golgi assembly after golgicide A (GCA) washout was not affected by the lack of LRBA (Fig. S1 F), indicating that LRBA is not essential for Golgi morphology establishment or overall function. Thus, even though at steady state most of LRBA is at the Golgi, it does not seem to strongly regulate Golgi function.

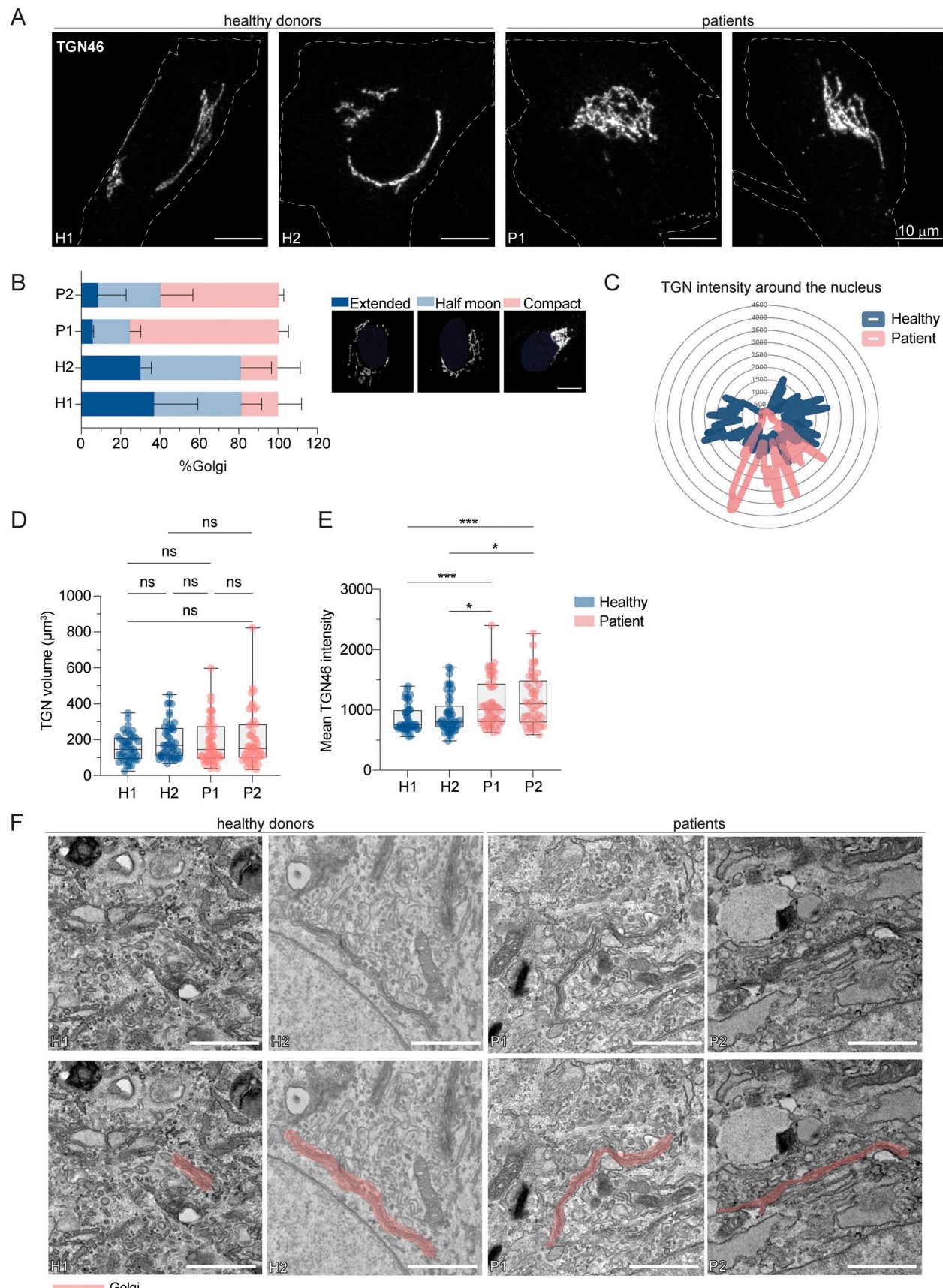

Figure 2. **LRBA deficiency promotes TGN compaction in patient-derived cells. (A)** TGN morphology analysis of two HDs and two patient-derived fibroblast lines visualized by the immunostaining of TGN46. Patient-derived cells show compacted TGN morphology. Representative confocal images from *n* = 3

biological replicates. Cell outlines are marked with a dashed line. H1: HD 1, H2: HD 2, P1: patient 1, P2: patient 2. **(B)** Measurements of TGN morphology (extended–dark blue, half moon–light blue, compact–pink) based on images taken in (A). Percentage of cells belonging to each category and standard deviation are shown; H1 = 47 cells, H2 = 47 cells, P1 = 53 cells, P2 = 49 cells from n = 3 biological replicates. The legend shows a representative image for each morphology category. **(C)** Representative TGN46 signal distribution around the nucleus in H1 and P1 cells. **(D and E)** Quantification of (D) TGN volume and (E) mean TGN46 intensity based on images taken in A. Mean and minimum to maximum are shown, the box ranges from the first (Q1–25th percentiles) to the third quartile (Q3–75th percentiles) of the distribution. All data points are shown.; H1 = 46 cells, H2 = 49 cells, P1 = 51 cells, P2 = 47 cells from n = 3 biological replicates; Kruskal–Wallis test using Dunn's multiple comparison, ***P = 0.0007 (H1 versus P1), ***P = 0.0005 (H1 versus P2), *P = 0.0248 (H2 versus P1), *P = 0.0170 (H2 versus P2). **(F)** TEM images of two HDs and two patient-derived fibroblasts show intact Golgi cisternae. In the lower row, pink masks highlight Golgi stacks. Scale bar, 1 μm.

## LRBA loss leads to the accumulation of enlarged endolysosomes in patient-derived fibroblasts

Surprisingly, we also observed a substantial increase in the secretion of endosomal/lysosomal proteins from the patient-derived samples (Fig. 3, E–G; and Fig. S2, A and B). In addition, the most striking and obvious phenotype of the patient-derived fibroblasts was a strong accumulation of enlarged endosomal/endolysosomal structures as observed by electron microscopy (Fig. 4 A). Endolysosomes are the fusion product of late endosomes (also dubbed multivesicular bodies, carrying cargo proteins for degradation) and lysosomes, and represent the active degradation compartment (Podinovskaia and Spang, 2018) (Fig. S2 C). In cells from healthy donors, endolysosomes appeared as electron-dense structures and contained many membrane layers as reported previously (Klumperman and Raposo, 2014) (Fig. 4 A, inlays, black arrowheads). In patient-derived cells, late endosome–lysosome fusion could probably mostly still occur, but the electron-dense lysosome-derived material was segregated to one side in the enlarged endolysosome (Fig. 4 A, inlays, white arrowheads), indicating that either lysosomal function or endolysosomal maturation/lysosome reformation could be impaired. First, we confirmed that these accumulating structures are indeed endolysosomes. We stained the cells for the endolysosomal/lysosomal marker LAMP1 (CD107a) (Chen et al., 1985). Indeed, we observed a strong accumulation of LAMP1-positive structures in patient-derived cells that were overall larger and more spread throughout the cell compared to cells from healthy donors (Fig. 4 B). Similarly, the vacuolar ATPase (V-ATPase) accumulated on the enlarged endolysosomal structures (Fig. S2 D), and LAMP1 protein expression levels were increased in patient cells (Fig. S2, E and F). The altered gel mobility of LAMP1 suggested the presence of under-glycosylated protein species. Next, we asked whether the enlarged endolysosomes in patient-derived cells retained their degradative capacity. Thus, we determined whether these endolysosomes were acidified and contained active proteases. To this end, we stained lysosomal compartments with Lyso-Tracker, which showed a similar pattern to LAMP1 in patient-derived cells, indicating proper acidification (Fig. S2, G and H). Similar results were obtained by cathepsin D immunostaining (Fig. S2 I). Cathepsins reach endolysosomes in an inactive pro-form, which are called pro-cathepsins. Upon arrival in the endolysosome, the pro-form is hydrolytically converted into the active form. This process requires a low pH and is dependent on active proteases (Zaidi et al., 2008). Magic Red is a substrate that becomes fluorescent upon cleavage by cathepsin B and thereby allows the detection of cathepsin B activity. The

Magic Red staining of catalytically active proteolytic structures was more spread and increased in intensity and number in the patient-derived fibroblasts (Fig. 4, C and D). In addition, we measured the levels of active, matured cathepsin D by western blot (Fig. 4, E and F). The levels of active cathepsin D were similar between healthy and patient-derived cells (Fig. 4, E and F). Furthermore, the late endosome/endolysosomal marker Rab7 showed a similar pattern as LAMP1 (Fig. S4 J). Taken together, our data suggest that enlarged proteolytically active endolysosomes accumulate in LRBA-deficient fibroblasts.

## Receptor degradation is not impaired in LRBA-deficient fibroblasts

If our assumption was correct, protein degradation in the enlarged endolysosomes in LRBA-deficient cells should still be functional. Thus, we followed the uptake and degradation of fluorescently labeled EGF, which upon binding to EGF receptors (EGFRs) on the cell surface is endocytosed and subsequently degraded in endolysosomes (Tomas et al., 2014). In cells from healthy donors, the EGF-TexasRed signal accumulated intracellularly within 15 min and had largely decayed after 60 min (Fig. 5, A and B). Even though the initial uptake of EGF within 15 min was lower in patient-derived cells, EGF degradation occurred with similar kinetics to those of the healthy control confirming that the enlarged endolysosomes are still active. The lower intracellular EGF levels at 15 min of uptake could be explained by the reduced levels of EGF receptor observed in LRBA-deficient cells (Fig. S3, A and B). Yet, these data still did not explain why endolysosomes were bigger in patient-derived cells. One possibility could be that the entire endosomal pathway would be affected, and already early endosomes would be enlarged. However, this does not appear to be the case because early endosomes were not consistently enlarged in the patient-derived cell lines. (Fig. 5, C and D), suggesting that the role of LRBA is downstream of early endosomes. To study later events of endosome maturation we tested EGFR signaling in LRBA-deficient cells (Fig. S3, C–E). EGF stimulation of EGFR on the cell surface induces autophosphorylation and tyrosine-kinase activity which stimulates signaling cascades in the cell. Activated EGFRs undergo rapid endocytosis and remain active on the surface of early endosomes (Tomas et al., 2014). To attenuate signaling, EGFRs need to be sorted into intraluminal vesicles (ILV) inside the endosome. Later, these multivesicular bodies/late endosomes fuse with lysosomes, and their content is degraded. If ILV formation is perturbed, prolonged EGFR activation and downstream signaling can be observed (Eden et al., 2012). Thus, we stimulated LRBA-deficient and healthy cells

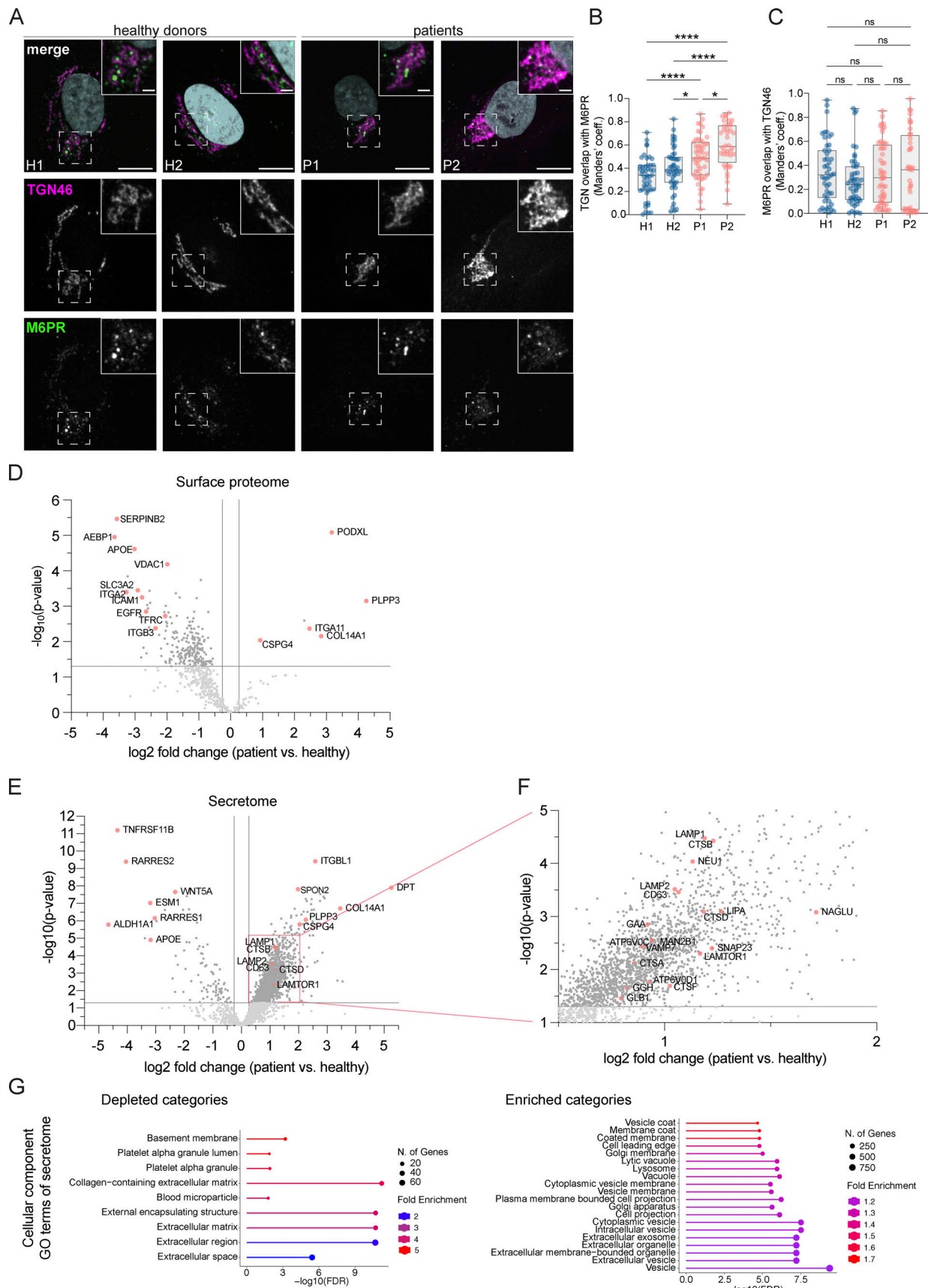

Figure 3. **Golgi-endosome traffic and secretion are only slightly altered in LRBA deficiency. (A)** Colocalization analysis of TGN46 and M6PR in two HDs and two patient-derived fibroblast lines. Representative confocal immunofluorescence images of single focal planes. Squares show the magnified area. Inlays

are shown in the top right corner of the images. The labeling of the single channels represents the color of the channel on the merged image. Scale bar, 10 µm, inlays 2 µm. **(B and C)** Colocalization between TGN46 and M6PR was measured using Mander's colocalization index. Mean and minimum to maximum are shown, box ranges from the first (Q1–25th percentiles) to the third quartile (Q3–75th percentiles) of the distribution. H1 = 47 cells, H2 = 47 cells, P1 = 52 cells, P2 = 40 cells from $n$ = 3 biological replicates. **(B)** Mander's coefficient of TGN46 overlap with M6PR. One-way ANOVA using Tukey's multiple comparison, ****$P$ < 0.0001, *$P$ = 0.0432 (H2 versus P1), *$P$ = 0.0324 (P1 versus P2). **(C)** Mander's coefficient of M6PR overlap with TGN46, Kruskal–Wallis test using Dunn's multiple comparison. **(D)** Scatter plot of LRBA deficient versus healthy donor fibroblasts' protein abundances in the cell surface proteome determined with cell surface biotinylation and LC-MS analysis. Datapoints above $P$ value scores of 0.05 are indicated in light grey. Highlighted, significantly altered proteins ($P$ < 0.05) are indicated in pink colors. **(E)** Volcano plot of patient/healthy donor protein abudances in the secretome determined by LC-MS analysis. **(F)** Inlay of volcano plot shown in panel E. **(G)** Gene ontology analysis of enriched and depleted cellular component categories in the patients' secretome.

with EGF and followed activated and total EGFR levels (Fig. S3, C–E). Again, we observed lower levels of EGFR in patient-derived cells, yet the signaling pathway was activated, which we measured by blotting for pEGFR and pERK (Fig. S3, C–E). The signaling attenuation and EGFR degradation kinetics were similar between healthy and patient-derived cells. Thus, our data indicate that EGFR sorting into ILVs is not severely affected and that there is most likely a defect either during endosomal recycling or endolysosome-to-lysosome maturation.

## LRBA does not regulate the degranulation of CD8[+] T or NK cells

Our data show that enlarged endolysosomes accumulate in LRBA-deficient fibroblasts but that they still can degrade their content. Intriguingly, similar endolysosomal structures have been observed in Chediak–Higashi syndrome (CHS) patient-derived cells from several cell types (Burkhardt et al., 1993; Stinchcombe et al., 2000). The LYST protein, responsible for CHS disease, is a member of the BEACH domain protein family to which LRBA also belongs. This disease emerges from impaired cytotoxicity of natural killer (NK) and cytotoxic T cells (Abo et al., 1982; Baetz et al., 1995). These cells use lysosome-related lytic granules that contain, in addition to typical lysosomal proteins, perforin and granzymes (Fig. S4 A). The exocytosis of these granules at the cell-cell contact site—the immunological synapse—results in the induction of apoptosis in the target cell (Krzewski and Coligan, 2012). In CHS, although granule biogenesis is unimpaired, lytic granules are enlarged and fail to fuse with the plasma membrane (Stinchcombe et al., 2000; Gil-Krzewska et al., 2018). To test whether LRBA deficiency also leads to impaired exocytosis of lytic granules (degranulation) in CD8[+] cytotoxic T cells and NK cells, we performed degranulation assays on patient-derived cells (Fig. S4). During the fusion of lytic granules with the plasma membrane, LAMP1 (CD107a) becomes exposed on the cell surface, which can be used as a marker for flow cytometric analysis of degranulating cells (Alter et al., 2004). Degranulation and surface exposure of LAMP1 was promoted in T cells with PMA and ionomycin (Fig S4, B–F), while NK-cells were stimulated with interleukin-2 (IL-2) and K562 cells (Fig. S4, G and H). The difference in degranulation was given by the ratio of mean LAMP1 fluorescence intensities (MFI) between the patient and healthy donor (Fig. S4, F and H). As expected, under basal conditions, T cells did not show degranulation in either healthy donors or in the two LRBA-deficient patients (Fig. S4 D). Upon stimulation, degranulation of patient-derived CD8[+] T cells (Fig. S4, D and F) and NK cells (Fig.S4, G and H) was almost as efficient as that of cells from healthy donors. These results indicate that in spite of the similarity in the accumulation of enlarged endolysosomes in CHS and LRBA deficiency, LYST and LRBA have distinct cellular functions.

## LRBA is recruited by Arf1 and Arf3 onto Rab4[+] endosomes

To gain more mechanistic insights into the role of LRBA in endosomal trafficking, we switched to HeLa cells, which also endogenously express LRBA (Fig. S5, A and B). Similar to what we observed in fibroblasts, most of the LRBA accumulated at the TGN and colocalized with TGN46 and the small GTPase Arf1 and to a lesser extent with cis-Golgi marker giantin (Fig. 6, A and C; and Fig. S5, C and E). More importantly, we also detected LRBA on vesicular structures in the cell periphery (Fig. S5 B, inlay), like in fibroblasts from healthy donors (Fig. 1 D). We next aimed to uncover the identity of these structures by determining LRBA colocalization with different endosomal markers (Fig. 6 B). To our surprise, LRBA did not colocalize with Rab7[+] late endosomes/endolysosomes or LAMP1[+] lysosomes/endolysosomes (Fig. 6, B and C). As expected, LRBA did not colocalize with M6PR either (Fig. S5, D and E). These data indicate that the observed endolyosomal enlargement in LRBA-deficient fibroblasts might be a consequence of disturbances upstream in the pathway. LRBA appeared juxtaposed to Rab5[+] compartments; clearly on separate domains or structures (Fig. 6 B). There was a modest overlap between LRBA and Rab11 (Fig. 6, B and C). However, we observed the strongest colocalization of LRBA with Rab4[+] endosomes (Fig. 6, B and C), TGN46, and Arf1 in the cell periphery (Fig. 7, A and B). Consistent with this finding, LRBA has been reported to be sensitive to the ArfGEF inhibitor BFA (Kurtenbach et al., 2017; Martinez-Jaramillo and Trujillo-Vargas, 2020). Indeed, LRBA localization was sensitive to both BFA and GCA treatment (Fig. S5 F), indicating that Arf proteins are required for LRBA recruitment to Rab4[+] endosomes. It has been shown previously that Arf1 and Arf3 are present in Rab4[+] endosomes (D'Souza et al., 2014; Wong-Dilworth et al., 2023). To test whether Arf1 and/or Arf3 are required for LRBA recruitment, we used ARF1 knockout (KO), ARF3 KO, and ARF1+3 double KO (dKO) HeLa cells (Fig. 7 C) (Pennauer et al., 2022) and stained them for endogenous LRBA (Fig. 7 D). While LRBA showed a similar distribution in the single ARF1 KO and ARF3 KO cells as in the parental HeLa cell line, it was lost from endosomal structures in ARF1+3 dKO cells (Fig. 7, D and E). This effect was specific because the expression of Arf1-EGFP rescued the endosomal localization of LRBA (Fig. 7, F and G). We also tested whether LRBA could play a role in the recruitment of Arf1 onto Rab4[+] endosomes. However, the level of colocalization

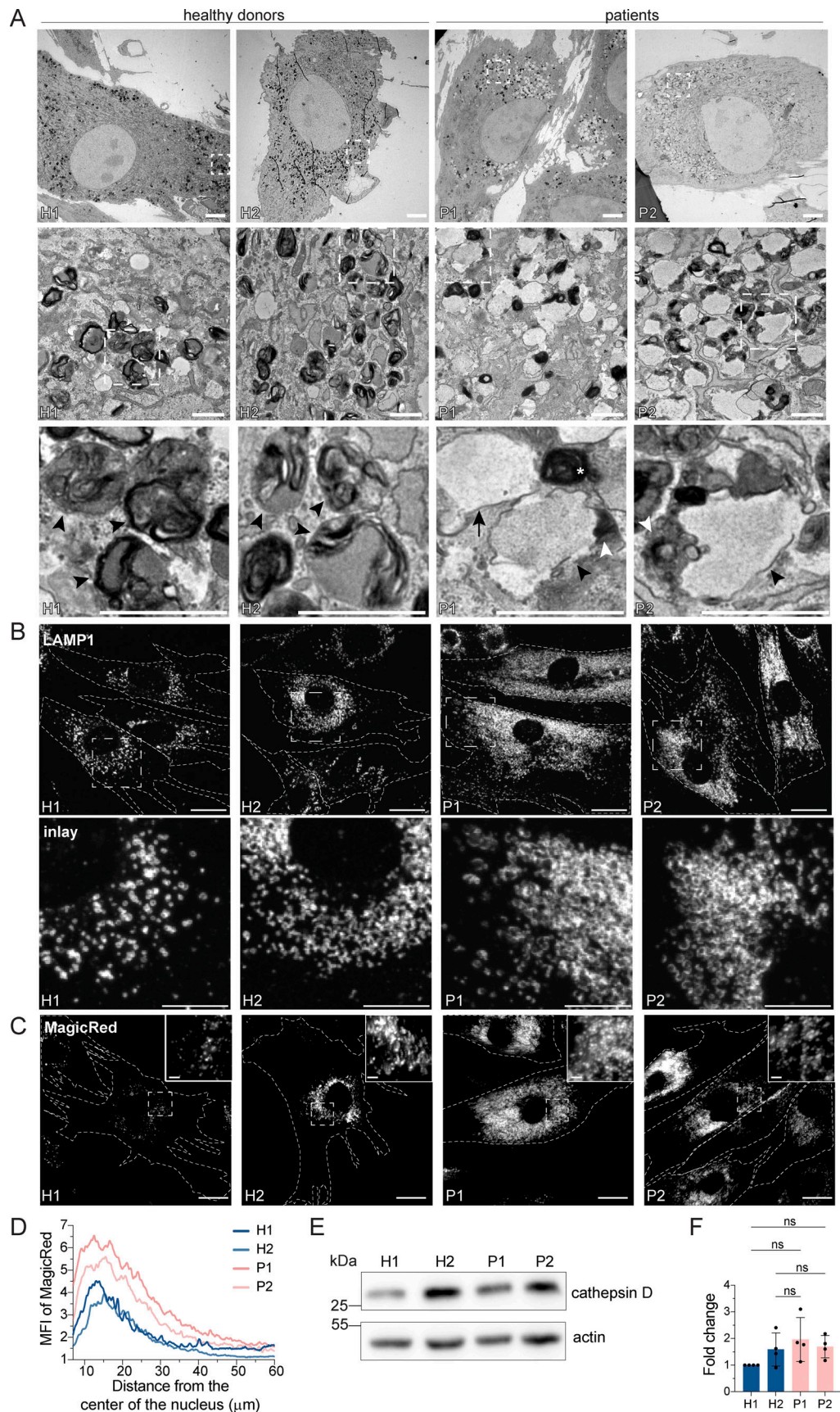

Figure 4. **LRBA-deficient fibroblasts accumulate enlarged endolysosomes. (A)** TEM analysis of accumulating endolysosomes in patient-derived cells. Squares show magnification of the endolysosomal structures. HDs showed electron-dense endolysosomes (black arrowheads). In contrast, in patient-derived

cells, endolysosomes (black arrowheads) showed restricted degradative (electron-dense) domains (white arrowheads). Lysosomes (white star) and endosomes (black arrow) are shown. Two HDs and two patient-derived fibroblast lines were embedded and analyzed. Scale bar, 5 µm, inlays in the third row 1 µm. **(B)** Immunofluorescence analysis of accumulating, enlarged (endo)lysosomes in patient-derived cells. Fibroblasts were fixed with methanol and stained for LAMP1. Cell outlines are marked with a dashed line. Squares show magnification of the (endo)lysosomes. Representative confocal images are shown from $n$ = 3 biological replicates. Scale bar, 20 µm, inlays 10 µm. **(C)** Analysis of cathepsin B activity in LRBA-deficient fibroblasts. LRBA-deficient patient-derived and healthy fibroblasts were plated onto imaging chambers and their lysosomes were visualized with Magic Red (indicating cathepsin B activity) and imaged live at 37°C, 5% $CO_2$. Representative wide-field images are shown from $n$ = 3 biological replicates. Cell outlines are marked with a dashed line. Squares show the magnified area. Inlays are shown in the top right corner of the images. Scale bar, 20 µm, inlays 2 µm. **(D)** Quantification of the MagicRed intensity along the nucleus-cell periphery axis. A line ROI was drawn from the nucleus to the cell periphery and Magic Red intensity was measured. Values were normalized to the maximum of each cell and averaged per experiment. The mean of three biological replicates is plotted along the axis; H1 = 45 cells, H2 = 53 cells, P1 = 51 cells, P2 = 49 cells were analyzed. **(E)** Western blot analysis of matured cathepsin D (heavy chain) protein levels in healthy and patient-derived fibroblasts using actin as a loading control. **(F)** Quantification of cathepsin D levels based on immunoblots shown on panel E from $n$ = 4 biological replicates mean ± SD; one-way ANOVA using Tukey's multiple comparison. Source data are available for this figure: SourceData F4.

between Arf1 and Rab4 was independent of the presence of LRBA (Fig. S5, G–I). Intriguingly, in the dKO cells, LRBA localization to the Golgi was not affected. We assume that Arf4 and Arf5, which reside on the Golgi, could most likely compensate for the loss of Arf1 and 3 in the recruitment of LRBA to the Golgi.

Our data indicate that Arf1 and Arf3 are required for the recruitment of LRBA to Rab4+ endosomes and suggest that loss of the endosomal LRBA pool is responsible for the accumulation of enlarged endolysosomes. Thus, we hypothesized that ARF1+3 dKO cells would also display enlarged endolysosomes. Indeed, LAMP1+ endolysosomes were enlarged in the absence of Arf1 and Arf3 (Fig. 7, H and I). To determine whether LRBA and Arf1/3 could interact directly, we turned to an in silico approach. We first predicted the structure of LRBA using Alphafold monomer (Evans et al., 2021, *Preprint*; Jumper et al., 2021) (Fig. 8, A and B). The predicted LRBA structure showed four α-solenoid regions (green), which are classical protein–protein interaction regions, connected by flexible linkers creating a rod-like structure. These α-solenoid regions contain the earlier described DUF domain (dark green). These regions are flanked on one side by several domains, which include concanavalin A like (cyan), PH (magenta), BEACH (red), and WD40 domain (yellow). Next, we used Alphafold multimer (Evans et al., 2021, *Preprint*; Jumper et al., 2021) to predict potential interaction sites of LRBA with either Arf1 (Fig. 8 C) or Arf3. In most of our models, we observed that both Arf1 and Arf3 are predicted to potentially interact using the conserved amino acids Ile[46] and Ile[49] (Fig. 8, C–E and Fig. S5 J) with two different hydrophobic pockets within the LRBA solenoid regions around either the Leu[861] or Ile[918], both of which are evolutionary conserved (Fig. 8 C and Fig. S5 K). Additionally, several models showed Arf1 Asp[52] interacting with LRBA Arg[910]. Thus, our in silico data revealed potential conserved binding sites between Arfs and LRBA, suggesting a direct interaction of Arf1 with LRBA.

If the prediction of the LRBA binding site in Arf1 was correct, mutation of this site should abolish the Arf1-dependent recruitment of LRBA onto Rab4+ endosomes. We mutated Ile[46] and Ile[49] to either Ala (Arf1[I46A, I49A]-EGFP) or Ser (Arf1[I46S, I49S]-EGFP) and to a triple mutant in which also the neighboring Asn[52] was mutated to alanine (Arf1[I46A, I49A, N52A]-EGFP) (Fig. 8, C and E). Both Arf1 double mutants still localized to Rab4+ endosomes (Fig. 8, F–H), while the triple mutation abolished the colocalization. Next, we tested whether the double mutants were still capable of recruiting LRBA to endosomes. While the Ile->Ala

mutant still rescued LRBA recruitment in ARF1+3 dKO cells, expression of the Ile->Ser mutant did not promote recruitment of LRBA to endosomes (Fig. 8, I and J). Our experimental data support the prediction of the Alphafold model for the LRBA binding site in Arf1. To test our model biochemically, we aimed to copurify Arf1 and LRBA. Therefore, we expressed Arf1-EGFP, Arf1[Q71L]-EGFP (constitutively active), Arf1[I46S, I49S]-EGFP, and Arf1[T31N]-EGFP (dominant-negative) in HEK293A cells and aimed to pull down endogenous LRBA (Fig. 9, A and B). Arf1-EGFP coimmunoprecipitated endogenous LRBA. This interaction was stronger when the constitutive active Arf1[Q71L]-EGFP was used (Fig. 9, A and B). On the other hand, the Arf1[I46S, I49S]-EGFP binding mutant and the dominant-negative Arf1[T31N]-EGFP precipitated less LRBA compared with the constitutive active Arf1 and both Arf1 mutants tend to bind less LRBA compared with wild-type Arf1. Our data suggest that active Arf1 directly interacts with LRBA and recruits it to Rab4+ endosomes.

## Discussion

In this study, we show that LRBA is recruited to Rab4+ endosomes by Arf1 and Arf3 through a direct interaction with these small GTPases. This LRBA localization seems to regulate the function of the endolysosomal pathway; when endosomal LRBA recruitment is perturbed, endolysosomes become enlarged. The endolysosomes are probably still functional when LRBA is lost because they were acidified, and transport from the TGN to the lysosomes and the maturation of lysosomal enzymes appeared to be functional. Thus, our data reveal a novel role of LRBA in the endosomal pathway (Fig. 9 C). Whether the enlarged endolysosomes, which appear to be defective in the lysosomal maturation process, are a direct consequence of LRBA deficiency or whether they are a consequence of the defects in endosomal recycling needs to be established. We currently favor a model in which the defects in sorting and transport to the plasma membrane at Rab4+ endosomes would prevent the endosome/endolyosomes from maturing (Fig. 9 C). At least in T cells, the recycling of CTLA-4 to the plasma membrane is LRBA-dependent (Lo et al., 2015). In addition, we observed reduced surface expression of EGFR in the absence of LRBA, consistent with the role of LRBA in recycling to the plasma membrane (Fig. 3 D and Fig. S3, A–C). Moreover, interfering with Rab11-dependent recycling to the plasma membrane yields functional, enlarged endolysosomes (Zulkefli et al., 2019).

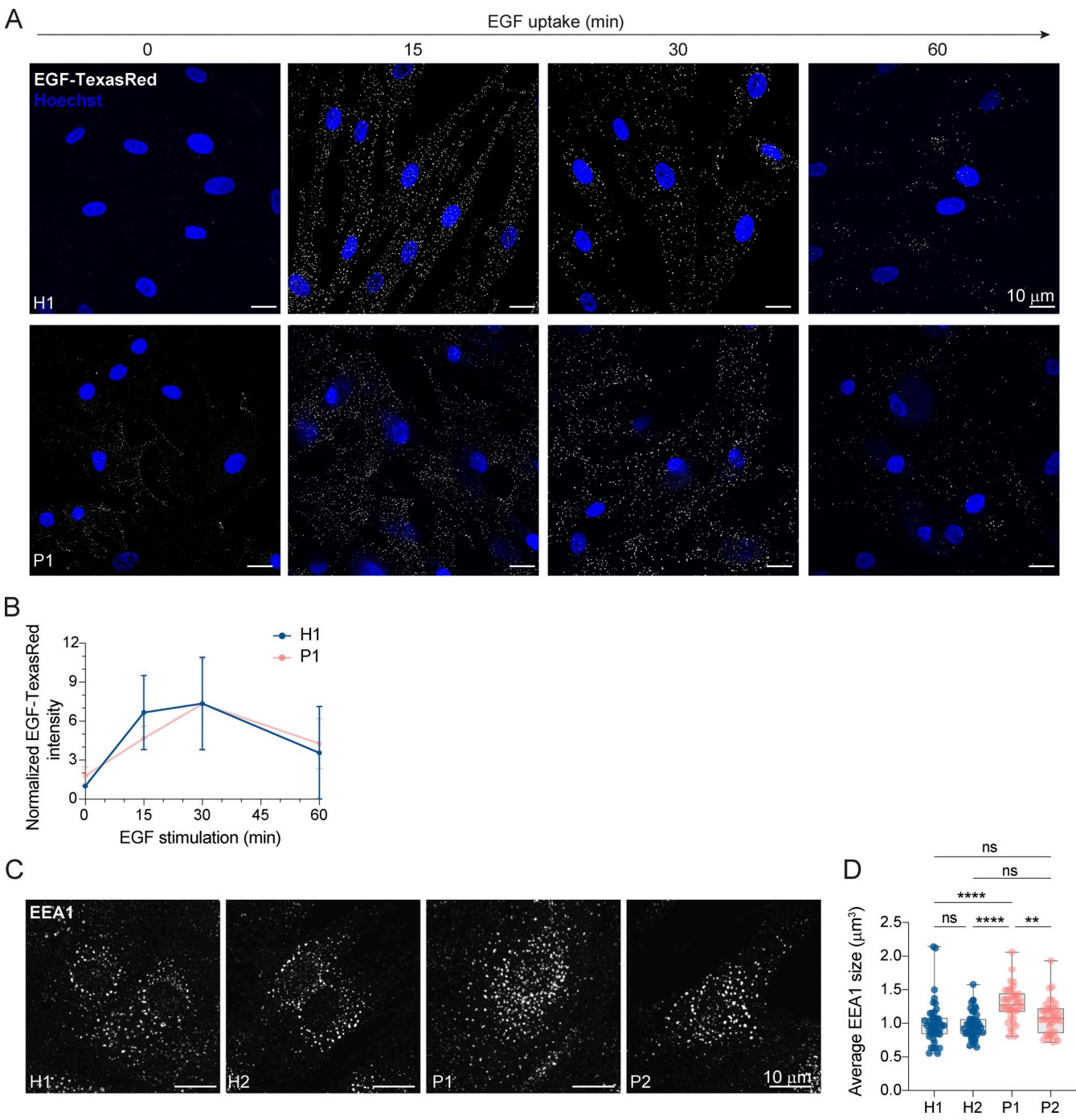

Figure 5. **Lysosomal degradation is unimpaired in LRBA deficiency. (A)** EGF-TexasRed uptake and degradation assay show unimpaired degradation in LRBA deficiency. H1 and P1 fibroblasts were serum-starved for 3 h and then were incubated on ice with EGF-TexasRed for 30 min. Cells were then washed 3× with ice-cold PBS and incubated with unlabeled EGF for indicated time points at 37°C. Cells were rinsed with ice-cold PBS, fixed with 4% PFA, and mounted in the presence of Hoechst dye to visualize nuclei. Overview confocal images of EGF-TexasRed signal are shown for each time point. Background signal has been subtracted using Gaussian blurring and image subtraction in Fiji. Maximum intensity projection of confocal Z-stacks. **(B)** Measurement of EGF-TexasRed fluorescence intensity based on images shown in A. Integrated density of EGF-TexasRed per cell was measured, averaged and normalized to H1 levels at 0 time point for each experiment; H1(0 min) = 38 cells, H1(15 min) = 58 cells, H1(30 min) = 67 cells, H1(60 min) = 44 cells, P1(0 min) = 36 cells, P1(15 min) = 51 cells, P1(30 min) = 45 cells, P1(60 min) = 50 cells were analyzed from $n$ = 3 biological replicates. Mean and SD is shown. **(C)** Immunofluorescence analysis of EEA1-positive early endosomes in healthy and LRBA-deficient patient-derived fibroblasts. Maximum intensity projection of confocal Z-stacks. **(D)** Measurement of the average size of early endosomes per cell was measured. Mean and minimum to maximum are shown, box ranges from the first (Q1–25th percentiles) to the third quartile (Q3–75th percentiles) of the distribution. All data points are shown. H1 = 43 cells, H2 = 44 cells, P1 = 40 cells, P2 = 40 cells were analyzed from $n$ = 3 independent experiments; Kruskal–Wallis test using Dunn's multiple comparison. ****$P < 0.0001$, **$P = 0.0093$.

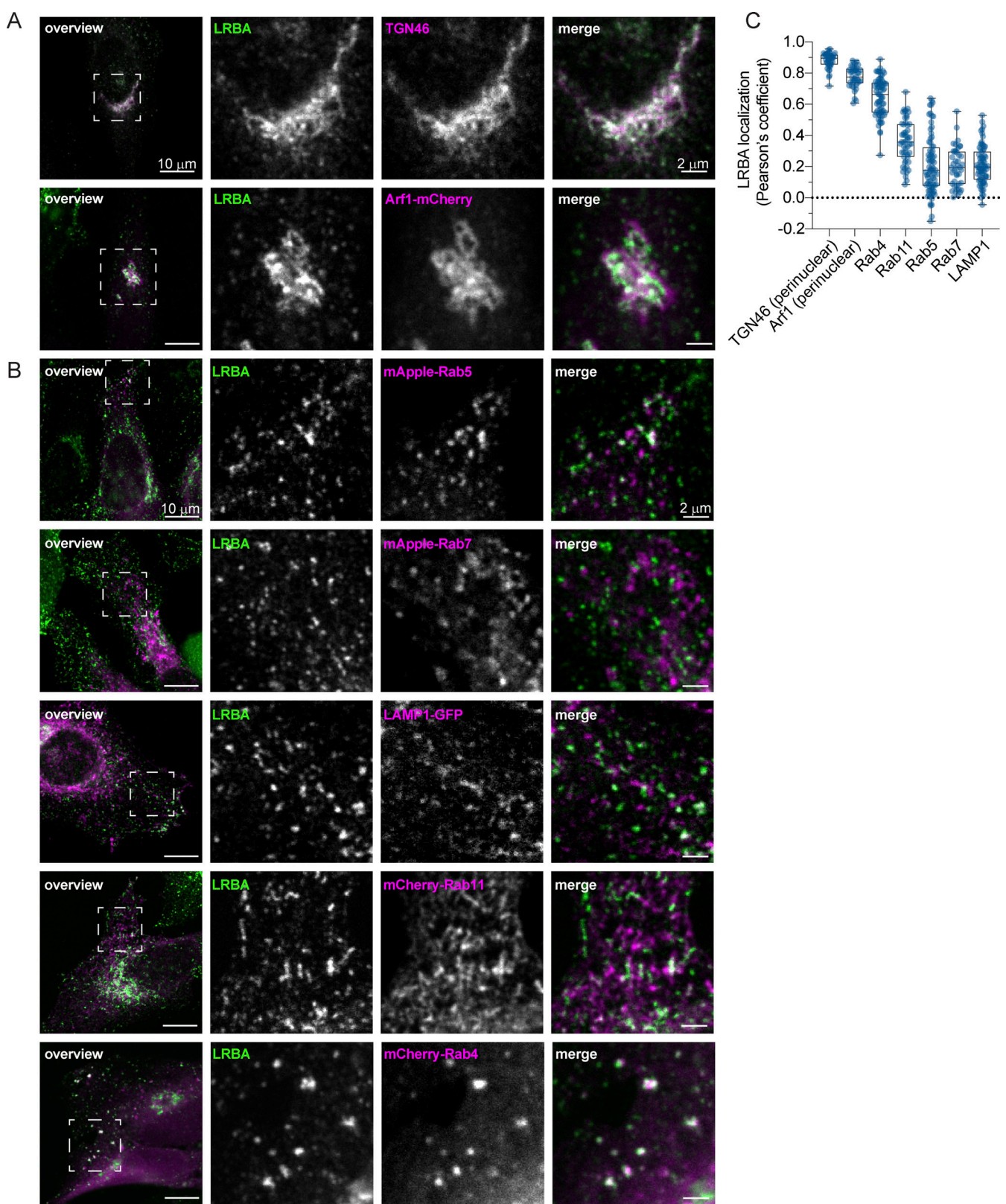

**Figure 6. LRBA colocalizes with the TGN and with Rab4⁺ endosomes in HeLa cells. (A)** Colocalization analysis of LRBA with TGN46 and Arf1 in HeLa cells. For the colocalization analysis with the TGN, HeLa cells were fixed with 4% PFA and stained for TGN46 and endogenous LRBA. For colocalization analysis with Arf1, HeLa cells were transfected with ARF1-mCherry, fixed with 4% PFA, and stained for endogenous LRBA. Squares show magnification of the perinuclear area. The labeling of the single channels represents the color of the channel on the merged image. **(B)** Colocalization analysis of LRBA and different endosomal markers. LRBA colocalizes with Rab4 and is found in juxtaposition to Rab11 recycling endosomes and to Rab5 early endosomes. HeLa cells were transfected with mApple-Rab5, mApple-Rab7, LAMP1-GFP, mCherry-Rab11, mCherry-Rab4, respectively, and stained for endogenous LRBA. Representative confocal

images of single focal planes are shown. Squares show the magnified areas. The labeling of the single channels represents the color of the channel on the merged image. **(C)** Colocalization measurements of LRBA and intracellular organelles. To measure LRBA colocalization with TGN46 and with Arf1 at the Golgi, one ROI at the perinuclear region was analyzed. For the endosomal markers, two ROIs per cell at the cell periphery were analyzed and the Pearson's coefficient was measured using the JACoP plugin in Fiji. Mean and minimum to maximum are shown, box ranges from the first (Q1–25th percentiles) to the third quartile (Q3–75th percentiles) of the distribution. All data points are shown. TGN46(perinuclear) = 40 cells, Arf1 (perinuclear) = 39 cells, Rab4 = 34 cells, Rab11 = 26 cells, LAMP1 = 35 cells, Rab5 = 35 cells, Rab7 = 21 cells from $n$ = 3 biological replicates.

In addition, LRBA was reported to be on Rab11[+] recycling endosomes in T cells and renal collecting duct tissue (Lo et al., 2015; Yanagawa et al., 2023). When we revisited this colocalization, we observed some overlap; LRBA was mostly juxtaposed to Rab11[+] domains in HeLa cells (Fig. 6, B and C). LRBA was mainly present on Rab4[+] endosomes in line with previous report showing that LRBA is enriched on Rab4a[+] compartments and was less present on Rab11a[+] and Rab25[+] endosomes (Wilson et al., 2023). Strikingly, we showed that the specific loss of the endosomal pool of LRBA is responsible for the major phenotype observed in patient cells, the accumulation of enlarged endolysosomes (Fig. 7, H and I). In line with this, Rab4[+] endosomes play a role in lysosome biogenesis, in particular in the absence of Rab7 (Wang et al., 2024, Preprint), and Arf1[+] vesicles were found to be critical for lysosomal tubule fission (Boutry et al., 2023). We speculate that LRBA on the Rab4[+] endosomes may play a role in endocytic lysosome reformation.

Still, the bulk of LRBA was present on the Golgi. Moreover, we and others showed that LRBA becomes dispersed when the Golgi is abrogated by BFA and GCA (Kurtenbach et al., 2017; Martinez-Jaramillo and Trujillo-Vargas, 2020). Consistent with the Golgi localization of LRBA, the Golgi morphology was altered in LRBA-deficient cells and became more compact. Nevertheless, we could only detect mild effects on Golgi function. LAMP1 appeared to be underglycosylated and a subset of proteins was less efficiently secreted or present at the plasma membrane when LRBA was lost (Fig. 3, D and E). In contrast, patient-derived cells secreted more endosomal/lysosomal proteins, which might be indicative of endolysosomes/lysosomes fusing with the plasma membrane and/or exosome secretion. Interestingly, similar phenotypes in terms of endosomal/lysosomal protein secretion have been observed when retromer function was diminished (Daly et al., 2023). We speculate that when the final stages of endosome maturation are perturbed, those late endosomes might be prone to secretion. Alternatively, but not mutually exclusively, the accumulation of endolysosomes may drive their secretion.

We further established that LRBA is recruited to endosomes specifically by Arf1 and Arf3. We propose that this recruitment is mediated via a direct interaction between Arf1/3 and LRBA through the Ile[46] and Ile[49] sites in the hydrophobic pocket of Arfs. Supporting our model, this hydrophobic pocket (Ile[49],Gly[50],Val[68],Gly[69],Ile[74],Leu[77]) together with three other amino acids referred to as hydrophobic triad (Phe[51], Trp[66], Tyr[81]) build a hydrophobic surface which is often recognized by Arf effectors like ARHGAP21, COPI, and GGA (Jacques et al., 2002; Sun et al., 2007; Ménétrey et al., 2007). This hydrophobic pocket is only exposed in the GTP-bound form, and despite high variability in the structures of effectors, a hydrophobic

residue, usually Ile or Leu from the effector, interacts with the pocket (Chavrier and Ménétrey, 2010). Similarly, our Alphafold prediction indicates the LRBA residues Leu[861] and Ile[918] for interaction with Arf1 and Arf3, respectively. While we could confirm our prediction for the binding site of LRBA in Arf1 by mutating Ile[46] and Ile[49] to serine, because of technical difficulties, we were unable to test mutations in LRBA. Thus, at this point, we could only predict the Arf1 binding site in LRBA.

Besides the most prominent and classical localization on the Golgi, Arf1 and Arf3 were found on different endosomal membranes, such as Rab4[+] endosomes (D'Souza et al., 2014) and TGN46-positive Golgi-derived vesicles (Boutry et al., 2023; Wong-Dilworth et al., 2023). Interestingly, we found LRBA on both Rab4[+] and on TGN46 and Arf1 positive structures where LRBA regulates endosomal recycling. LRBA also likely binds to Arf4 and/or Arf5 because in the ARF1+3 dKO, LRBA Golgi localization was unaffected. Moreover, the binding pocket is conserved among all these Arfs (Fig. 8 E). The most prominent phenotype, we observed in LRBA-deficient cells was the accumulation of enlarged endolysosomes, which is caused by the loss of the endosomal LRBA pool. Interestingly, similar findings have been reported in Chediak-Higashi syndrome (CHS) patients (Burkhardt et al., 1993; Stinchcombe et al., 2000; Bowman et al., 2019). In the CHS disease, giant lysosomes and lysosome-related organelles like melanosomes are present in patient-derived cells due to the aberrant function of lysosome-related organelles. Similar to CHS, we found that LRBA does not affect lysosomal biogenesis and degradation and does not colocalize with LAMP1 under steady-state conditions. Lysosomes were correctly acidified and contained catalytically active hydrolases like cathepsin B and D based on our Magic Red assay and western blot analysis. Degradation of EGF-TexasRed also occurred with similar kinetics as in control cells. In contrast to CHS disease, however, we could not detect any significant impairment in degranulation in patient-derived CD8[+] T cells and NK cells despite the observation that patient-derived cells stayed slightly below the range of healthy controls. Based on our current data, we believe that LRBA does not result in the same level of degranulation deficiency as seen in other BEACH-related diseases, such as CHS or Grey Platelet syndrome caused by mutations in NBEAL2 (Sowerby et al., 2017).

While lysosomal dysfunction was described for a plethora of diseases among which are SLE (Wang and Muller, 2015), Sjörgen disease (Sohar et al., 2005), Crohn's disease (Lassen et al., 2016), and rheumatoid arthritis (Ansari et al., 2021), the enlarged endolysosomes in CHS and LRBA deficiency are functional in terms of protein degradation. Enlarged endosomes/endolyosomes appear to be a common feature of BEACH-related diseases. It is tempting to speculate that the defect in α-granule precursor

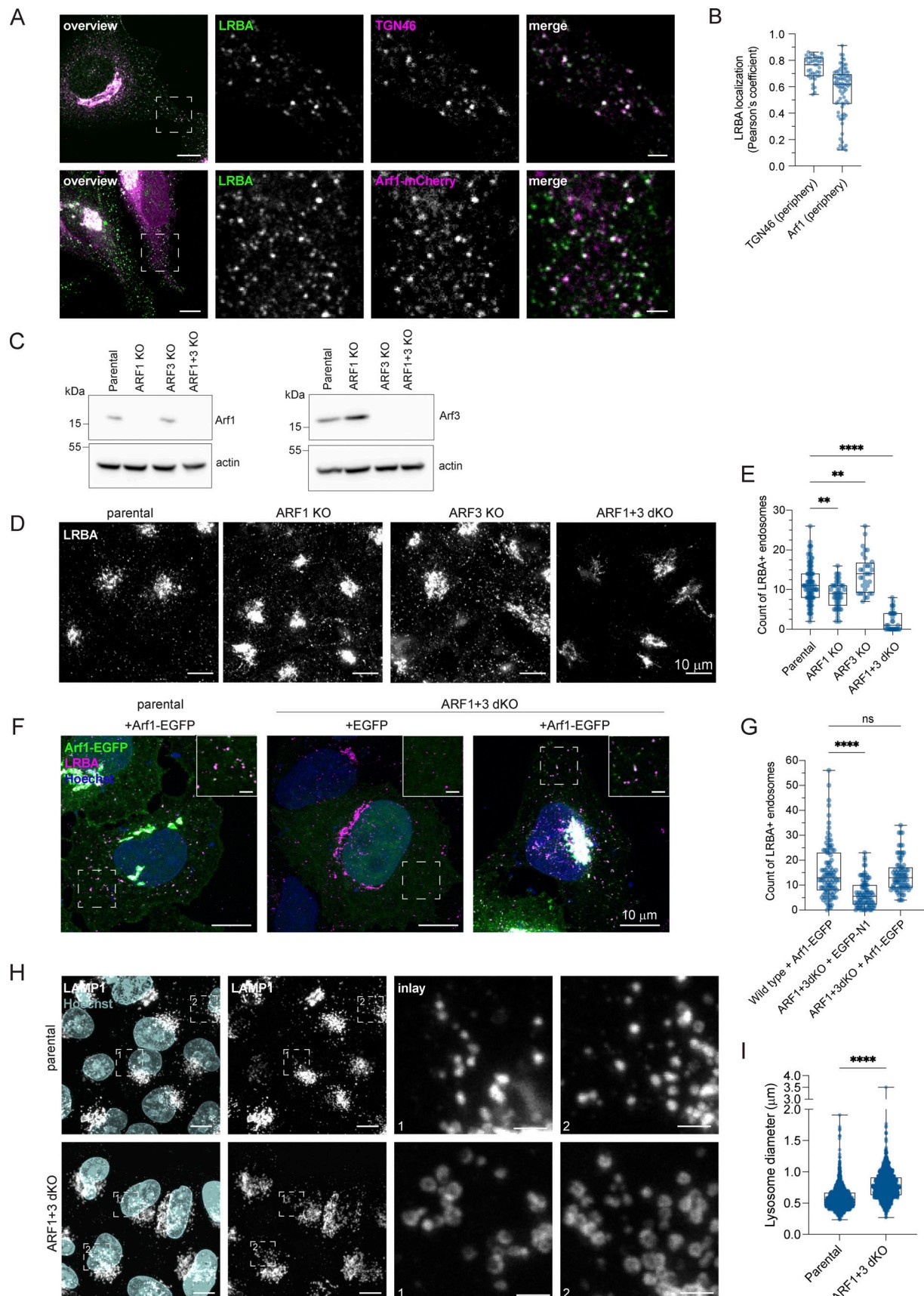

Figure 7. **LRBA is recruited onto endosomes by Arf1 and Arf3. (A)** Colocalization analysis of LRBA and TGN46 or Arf1 on endosomes in HeLa cells. For the colocalization analysis with the TGN, HeLa cells were fixed with 4% PFA and stained for endogenous TGN46 and LRBA. For colocalization analysis with Arf1,

HeLa cells were transfected with ARF1-mCherry, fixed with 4% PFA and stained for endogenous LRBA. Squares show magnification of the perinuclear area. The labeling of the single channels represents the color of the channel on the merged image. Scale bar, 10 μm, inlays 2 μm. **(B)** Colocalization measurements of LRBA with TGN46 and Arf1 at the cell periphery. To measure LRBA colocalization with TGN46 one ROI per image, with Arf1 two ROIs per image were analyzed and the Pearson's coefficient was measured using the JACoP plugin in Fiji. Mean and minimum to maximum are shown, the box ranges from the first (Q1–25th percentiles) to the third quartile (Q3–75th percentiles) of the distribution. All data points are shown. TGN46 (periphery) = 40 cells, Arf1 (periphery) = 35 cells from $n$ = 3 biological replicates. **(C)** Immunoblot analysis of Arf1 and Arf3 expression in parental, ARF1 KO, ARF3 KO, and ARF1+3 dKO HeLa cells. Actin was used as a loading control. **(D)** LRBA is absent from endosomes in ARF1 and ARF3 dKO HeLa cells. Note that LRBA is still present on the Golgi. Parental, ARF1 KO, ARF3 KO, and ARF1+3 dKO HeLa cells were seeded on coverslips, fixed, and stained for endogenous LRBA. Maximum intensity projections of confocal images are shown. **(E)** The number of LRBA[+] endosomes in parental, ARF1 KO, ARF3 KO, and ARF1+3 dKO cells was measured using two ROIs per cell. Mean and minimum to maximum are shown, box ranges from the first (Q1–25th percentiles) to the third quartile (Q3–75th percentiles) of the distribution. All data points are shown. Parental = 95 cells, ARF1 KO = 49 cells, ARF3 KO = 32 cells, and Arf1+3 dKO = 34 cells were analyzed from $n$ = 3 biological replicates; one-way ANOVA using Dunnett's multiple comparison, **P = 0.0010 (parental versus ARF1 KO), **P = 0.0097 (parental versus ARF3 KO), ****P < 0.0001. **(F)** Arf1-EGFP re-expression rescues LRBA[+] endosomes absent in ARF1+3 dKO cells. Parental HeLa cells were transfected with Arf1-EGFP, and ARF1+3 dKO cells were transfected either with EGFP as a control or with Arf1-EGFP. Cells were then fixed and stained for endogenous LRBA and with Hoechst. Maximum intensity projection of confocal Z-stacks is shown. Rectangles show the magnified area in the upper right corner. Scale bar on inlays 2 μm. **(G)** The number of LRBA[+] puncta are counted based on data in F. Two ROIs at the cell periphery per cell are analyzed. Mean and minimum to maximum are shown, box ranges from the first (Q1–25th percentiles) to the third quartile (Q3–75th percentiles) of the distribution. All data points are shown. Parental = 49 cells, EGFP rescue = 42 cells, Arf1-EGFP rescue = 39 cells were analyzed from $n$ = 3 biological replicates; Kruskal–Wallis test using Dunn's multiple comparison, ****P < 0.0001. **(H)** (Endo) lysosomal structures are enlarged in ARF1+3 dKO cells. Parental and ARF1+3 dKO HeLa cells were seeded on coverslips, fixed, and stained for LAMP1 and with Hoechst. Maximum intensity projections of confocal Z-stacks. Squares show magnification of the (endo)lysosomes. Scale bar, 10 μm, inlays 2 μm. **(I)** Quantification of lysosome diameter based on images shown in H. The diameter of round lysosomes was measured manually in Fiji. Mean and minimum to maximum are shown, box ranges from the first (Q1–25th percentiles) to the third quartile (Q3–75th percentiles) of the distribution. Parental = 60 cells, ARF1+3 dKO = 64 cells were analyzed from $n$ = 3 biological replicates. All data points are shown. Mann–Whitney test, ****P < 0.0001. Source data are available for this figure: SourceData F7.

biogenesis, which is the underlying cause of Grey Platelet syndrome in NBEAL2 deficiency (Lo et al., 2018), is due to the enlarged endosomes/endolysosomes that also fail to mature into α-granule precursors. Thus, it is plausible that the underlying cause of BEACH-related diseases is the inability to form functional lysosome-related organelles (LROs), such as cytolytic granules in CHS and α-granules in NBEAL2 deficiency (Bowman et al., 2019). Which LRO cannot be formed in LRBA deficiency remains elusive. Moreover, how the observed perturbed endosomal traffic and the endolysosome accumulation lead to the manifestation of LRBA deficiency syndrome needs further investigation. It will be important to determine how LRBA regulates sorting on Rab4[+] endosomes. Mutating Ile[46] and Ile[49] to Ser in Arf1 did not abolish the recruitment of Arf1 onto the Rab4[+] endosomes, but eliminated the recruitment of LRBA onto these structures, confirming the Alphafold prediction. Our data are consistent with a direct interaction and that LRBA might be an effector of Arf1 on Rab4[+] endosomes. Our study highlights the importance of improving transport through the endosomal system in the search for therapeutic options in LRBA deficiency.

## Materials and methods

### Patient material

Informed consent was available according to the approved protocols from the local institutional review board of Goethe University Frankfurt, Germany (IRB # 436/16). Skin fibroblast and peripheral blood samples were analyzed. Patient 1 is currently a 26-year-old female of Egyptian descent whose disease aggravated over the past years needing an alloHSCT at the age of 24 years, which was successful. Patient 2 is currently a 5-year-old boy of Libyan descent who developed severe early-onset inflammatory bowel disease, also necessitating alloHSCT. Both patients had received abatacept prior to alloHSCT. The clinical course of Patient 1 has been previously published by our group (Bakhtiar et al., 2016). Patients 3 and 4 are 14- and 17- year-old non-transplanted patients under abatacept treatment, their blood samples were analyzed for the degranulation assays.

### Cell culture

Dermal fibroblasts were isolated from skin biopsies of LRBA deficient patients 1 and 2 by explant culture and healthy donors according to the protocol (Zöller et al., 2008). The fibroblasts were cultured in high-glucose Dulbecco's modified Eagle's medium high glucose (4.5 g/l) (Sigma-Aldrich) with 5% (vol/vol) fetal bovine serum (FBS), 100 U ml⁻¹ penicillin G, and 100 ng ml⁻¹ streptomycin, at 37°C and 7.5% $CO_2$ and tested for mycoplasma.

*ARF1, ARF3,* and *ARF1+3* double KO HeLaα cells were established, mycoplasma tested, and described elsewhere (Pennauer et al., 2022). HeLa CCL2 and HEK293A cells were kindly provided by Dr. Martin Spiess, with its identity authenticated by STR analysis by Microsynth AG. Cell lines were tested for mycoplasma.

LRBA KO, wild type HeLa CCL2, ARF1, ARF3 and ARF1+3 double KO, parental HeLaα cells, HEK293A, and HeLa Flp-In-T-REx 3xFlag-EGFP-LRBA were grown in high-glucose Dulbecco's modified Eagle's medium (#D5796; Sigma-Aldrich) with 10% fetal bovine serum (FBS; Biowest), 2 mM *L*-glutamine (#25030; Gibco), 100 U ml⁻¹ penicillin G and 100 ng ml⁻¹ streptomycin (#P4333; Sigma-Aldrich Chemie GmbH), and 1 mM sodium pyruvate (#S8636; Sigma-Aldrich Chemie GmbH) at 37°C and 5% $CO_2$.

Expression of 3xFlag-EGFP-LRBA was induced with 1 μg/ml tetracycline in the HeLa Flp-In-T-REx 3xFlag-EGFP-LRBA cell line (kindly provided by Dr. Serhiy Pankiv and Prof. Anne G. Simonsen, University of Oslo, Norway).

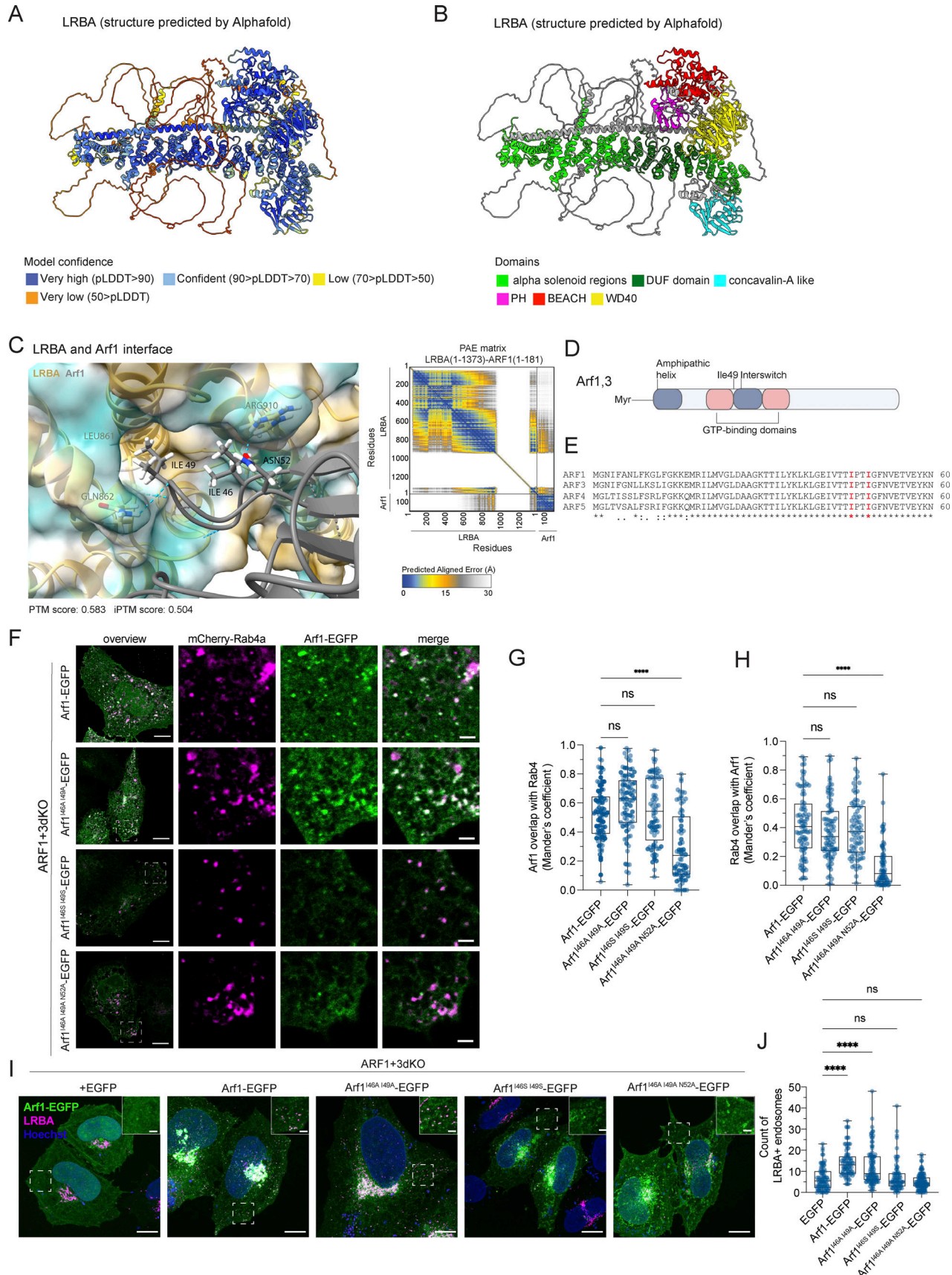

Figure 8. **Structure of LRBA and potential interaction site with Arfs as predicted by Alphafold. (A and B)** Alphafold monomer prediction of human LRBA structure. Model confidence values (A) and domains (B) are indicated with colors. **(C)** Predicted interaction sites between LRBA (amino acids 1–1373) and Arf1

(amino acids 1–181) using Alphafold multimer. PAE plot shows the confidence scores of the interaction. The LRBA–ARF1 model has PTM and iPTM scores of 0.583 and 0.504, respectively. **(D)** Schematic of Arf1/3 structure and its domains. The conserved amino acid isoleucine 49 of Arf1/3 is shown since most of our models indicated that it could potentially interact with LRBA. **(E)** The predicted LRBA interaction site of Arf1, isoleucine 46 and 49 are conserved among human Arf1, Arf3, Arf4, and Arf5. Sequence alignments were performed using Clustal Omega. Labels: (*) conserved sequence; (:) conservative mutation; (.) semi-conservative mutation; (-) gap. **(F)** Colocalization analysis of mutated Arf1-EGFP constructs with Rab4$^+$ endosomes. ARF1+3dKO HeLa cells were transfected with mCherry-Rab4a and with Arf1-EGFP constructs. Confocal images of single focal planes. Scale bar, 10 μm, inlays 2 μm. **(G and H)** Colocalization measurement of Rab4 and Arf1 mutants determined by Mander's coefficients. Arf1$^{I46A, I49A}$-EGFP and Arf1$^{I46S, I49S}$-EGFP are still efficiently recruited to Rab4$^+$ endosomes. Arf1$^{I46A, I49A, N52A}$-EGFP recruitment to Rab4 endosomes is strongly reduced compared with wild-type Arf1-EGFP. Arf1-EGFP = 42 cells, Arf1$^{I46A, I49A}$-EGFP = 43 cells, Arf1$^{I46S, I49S}$-EGFP = 38 cells, and Arf1$^{I46A, I49A, N52A}$ = 36 cells were analyzed from $n$ = 3 biological replicates, one-way ANOVA using Tukey's multiple comparison test; ****$P < 0.0001$. **(I)** Arf1$^{I46S, I49S}$-EGFP fails to rescue LRBA$^+$ puncta in ARF1+3 dKO cells. ARF1+3 dKO cells were transfected with EGFP, Arf1-EGFP, Arf1$^{I46A, I49A}$-EGFP, Arf1$^{I46S, I49S}$-EGFP or with Arf1$^{I46A, I49A, N52A}$-EGFP, fixed and stained for endogenous LRBA and with Hoechst. Scale bar, 10 μm, inlays 2 μm. **(J)** Quantification of the number of LRBA$^+$ puncta based on images as shown on panel I. Two ROIs at the cell periphery per cell were analyzed. Mean and minimum to maximum are shown, box ranges from the first (Q1–25th percentiles) to the third quartile (Q3–75th percentiles) of the distribution. All data points are shown. EGFP = 42 cells, Arf1-EGFP = 39 cells, Arf1$^{I46A, I49A}$-EGFP = 48 cells, Arf1$^{I46S, I49S}$-EGFP = 36 cells, Arf1$^{I46A, I49A, N52A}$ = 49 cells were analyzed from $n$ = 3 independent experiments; Kruskal–Wallis test using Dunn's multiple comparison, ****$P < 0.0001$. EGFP and Arf1-EGFP measurements are identical to the one shown in Fig. 7 G as the data were obtained in the same experiment.

Heparin-anticoagulated blood samples were obtained from patients and healthy donors and peripheral blood mononuclear cells (PBMCs) were isolated from whole blood within 24 h. Briefly, heparin-blood was diluted 1:2 with Dulbecco's phosphate-buffered saline (DPBS), layered on cell separation solution medium (Bicoll, Biochrom) following centrifugation for 20 min at 800 $g$ without deceleration. After removing the top layer, mononuclear cells containing the ring were collected and washed twice with DPBS. The number of isolated PBMCs was determined using a cell counter (DxH500; Beckman Coulter). Cells were cultured in RPMI 1640 medium (Gibco) supplemented with 10% FBS.

### CRISPR-Cas9 KO cell line generation

For CRISPR-Cas9–mediated KO, guide RNAs were selected using the GenScript CRISPR design tool. Two guide RNAs were designed from two different exons for the target LRBA gene, gRNA against exon 22: 5′-CCATGCAGTCAAATATGAGT-3′, and gRNA against exon 38: 5′-GGTTACGCACAAATCGTCGC-3′. Annealed oligonucleotides were cloned into Px458-GFP vector and Px459-Puro vector using the BbsI cloning site. In brief, 7.5 × 10$^5$ HeLa cells were seeded per 10-cm dish. The following day, cells were transfected with 6 μg of the plasmids (3–3 μg for both targeted exons of LRBA) with Helix IN transfection reagent (#HX10100; OZ Biosciences). For control cells, control vectors without gRNA insert were transfected. For selection, cells were treated with 1.5 μg/ml puromycin for 24 h before FACS sorting (for GFP$^+$ cells). FACS sorting was carried out 48 h after transfection. Cells were trypsinized and resuspended in cell-sorting medium (2% FCS and 2.5 mM EDTA in PBS) and sorted on a BD FACS Aria Fusion Cell Sorter. GFP$^+$ cells were collected and seeded into a 96-well plate. Cells were then expanded and LRBA expression was determined by western blot analysis.

### Antibodies

The following antibodies were used in this study: sheep polyclonal anti-TGN46 (Cat# AHP500G, RRID:AB_323104, 1:2,000; Bio-Rad), rabbit polyclonal LRBA antibody (Cat# HPA023567, RRID:AB_1853256, 1:1,000; Atlas Antibodies), mouse AP1 100/3 hybridoma antibody (home-made, 1:2,000), rabbit monoclonal

EEA1 (C45B10) antibody (Cat# 3288, RRID:AB_2096811, 1:2,000; Cell Signaling Technology), rabbit monoclonal anti-LAMP1 (D2D11) XP antibody (Cat# 9091, RRID:AB_2687579, 1:200 for immunofluorescence, 1:1,000 for western blot; Cell Signaling Technology), rabbit monoclonal Rab7 (D95F2) XP antibody (Cat# 9367, RRID:AB_1904103, 1:400 for immunofluorescence, 1:1,000 for western blotting; Cell Signaling Technology), mouse monoclonal anti-M6PR (cation independent) (2G11) antibody (Cat# ab2733, RRID:AB_2122792, 1:1,000; Abcam), rabbit polyclonal anti giantin antibody (Cat# 924302, RRID:AB_2565451, 1:500; BioLegend), goat polyclonal anti Vps35 antibody (Cat# ab10099, RRID:AB_296841, 1:50; Abcam), rabbit monoclonal anti-EGF receptor (D38B1) XP antibody (Cat# 4267, RRID:AB_2246311, 1:1,000; Cell Signaling Technology), rabbit monoclonal phospho-EGF receptor (Tyr1068) (D7A5) XP antibody (Cat# 3777, RRID:AB_2096270, 1:1,000; Cell Signaling Technology), mouse monoclonal anti p44/42 MAPK (Erk1/2) (L34F12) antibody (Cat# 4696, RRID:AB_390780, 1:2,000; Cell Signaling Technology), rabbit polyclonal anti phospho-p44/42 MAPK (Erk1/2) (Thr202/Tyr204) antibody (Cat# 9101, RRID:AB_331646, 1:1,000; Cell Signaling Technology), mouse anti-calnexin antibody clone 37 (Cat# 610523, RRID:AB_397883, 1:2,000; BD Biosciences), mouse monoclonal anti-Pan Actin (#LCU9001, 1:1,000; Linaris), mouse monoclonal anti-TCIRG1/lysosomal ATPase V0 subunit a3 antibody (M01), clone 6H3 (Cat# H00010312-M01, RRID:AB_464374, 1:100; Abnova), rabbit monoclonal recombinant anti-cathepsin D (EPR3057Y) antibody (Cat# ab75852, RRID:AB_1523267, 1:100 for immunofluorescence, 1:2,000 for western blot; Abcam), mouse monoclonal anti-Arf1 antibody clone AT1B3 (Cat# MAB10011, RRID:AB_10901483, 1:2,500; Abnova), mouse monoclonal anti-Arf3 antibody (Cat# 610784, RRID:AB_398105, 1:1,000; BD Biosciences), rabbit polyclonal anti-GFP (Cat# TP401, RRID:AB_10013661, 1:5,000; Torrey Pines Biolabs), goat anti-rabbit IgG(H+L) secondary antibody, AlexaFluor488 (#A-11034, 1:500; Invitrogen), goat anti-mouse IgG (H+L) secondary antibody, AlexaFluor488 (#A11001, 1:500; Invitrogen), goat anti-mouse IgG (H+L) secondary antibody, AlexaFluor633 (#A-21052, 1:500; Invitrogen), donkey anti-goat IgG (H+L) secondary antibody, AlexaFluor488 (#A11055, 1:500; Invitrogen), donkey anti-mouse IgG (H+L) secondary antibody, AlexaFluor568 (#A10037, 1:500;

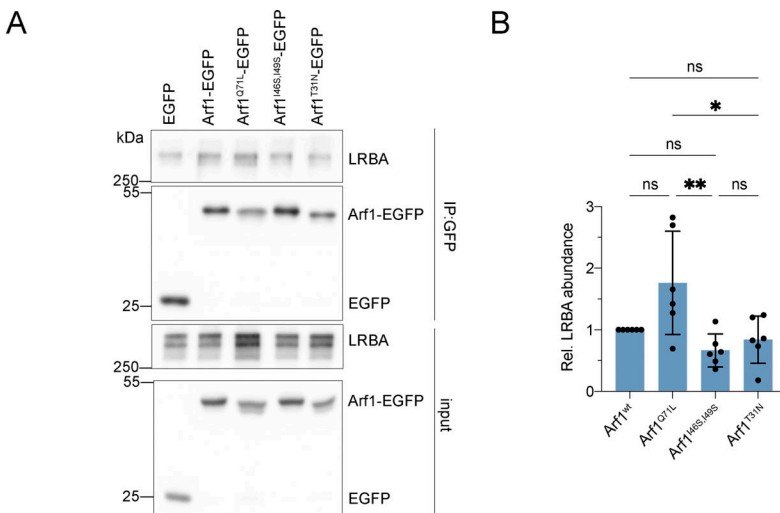

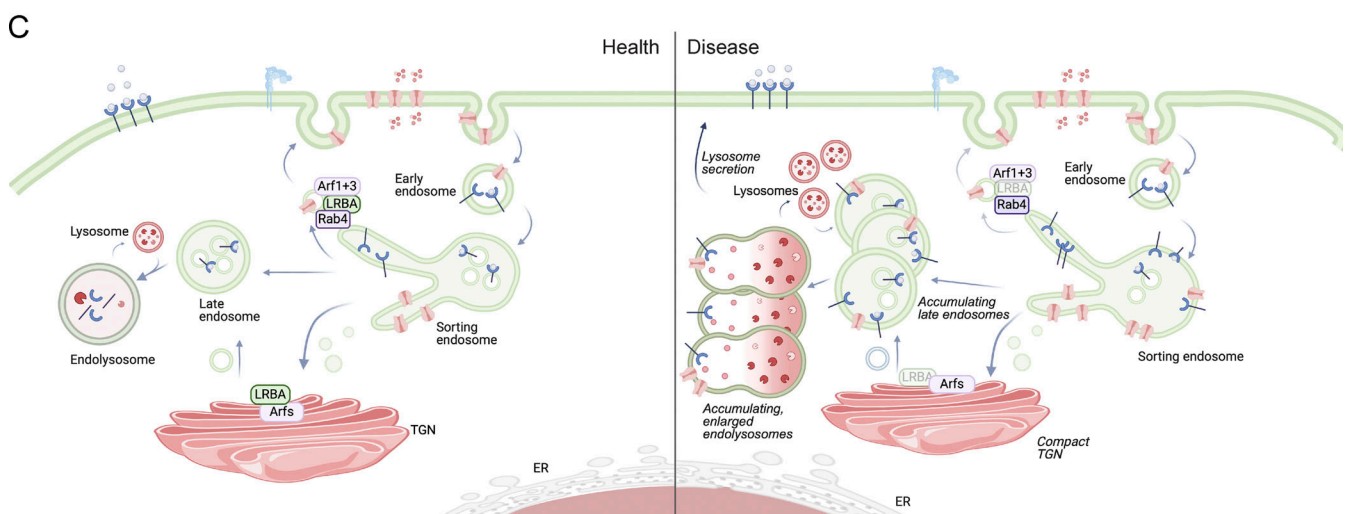

Figure 9. **Arf1 coimmunoprecipitates with LRBA. (A)** Coimmunoprecipitation analysis of Arf1-EGFP, Arf1$^{Q71L}$-EGFP (constitutively active [CA]), Arf1$^{I46S\ I49S}$-EGFP binding mutant and Arf1$^{T31N}$-EGFP (dominant-negative [DN]). Arf1-EGFP constructs were pulled-down using GFP-trap magnetic beads. The eluted proteins were immunoblotted and Arf1 constructs were detected using polyclonal GFP antibodies. LRBA was detected by the polyclonal LRBA antibody. Wild-type and the constitutively active Arf1 interact with LRBA. This interaction is reduced with the Arf1$^{I46S\ I49S}$-EGFP and Arf1$^{T31N}$-EGFP DN mutant. **(B)** Quantification of immunoblots as shown on panel (A) from $n$ = 6 biological replicates, one-way ANOVA using Tukey's multiple comparison test, **$P$ = 0.0040, *$P$ = 0.0163. **(C)** Model of endosomal trafficking in the presence and in the absence of LRBA. Source data are available for this figure: SourceData F9.

Invitrogen), donkey anti-sheep IgG (H+L) secondary antibody, AlexaFluor568 (#A21099, 1:500; Invitrogen), mouse monoclonal alpha-tubulin antibody (Cat# T5168, RRID:AB_477579, 1:10,000; Sigma-Aldrich), goat anti-mouse IgG, (H+L), HRP-coupled (#31430, 1:10,000; Pierce/Invitrogen), goat anti-rabbit IgG, (H+L), HRP-coupled (#31460, 1:10,000; Pierce/Invitrogen), anti-CD107a-PB (Cat#B13978 RRID: AB_3105968; Beckman Coulter), anti-CD45-KrO (Cat# B36294, RRID:AB_2833027; Beckman Coulter), anti-CD3-APC (Cat# IM2467, RRID:AB_130788; Beckman Coulter), anti-CD8-APC-AF700 (Cat#B49181 RRID: AB_3106368; Beckman Coulter), anti-CD69-PE (Cat# IM1943U, RRID:AB_2801272; Beckman Coulter), and CD56-PC7 (Cat# A21692, RRID:AB_2892144; Beckman Coulter).

**DNA and plasmid sources**
The following commercially available plasmids were obtained: mApple-Rab5a (#54944; Addgene), mApple-Rab7a (#54945; Addgene), LAMP1-GFP (#34831; Addgene), mCherry-Rab11a (#55124; Addgene, mCherry-Rab4a [#55125; Addgene], pSpCas9(BB)–2 A-GFP [pX458]) (#48138; Addgene), pSpCas9(BB)–2A-puro (pX459) (#48139; Addgene), and pEGFP-N1-Arf1 (#39554; Addgene).

pEGFP-N1 vector, Arf1$^{T31N}$-EGFP, and Arf1$^{Q71L}$-EGFP were a kind gift from Dr. Martin Spiess.

Arf1-mCherry was cloned by replacing EGFP in the pEGFP-N1-Arf1 (#39554; Addgene) plasmid with mCherry. mCherry was amplified by PCR from the mCherry-Rab4a (#55125;

Addgene) plasmid and inserted with NEBuilder HiFi Assembly cloning kit (#E5520S; New England Biolabs).

Arf1$^{I46A, I49A}$-EGFP, Arf1$^{I46S, I49S}$-EGFP, and Arf1$^{I46A, I49A, N52A}$-EGFP mutants were generated in the pEGFP-N1-Arf1 (#39554; Addgene) using PCR KAPA HiFi DNA Polymerase and its Hot-Start ReadyMix. Primers were designed with the QuikChange Primer Design Program from Agilent, Arf1$^{I46A, I49A}$-EGFP f: 5′-GTTTCCACGTTGAAGCCTGCGGTGGGAGCGGTGGTCACGATCTCACC-3′, r: 5′-GGTGAGATCGTGACCACCGCTCCCACCGCAGGCTTCAACGTGGAAAC-3′; Arf1$^{I46S, I49S}$-EGFP f: 5′-CCACGTTGAAGCCGCTGGTGGGACTGGTGGTCACGATCTCA-3′, r: 5′-TGAGATCGTGACCACCAGTCCCACCAGCGGCTTCAACGTGG-3′; Arf1$^{I46A, I49A, N52A}$-EGFP f: 5′-CTCCACGGTTTCCACGGCGAAGCCTGCGGTGGGAGCGGTGGTCACGATCTCA-3′, r: 5′-TGAGATCGTGACCACCGCTCCCACCGCAGGCTTCGCCGTGGAAACCGTGGAG-3′.

## TEM

Fibroblasts were grown to 90% confluency on fibronectin (#F1141; Sigma-Aldrich) coated 18-mm round glass coverslips. Cells were fixed by adding a warm double-strength fixative solution (5% glutaraldehyde [GA, #16310; Electron Microscopy Sciences] and 4% paraformaldehyde [PFA, #15710; Electron Microscopy Sciences] in 0.1M PIPES buffer [#P6757; Sigma-Aldrich supplemented with 2 mM CaCl$_2$, pH 7–7.3]) to the culture medium into the dish (ratio 1+1) and mix very gently and incubated for 15 min at room temperature. Then the fixative-medium mixture was replaced by fresh single-strength fixative solution (2% PFA, 2.5% GA in 0.1 M PIPES) and incubated for 2 h at RT and for 16 h at 4°C. Then the cells were washed three times for 10 min with cold 0.1 M PIPES buffer. Fixed cells were rinsed first in PIPES buffer and then once in cacodylate buffer (0.1 M, pH 7.3) for 10 min. After two additional washes in cacodylate buffer, cells were post-fixed in 1% osmium tetroxide (#19100; Electron Microscopy Sciences), 0.8% potassium ferracyanide in 0.1 M cacodylate buffer for 1 h at 4°C. Coverslips were rinsed several times in cacodylate buffer and ultrapure distilled water then, en block stained with 1% aqueous uranyl acetate for 1 h at 4°C in the dark. After several wash steps in ultrapure distilled water, cells were dehydrated in an ethanol series (30%, 50%, 75%, 95%, and 100%, #15056; Electron Microscopy Sciences) at 4°C followed by three additional changes of absolute ethanol. Samples were washed in acetone (#15056; Electron Microscopy Sciences) and finally embedded in a mixture of resin/acetone first and then in pure Epon 812 resin (#14120; Electron Microscopy Sciences) overnight. Coverslips were placed cell-side down on BEEM capsules (#70010-B; Electron Microscopy Sciences) filled with EPON. Polymerization was carried out for 48 h at 60°C. After complete polymerization, coverslips were removed from the EPON block using the nitrogen-hot water method. During the removal of the coverslip from the EPON block, cells were transferred from the coverslip to the block surface. 70-nm thin serial sections were cut with a diamond knife and placed on formvar-carbon coated copper grids, stained with uranyl acetate (#22400; Electron Microscopy Sciences) and Reynolds's lead citrate and observed in a FEI Tecnai G2 Spirit Transmission Electron Microscope (TEM) operating at 80 kV. Images were acquired using an EMSIS Veleta camera (top, side-mounted). The Camera operates using the RADIUS software from EMSIS.

## Immunostaining

HeLa cells were plated onto coverslips 24 h prior to fixation. When indicated, fixation of transfected cells was performed 48 h after cell plating. Cells were fixed in 4% paraformaldehyde, permeabilized with 0.1 % Triton X-100, blocked in PBS containing 5% FBS, and stained with the indicated primary antibodies followed by AlexaFluor conjugated secondary antibodies. Coverslips were mounted onto glass slides with Fluoromount G (Southern Biotech) or Vectashield and sealed with nail polish.

To visualize endolysosomes in LRBA-deficient fibroblasts, cells were plated onto fibronectin-coated coverslips 24 h before fixation. Cells were then fixed with ice-cold 100% methanol on ice for 15 min, rinsed 3 × 5 min with PBS, blocked in PBS containing 5% FBS for 1 h, and stained with the indicated primary antibodies diluted in 0.2% saponin, 0.1% BSA, and 0.02% NaN$_3$ in PBS (blocking reagent) overnight at 4°C. The next day coverslips were washed 3 × 5 min with PBS and stained with AlexaFluor conjugated secondary antibodies diluted in blocking reagent. Coverslips were mounted onto glass slides with Vectashield and sealed with nail polish.

Nuclei were stained with Hoechst 33342 Fluorescent Nucleic Acid Stain (Cat# 639, RRID:AB_2651135, 1:200; ImmunoChemistry technologies).

## Microscopy

Confocal images were acquired with Olympus Fluoview FV3000 system using an UPLSAPO 60×/1.30 objective with silicone oil and FV31S-SW software. The sampling speed was 8.0 μs/pixel. Arf1 rescue experiments were acquired with a Leica Stellaris point scanning confocal microscope equipped with a white light laser, a 405 DMOD laser, and an HC PL APO 63×/1.40 oil objective using LAS X acquisition software. 2 Power HyD S detector for imaging Hoechst and EGFP and Power HyD X detector for mCherry and AlexaFluor 594 detection were set.

Live-cell imaging of Arf1-EGFP recruitment onto mCherry-Rab4 structures in control and LRBA KO HeLa cells was performed in complete growth medium lacking phenol red at 37°C with 5% CO$_2$ using an inverted Axio Observer microscope (Zeiss) with a Plan Apochromat N 63×/1.40 oil DIC M27 objective, a Photometrics Prime 95B camera, and Zen 2.6 acquisition software. Filters with standard specifications for GFP and TexasRed were used to image EGFP and mCherry respectively.

All images are representative of at least three independent sets of experiments. All images for corresponding experiments were processed with the same settings to ensure comparable results.

## Image analysis

Golgi morphology analysis was based on confocal Z-stack images of TGN46. Different Golgi phenotypes were categorized according to their extension around the nucleus (extended, half-moon, and compact).

For the TGN radar plots, two-pixel thick segmented lines with spline fit were drawn around the full perimeter of the

nucleus starting at the opposite side of the nucleus from where Golgi localized, and histogram measurements were obtained of fluorescence intensity along the length of the line in Fiji (Schindelin et al., 2012). Radar plots were drawn based on a representative healthy and a representative patient Golgi with Microsoft Excel.

To measure the volume and the mean intensities of the TGN, Z-stacks at 0.13 µm per slice were acquired and analyzed in Fiji using the "3DGolgiCharacterization" script (https://doi.org/10.5281/zenodo.10566786, Guerard, 2024a).

For colocalization analysis of M6PR and TGN46, M6PR, and Vps35 in LRBA deficient fibroblasts, Arf1-EGFP mutant constructs and mCherry-Rab4 in ARF1+3 dKO HeLa cells, and Arf1-EGFP and mCherry-Rab4 in LRBA KO HeLa cells, a rectangular ROI at the perinuclear area was drawn and Mander's coefficient was obtained using the JACoP plugin in FIJI (Bolte and Cordelières, 2006; Schindelin et al., 2012).

For colocalization analysis of Arf1-EGFP and mCherry-Rab4 in LRBA KO HeLa cells, wide-field images were deconvolved using Huygens Remote Manager 3.10.0 (Ponti et al., 2007). Parameters were set as follows: theoretical PSF, automatic aberration correction, classic maximum likelihood estimation, 50 iterations, quality factor 0.0002, acuity mode: 50, automatic background estimation. Using the deconvolved images, a rectangular ROI at the perinuclear area was drawn and Mander's coefficient was obtained using the JACoP plugin in FIJI (Bolte and Cordelières, 2006; Schindelin et al., 2012).

To measure EEA1 size, Z-stacks at 0.13 µm per slice were acquired and analyzed in Fiji using the "particleCount3D" script (https://doi.org/10.5281/zenodo.10566810, Guerard, 2024b). The average EEA1 size per cell represents a data point on the graph which was generated with GraphPad Prism 10.

To measure the colocalization of LRBA and TGN46 or Arf1, a rectangular ROI at the Golgi was drawn and Pearson's coefficient was obtained using the JACoP plugin in FIJI (Bolte and Cordelières, 2006; Schindelin et al., 2012). Pearson's coefficients between LRBA and endosomal markers were determined in two ROI s per cell at the cell periphery using the JACoP plugin in FIJI. Values per ROI were plotted using GraphPad Prism 10.

To count LRBA+ endosomes in parental, ARF1 KO, ARF3 KO, and ARF1+3 dKO cells and in the rescue experiments, re-expressing EGFP, Arf1-EGFP, Arf1$^{I46A, I49A}$-EGFP, Arf1$^{I46S, I49S}$-EGFP, and Arf1$^{I46A, I49A, N52A}$-EGFP images were thresholded and the number of endosomes were counted within two ROIs per cell outside of the Golgi area with analyze particles plugin using FIJI (Schindelin et al., 2012). Datapoints per ROI were plotted using GraphPad Prism 10.

To determine lysosome diameter in parental and ARF1+3 dKO HeLa cells, a line ROI was drawn in Fiji (Schindelin et al., 2012) through the lysosome, and the length of ROI was noted. Single lysosome diameters were plotted using GraphPad Prism 10.

All image panels were assembled using the OMERO image data management system (Allan et al., 2012).

## Live-cell imaging of BFA and GCA inhibition
For live-cell imaging, cells were seeded on an ibidi µ-slide VI 0.4 channel slide (#80606; Ibidi) and tetracycline induced (1 µg/ml)

for LRBA expression 24 h prior to data acquisition. Live-cell imaging was performed in a complete growth medium lacking phenol red at 37°C with 5% $CO_2$ using an inverted Axio Observer microscope (Zeiss) with a Plan Apochromat N 63×/1.40 oil DIC M27 objective and a Photometrics Prime 95B camera. Filters with standard specifications for GFP were used to image 3xFlagEGFP-LRBA. After selecting cells to be imaged, first, a snapshot was taken and then BFA (at 2 µg/ml final concentration) or GCA (at 10 µM final concentration) was applied and cells were imaged at the indicated timepoints.

## Live-cell imaging of lysosomes using Magic Red and Lysotracker Green
For live-cell imaging, cells were seeded on an imaging chamber (ibidi µ-slide) 24 h prior to data acquisition. Magic Red (#SKU: 937; ImmunoChemistry Technologies) and Lysotracker Green (10 nM, #L7526; Invitrogen) were diluted in an imaging medium (phenol-red free complete media). This staining solution was topped up with fresh imaging medium after 1 h incubation. Live-cell imaging was performed at 37°C with 5% $CO_2$ using an inverted Axio Observer microscope (Zeiss) with a Plan Apochromat N 63×/1.40 oil DIC M27 objective and a Photometrics Prime 95B camera. Filters with standard specifications for GFP and dsRed were used to image LysoTracker Green or MagicRed signal. To measure lysosome distribution, Magic Red and LysoTracker Green signals were quantified by drawing a line starting from the nucleus toward the cell periphery in Fiji (Schindelin et al., 2012). Histogram measurements were obtained by plotting the fluorescence intensity along the length of the line and normalized to the maximum value of each cell which were then averaged to each cell line per experiment. Graphs were made using GraphPad Prism 10.

## Golgi reassembly analysis
Fibroblasts were seeded onto fibronectin-coated glass coverslips and let adhere overnight. Coverslips were treated with 10 µM golgicide A or DMSO (diluted in complete media) for 2 h (two coverslips at this point were fixed). Cells were washed 3× with PBS and incubated in complete media for indicated time points. After incubation, coverslips were rinsed with PBS and fixed with 4% PFA for 10 min RT. Samples were washed 3× with PBS and permeabilized using 0.1% Triton in PBS for 5 min at RT. Coverslips were rinsed once with PBS, blocked with 5% FBS for 1 h and incubated with anti-TGN46 antibody (#AHP500GT, 1:2,000; BioRad) at 4°C overnight. Coverslips were washed 3× with PBS prior to secondary antibody incubation. After 1 h incubation with AlexaFluor568 secondary antibody, coverslips were washed 3 × 5 min with PBS and mounted with Vectashield (#H-1000; Vector Laboratories, Inc.) mounting medium containing Hoechst33342 (#639; ImmunoChemistry Technologies, 200×) and sealed with nail polish. Coverslips were imaged using the confocal Olympus Fluoview FV3000 system, an UPLSAPO 60×/1.30 objective with silicone oil.

## EGF uptake assay
Fibroblasts were seeded on fibronectin-coated glass coverslips and let adhere overnight. Cells were washed 3× with PBS and

incubated in serum-free DMEM for 3 h. Cells were then washed 3× with ice-cold PBS and incubated with 2 μg/ml EGF-TexasRed (diluted in serum-free DMEM) at 4°C for 30 min. Then cells were washed three times with ice-cold PBS and endocytosis was stimulated with unlabeled 2 μg/ml EGF (diluted in serum-free DMEM) at 37°C for indicated time points (0–15–30–60 min). Cells were then rinsed fast with ice-cold PBS and fixed with 4% paraformaldehyde for 10 min at RT. Coverslips were washed three times with PBS and mounted in Vectashield containing Hoechst33342 dye to visualize nuclei and sealed with nail polish. Coverslips were imaged using the confocal Olympus Fluoview FV3000 system, an UPLSAPO 60×/1.30 objective with silicone oil. EGF uptake was quantified by measuring the integrated density of the fluorescence signal in each cell on images where the background signal was removed using Gaussian's blurring and image subtraction in FiJi (Schindelin et al., 2012). The values for individual cells were averaged and expressed as fold change of the values of healthy 0 min timepoints (normalized EGF-TexasRed intensity). Graphs were generated using GraphPad Prism 10.

### EGFR signaling measurements

Fibroblasts were seeded on 6 well plates and let adhere overnight. Cells were washed 3× with PBS the next day and incubated in serum-free DMEM overnight. Cells were stimulated with unlabeled 2 μg/ml EGF (diluted in serum-free DMEM) at 37°C for the indicated time points (0–5–15–30–45–60 min). Cells were then rinsed fast with ice-cold PBS and lysed using M-PER lysis buffer (supplemented with protease inhibitor (cOmplete EDTA-free cocktail protease inhibitor [#11873580001; Roche] and Halt phosphatase inhibitor [#78420; Thermo Fisher Scientific]) and incubated in a multi shaker for 10 min at 4°C. Lysates were centrifuged at 4°C and 13,000 rpm for 10 min. The protein concentration was determined in the supernatant using the BCA assay (#23228; Thermo Fisher Scientific). Lysates were adjusted to the same concentration with lysis buffer and diluted with Laemmli. Samples were denatured at 65°C for 10 min. Samples were resolved by 10% SDS-PAGE and transferred onto nitrocellulose membrane (Amersham) using wet transfer for 3 h. Membranes were blocked with TBST (20 mM Tris, 150 mM NaCl, pH 7.6, 0.1% Tween20) with 5% non-fat dry milk for 30 min and incubated with primary antibodies, rabbit monoclonal anti-EGF receptor (D38B1) XP antibody (#4267, 1:1,000; Cell Signaling), rabbit monoclonal phospho-EGF receptor (Tyr1068) (D7A5) XP antibody (#3777, 1:1,000; Cell Signaling), mouse monoclonal anti-p44/42 MAPK (Erk1/2) (L334F10) antibody (#4696, 1:2,000; Cell Signaling), rabbit polyclonal anti-phospho-p44/42 MAPK (Erk1/2) (Thr202/Tyr204) antibody (#9101, 1:1,000; Cell Signaling) overnight at 4°C. Mouse anti-calnexin antibody clone 37 (#610523, 1:2,000; BD Biosciences) was used as a loading control. After 2 h incubation with HRP-conjugated secondary antibody (1:10,000; anti-mouse or anti-rabbit, #31430 and #31460; Invitrogen) in TBST, chemiluminescence signals were detected using Western Bright ECL reagent (#K-12045-D50; Advansta) and imaged using a FusionFX (Vilber Lourmat). pEGFR and EGFR levels were measured with FIJI and normalized to calnexin loading controls (pEGFRnorm, EGFRnorm). Then pEGFRnorm values were normalized to EGFRnorm values and plotted over time. Similarly pERK and total ERK levels were measured and normalized to calnexin loading controls (pERKnorm, ERKnorm). Then pERKnorm values were normalized to ERKnorm values and plotted over time using GraphPad Prism 10.

### Western blot analysis

Cells were lysed 24 h after seeding with lysis buffer (1% Triton X-100, 150 mM NaCl, 20 mM Tris pH 7.5, 1 mM EDTA, 1 mM EGTA, Halt protease inhibitor (#78439; Thermo Fisher Scientific) and denatured in Laemmli buffer at 65°C for 10 min. Samples were resolved by 10% SDS-PAGE and transferred onto nitrocellulose membrane (Amersham). Membranes were blocked with TBST (20 mM Tris, 150 mM NaCl, pH 7.6, 0.1% Tween20) with 5% non-fat dry milk for 30 min and incubated with primary antibody, overnight at 4°C, followed by 2 h incubation with HRP-conjugated secondary antibody (1:10,000; anti-mouse or anti-rabbit, #31430 and #31460; Invitrogen ) in TBST. Chemiluminescence signals were detected using Western Bright ECL reagent (#K-12045-D50; Advansta) and imaged using a FusionFX (Vilber Lourmat). Western blots were quantified according to Stael et al. (2022) in Fiji (Stael et al., 2022).

### Total RNA isolation and qRT-PCR

Total RNA was extracted and purified from $2 \times 10^6$ fibroblasts cells using the RNeasy kit following manufacturer's instructions. RNA was subjected to reverse-transcription using GoScript reverse transcriptase (#A501C; Promega) primed with a mix of Oligo(dT)s (Promega) and random hexamers (Promega). qRT-PCR was performed using GoTaq qPCR master mix (#A600A; Promega) and primers specific for LRBA (forward: 5′-CCAACTTCAGAGATT TGTCCAAGC-3′; reverse: 5′-ATGCTGCTCTTTTTGGGTTCAG-3′) (Wang et al., 2004), MAB21L2 (forward: 5′-CCAGGTGGA AAACGAGAGTG-3′; reverse: 5′-GGTAGAGCACCACCTCAAATT C-3′) (Deml et al., 2015), GAPDH (forward: 5′-TCAAGGCTG AGAACGGGAAG-3′, reverse; 5′-CGCCCCACTTGATTTTGGAG-3′) (Dahn et al., 2020). qRT-PCR was carried out with the qTower 3 G Real-Time PCR Thermal Cycler (Analytik Jena). The changes in the expression levels were estimated by the ΔΔCt method.

### Alphafold modeling

To evaluate the structure of LRBA we used Alphafold monomer v2.2 developed by Deepmind (Evans et al., 2021, *Preprint*; Jumper et al., 2021). Due to the size of LRBA, we had to create three separate models, AA1-1566, AA1000-2000, and AA1567-2863. These models were stitched together using ChimeraX according to the highest reliability values (pLDDT) of each part. To evaluate the interaction between Arf1 or Arf3 and LRBA we used Alphafold multimer v2.2 (AF-M) developed by Deepmind. We had to run Arf1 or 3 with either AA1-1373 or AA1374-2863 of LRBA due to size limitations. We ran all five of AF-M models for three recycles with three seeding points resulting in 15 models per run. The models were evaluated using ChimeraX.

### Stimulation of lymphocytes for functional analyses

For functional analyses of CD8+ lymphocytes, $2 \times 10^5$ cells were resuspended in RPMI plus 10% FCS and cultured on a 96-well

plate. Anti-CD107a-PB (#B13978; Beckman Coulter) antibody and monensin (Golgi Stop, prediluted 1:10 with RPMI + 10% FCS) were added to all conditions. Stimulation included phorbol-12-myristat-13-acetat (PMA; #P8139-5M6; Sigma-Aldrich, concentration of 2 ng/µl) and ionomycin (#I0634-1M6 0.1 µg/µl; Sigma-Aldrich) for 4 h at 37°C and 5% $CO_2$. NK cell degranulation was analyzed according to an adapted protocol described by Bryceson et al. (2012). $2 \times 10^6$ PBMCs were either stimulated with IL-2 (#1447583; Novartis) at a concentration of 600 U/ml on a 24-well plate or kept in medium. After 16 h a manual cell analysis was performed using a Neubauer counting chamber (Optik Labor) and 200,000 cells were transferred on a 96-well plate. An anti-CD107a-PB antibody and monensin were added to all samples prior to stimulation. IL-2 stimulated cells were co-cultivated with K562 (#89121497) cells at a ratio 1:2 for 3 h at 37°C and 5% $CO_2$.

## Surface phenotyping and flow cytometry
Cells were washed in DPBS plus 0.5% human serum albumin and incubated with anti-CD45-KrO (#B36294; Beckman Coulter), anti-CD3-APC (#IM2467; Beckman Coulter), anti-CD8-APC-AF700 (#B49181; Beckman Coulter), anti-CD69-PE (#IM1943U; Beckman Coulter) and CD56-PC7 (#A21692; Beckman Coulter) for 15 min at RT. Cells were washed once, diluted in 300 µl DPBS/HSA and measured using a ten-color flow cytometer (Navios; Beckman Coulter).

FACS-data were analyzed using Kaluza version 2.1 (Beckman Coulter). The gating strategy is described in Fig. S4. For NK cells, CD107a$^+$ gates were set after a healthy and unstimulated control for each experiment, given the population as percentage of total NK cells. For CD8$^+$ T cells, the mean fluorescence intensity (MFI) was determined as an indicator of CD107a expression.

## Clustal Omega alignments
Amino acid sequence alignments of Arf1, Arf3, Arf4, and Arf5, the alignments of *S. cerevisiae*, *C. elegans*, *Drosophila melanogaster*, mouse, and human Arf1, furthermore the alignments of the *C. elegans*, mouse and human LRBA, and human neurobeachin were generated using Clustal Omega (Madeira et al., 2024).

## Surface biotinylation
800,000 LRBA-deficient and healthy donor fibroblasts were seeded onto 60 mm Petri-dishes and incubated overnight. The next day the culturing media was removed and cells were washed 5× with ice-cold PBS at 4°C and were biotinylated with 1 mg/ml EZ Link Sulfo-NHS-SS-Biotin (#21331; Thermo Fisher Scientific) for 30 min on ice. After incubation, excess biotin was quenched with 20 mM glycine in PBS for 3 × 7 min and rinsed once with PBS. Cells were then lysed using 150 µl lysis buffer (1% Triton X-100, 20 mM Tris/HCl pH 7.5, 150 mM NaCl, 1 mM EDTA, 1 mM EGTA supplemented freshly with protease inhibitor [#11873580001; Roche], Halt phosphatase inhibitor [#78420; Thermo Fisher Scientific]). Cell lysates were collected in pre-chilled low-binding Eppendorf tubes, incubated for 10 min on ice and cleared by centrifuging at 13,000 *g*, 10 min, 4°C. The protein concentration of the cleared lysates was measured using the BCA assay (Pierce BCA Protein Assay kit, #123227; Thermo

Fisher Scientific) and adjusted to equal amounts of proteins. Lysates were filled up to 1 ml with lysis buffer (supplemented with inhibitors) and pull-down of biotinylated proteins was performed using equilibrated Streptavidin C1 DynaBeads (Dynabeads MyOne Streptavidin C1, #65001; Invitrogen by Life Technologies) for 1.5 h at 4°C on an end-to-end rotator. After incubation, beads were washed 2× with lysis buffer and 3× with washing buffer (20 mM Tris/HCl pH7.5, 150 mM NaCl, 1 mM EDTA, 1 mM EGTA) using a magnetic stand to collect beads after each washing at 4°C. With the last wash, beads were transferred to a new tube, then all traces of the wash buffer were removed prior to subsequent MS processing.

## Cell surface biotinylation proteomics
Affinity-purified proteins were eluted from resin by incubation in lysis buffer (5% SDS, 10 mM TCEP (Tris(2-carboxyethyl) phosphine), and 0.1 M TEAB (Triethyloammonium bicarbonate)) for 10 min at 95°C shaking at 500 rpm. The resin was removed using a magnetic stand, and proteins were alkylated in 20 mM iodoacetamide for 30 min at 25°C and digested using S-Trap micro spin columns (Protifi) according to the manufacturer's instructions. Shortly, 12% phosphoric acid was added to each sample (final concentration of phosphoric acid 1.2%) followed by the addition of S-trap buffer (90% methanol, 100 mM TEAB pH 7.1) at a ratio of 6:1. Samples were mixed by vortexing and loaded onto S-trap columns by centrifugation at 4,000 *g* for 1 min followed by three washes with S-trap buffer. Digestion buffer (50 mM TEAB pH 8.0) containing sequencing-grade modified trypsin (Promega) was added to the S-trap column, and samples were incubated for 1 h at 47°C. Peptides were eluted by the consecutive addition and collection by centrifugation at 4,000 *g* for 1 min of 40 µl digestion buffer, 40 µl of 0.2% formic acid, and finally 35 µl 50% acetonitrile and 0.2% formic acid. Samples were dried under vacuum and stored at –20°C until further use.

Dried peptides were resuspended in 0.1% aqueous formic acid, loaded onto Evotip Pure tips (Evosep Biosystems), and subjected to LC–MS/MS analysis using a Q Exactive HF Mass Spectrometer (Thermo Fisher Scientific) fitted with an Evosep One (EV 1000; Evosep Biosystems). Peptides were resolved using a Performance Column—30 SPD (150 µm × 15 cm, 1.5 µm, EV1137; Evosep Biosystems) kept at 40°C fitted with a stainless steel emitter (30 µm, EV1086; Evosep Biosystems) using the 30 SPD method. Buffer A was 0.1% formic acid in water and buffer B was acetonitrile, 0.1% formic acid.

The mass spectrometer was operated in DDA mode with a total cycle time of ~1 s. Each MS1 scan was followed by high-collision-dissociation (HCD) of the 20 most abundant precursor ions with dynamic exclusion set to 30 s. MS1 scans were acquired at a resolution of 120,000 FWHM (at 200 m/z), a scan range set to 350–1,600 m/z with an AGC target of 3e6, and a maximum injection time of 100 ms. MS2 scans were acquired at a resolution of 15,000 FWHM (at 200 m/z), fixed first mass of 100 m/z, with an AGC target of 1e5 and a maximum injection time of 50 ms. Singly charged ions and ions with unassigned charge state were excluded from triggering MS2 events. The normalized collision energy was set to 28%, the mass isolation

window was set to 1.4 m/z, and one microscan was acquired for each spectrum.

## Data analysis

The acquired raw-files were searched using MSFragger (v. 4.0) implemented in FragPipe (v. 21.1) against a *Homo sapiens* database (consisting of 20360 protein sequences downloaded from Uniprot on 20220222) and 392 commonly observed contaminants using the "LFQ-MBR" workflow with minor modifications: in the MSFragger tab, Spectral Processing "Require precursor" was unchecked, in the Validation tab MSBooster was disabled and "Generate MSstats files" was enabled, in the Quant (MS1) tab "MBR top runs" was set to 100,000, "Top N ions" was set to 3, and "Min freq" was set to 0.5. The quantitative data export from FragPipe ("MSstats.csv") was analyzed using the MSstats R package v.4.9.9. (https://doi.org/10.1093/bioinformatics/btu305, Choi et al., 2014). Data was not normalized, imputed using "AFT model-based imputation," and P values for pairwise comparisons were calculated as implemented in MSstats.

Volcano plots were constructed in GraphPad Prism 10. Thresholds of $\log_2$ fold change of ±0.26 (1.2-fold enrichment or depletion) and $-\log_{10}$ P value of 1.3 (corresponding to the P value of 0.05) were set.

Proteomic data has been deposited to the ProteomeXchange Consortium (https://www.proteomexchange.org/) via the MassIVE partner repository with MassIVE data set identifier MSV000095230 and ProteomeXchange identifier PXD053605.

## Secretome analysis

800,000 LRBA-deficient and healthy donor fibroblasts were seeded onto 60-mm Petri dishes and incubated overnight. The next day cell culture media was removed and cells were washed 1× with preheated PBS and 4× with preheated Opti-MEM (#51985034; Gibco). Cells were incubated in 5 ml Opti-MEM (Gibco) for 16 h. After incubation, 1.5 ml of supernatant was collected.

Cell supernatants were TCA precipitated according to a protocol originally from Luis Sanchez (https://www.its.caltech.edu/~bjorker/TCA_ppt_protocol.pdf) as follows: 1 vol of TCA was added to every 4 vol of sample, mixed by vortexing, and incubated for 10 min at 4°C followed by collection of precipitate by centrifugation for 5 min at 23,000 *g*. The supernatant was discarded, pellets were washed twice with acetone precooled to −20°C, and the washed pellets were incubated in the open tube for 2 min at RT to allow residual acetone to evaporate. Pellets were resuspended in 5% SDS, 10 mM TCEP, and 0.1 M TEAB by sonication followed by incubation at 95°C shaking at 500 rpm for 10 min. Resuspended proteins were alkylated in 20 mM iodoacetamide for 30 min at 25°C and digested using S-Trap micro spin columns (Protifi) according to the manufacturer's instructions. Shortly, 12% phosphoric acid was added to each sample (final concentration of phosphoric acid 1.2%) followed by the addition of S-trap buffer (90% methanol, 100 mM TEAB pH 7.1) at a ratio of 6:1. Samples were mixed by vortexing and loaded onto S-trap columns by centrifugation at 4,000 *g* for 1 min followed by three washes with S-trap buffer. Digestion buffer

(50 mM TEAB pH 8.0) containing sequencing-grade modified trypsin (Promega) was added to the S-trap column and samples were incubated for 1 h at 47°C. Peptides were eluted by the consecutive addition and collection by centrifugation at 4,000 *g* for 1 min of 40 µl digestion buffer, 40 µl of 0.2% formic acid, and finally 35 µl 50% acetonitrile, 0.2% formic acid. Samples were dried under vacuum and stored at −20°C until further use.

Dried peptides were resuspended in 0.1% aqueous formic acid and 0.02% DDM (n-Dodecyl-B-D-maltoside) and subjected to LC–MS/MS analysis using a timsTOF Ultra Mass Spectrometer (Bruker) equipped with a CaptiveSpray nanoelectrospray ion source (Bruker) and fitted with a Vanquish Neo (Thermo Fisher Scientific). Peptides were resolved using a RP-HPLC column (100 µm × 30 cm) packed in-house with C18 resin (ReproSil Saphir 100 C18, 1.5 µm resin; Dr. Maisch GmbH) at a flow rate of 0.4 µl/min and column heater set to 60°C. The following gradient was used for peptide separation: from 5% B to 14% B over 6 min to 35% B over 22 min to 40% B over 2 min to 95% B over 1 min followed by 5 min at 95% B to 5% B over 1 min followed by 3 min at 5% B. Buffer A was 0.1% formic acid in water and buffer B was 80% acetonitrile and 0.1% formic acid in water.

The mass spectrometer was operated in dia-PASEF mode with a cycle time estimate of 0.95 s. MS1 and MS2 scans were acquired over a mass range from 100 to 1,700 m/z. A method with 8 dia-PASEF scans separated into three ion mobility windows per scan covering a 400–1,000 m/z range with 25 Da windows and an ion mobility range from 0.64 to 1.37 Vs cm² was used. Accumulation and ramp time were set to 100 ms, capillary voltage was set to 1,600 V, dry gas was set to 3 liters/min, and dry temperature was set to 200°C. The collision energy was ramped linearly as a function of ion mobility from 59 eV at 1/K0 = 1.6 V s cm$^{-2}$ to 20 eV at 1/K0 = 0.6 V s cm$^{-2}$.

## Data analysis

The acquired files were searched using the Spectronaut (v18.6; Biognosys) directDIA workflow against a *Homo sapiens* database (consisting of 20360 protein sequences downloaded from Uniprot on 20220222) and 392 commonly observed contaminants. Quantitative fragment ion data (F.Area) was exported from Spectronaut and analyzed using the MSstats R package v.4.9.9. (https://doi.org/10.1093/bioinformatics/btu305, Choi et al., 2014). Data was not normalized, imputed using "AFT model-based imputation," and P values for pairwise comparisons were calculated as implemented in MSstats.

Volcano plots were constructed in GraphPad Prism 10. Thresholds of $\log_2$ fold change of ±0.26 (1.2-fold enrichment or depletion) and $-\log_{10}$ P value of 1.3 (corresponding to 0.05 P value) were set.

## Gene ontology analysis

Gene ontology analysis was performed using ShinyGO 0.80 (Ge et al., 2020) to represent the cellular component and biological process GO term analysis and KEGG categories (Luo and Brouwer, 2013; Kanehisa et al., 2021). FDR cutoff was set to 0.05 and the background gene set was customized to all genes which were detected by our proteomics experiment.

## Coimmunoprecipitation

800,000 HEK293A cells were seeded onto 6-well plates 24 h prior to transfection. Cells were transfected with 2 µg EGFP, Arf1-EGFP, Arf1$^{Q71L}$-EGFP (constitutively active), Arf1$^{I46S\ I49S}$-EGFP, and Arf1$^{T31N}$-EGFP plasmids using Helix IN reagent (#HX11000; OZ Biosciences) following the manufacturer's instructions. 24 h after transfection, cells were rinsed gently with ice-cold PBS and 150 µl lysis buffer (20 mM TrisHCl pH 7.5, 150 mM NaCl, 5 mM $MgCl_2$, 0.5% CHAPS, 1 mM $CaCl_2$, 1× cOmplete EDTA-free cocktail protease inhibitor (#11873580001; Roche), Halt phosphatase inhibitor (#78420; Thermo Fisher Scientific), and 1 mM phenylmethylsulfonyl fluoride (PMSF) was added into each well. Lysis buffer was supplemented with 100 nM GTP for EGFP, Arf1-EGFP, and Arf1$^{I46S\ I49S}$-EGFP lysates with 100 nM GTPγS for the Arf1$^{Q71L}$-EGFP lysate and with 100 nM GDPβS for the Arf1$^{T31N}$-EGFP lysate. Cells were scraped down and collected in prechilled tubes. Then lysates were incubated for 1 h at 4°C on an end-to-end rotator and lysates were cleared of insoluble material by centrifugation at 4°C, 13,000 rpm for 10 min. Clear lysates were collected in a new tube. GFP-trap magnetic beads (Chromotek GFP-Trap Magnetic Agarose #gtma-20) were blocked with 1% BSA (#A7030; Sigma-Aldrich) in PBS for 1 h, then equilibrated with the dilution buffer (20 mM Tris HCl pH 7.5, 150 mM NaCl, 5 mM $MgCl_2$, 1 mM $CaCl_2$, 1× cOmplete EDTA-free cocktail (#11873580001; Roche), Halt phosphatase inhibitor (#78420; Thermo Fisher Scientific), and 1 mM PMSF. Lysates were diluted with 225 µl dilution buffer supplemented with the corresponding nucleotides and incubated with the beads in low retention tubes at 4°C on an end-to-end rotator for 2 h. After incubation, beads were collected with a magnetic stand and washed 3× with 500 µl washing buffer (20 mM Tris HCl pH 7.5, 150 mM NaCl, 5 mM $MgCl_2$, 0.05% CHAPS, 1 mM $CaCl_2$). At the first and the last washing step, beads were transferred to new Eppendorf tubes. Proteins were eluted using 50 µl 2× SDS-sample buffer at 65°C for 10 min. Samples were resolved by 10% SDS-PAGE and transferred onto nitrocellulose membrane (Amersham) using wet transfer for 3 h. Membranes were blocked with 5% non-fat dry milk in TBST for 30 min and incubated with primary antibodies anti-GFP (#TP401, polyclonal, rabbit, 1:5,000; Torrey Pines) dissolved in 5% milk and with anti-LRBA (#HPA023567, 1:2,000, polyclonal, rabbit; Atlas Antibodies) dissolved in Can get signal immunoreaction enhancer solution (#NKB-101; Toyobo) overnight at 4°C. The next day, membranes were washed for 20 min in TBST and incubated for 2 h with HRP-conjugated secondary antibody (#31460, 1:10,000; anti-rabbit; Invitrogen) in TBST. Chemiluminescence signals were detected using Western Bright ECL reagent (#K-12045-D50; Advansta) for EGFP and Super SignalTM West Femto Maximum Sensitivity Substrate (#34095; Thermo Fisher Scientific) for LRBA and imaged using a FusionFX (Vilber Lourmat). LRBA levels were measured with FIJI, normalized to GFP levels in the pull-down fraction, and plotted using GraphPad Prism 10.

## Statistics

Statistical analysis was performed using GraphPad Prism 10.1.1 (GraphPad Software, Inc.). For statistical evaluation, the normality of the data was routinely tested using the Shapiro–Wilk or Kolmogorov–Smirnov normality tests. For the comparison of two groups, a t test was employed, whereas to compare three or more groups, a one-way ANOVA was performed followed by a posthoc multiple comparison test.

## Online supplemental material

Fig. S1 shows that LRBA does not regulate M6PR trafficking and Golgi assembly. Fig. S2 characterizes enlarged (endo)lysosomal structures in LRBA-deficient patient-derived fibroblasts. Fig. S3 characterizes EGFR signaling in LRBA-deficient patient-derived fibroblasts. Fig. S4 characterizes degranulation of CD8$^+$ T cells and NK cells derived from LRBA-deficient patients. Fig. S5 shows that LRBA is endogenously expressed in HeLa cells and recruited to endosomes by Arfs.

## Data availability

All data are included in the manuscript or supplements and are available from the corresponding authors upon reasonable request. Proteomic data have been deposited to the ProteomeXchange Consortium (https://www.proteomexchange.org/) via the MassIVE partner repository (https://massive.ucsd.edu/) with MassIVE data set identifier MSV000095230 and ProteomeXchange identifier PXD053605.

## Acknowledgments

We are grateful to the patients, their families, and healthy donors. We thank the Stem Cell Transplantation laboratory of the Childrens Hospital Frankfurt (Goethe University), especially S. Erben and Dr. E. Jacobsen (University Hospital Ulm), for suggestions regarding CTL degranulation analysis, and the pediatric surgery for a skin biopsy. The HeLa Trex FlpIn-3xFLAG-EGFP-LRBA cell line was kindly provided by Dr. Serhiy Pankiv and Prof. Anne G. Simonsen (University of Oslo, Oslo, Norway). We acknowledge the support of the FACS facility of the Biozentrum, Cinzia Tiberi Schmidt from the BioEM facility, and IMCF facility of the Biozentrum, especially Laurent Guérard and Alexia Loynton-Ferrand. We thank Dominik Buser for helpful discussions and antibodies. Calculations were performed at sciCORE (http://scicore.unibas.ch/) scientific computing center at the University of Basel. We would like to thank Carlos Omar Oueslati Morales and Ludovic Enkler for the critical reading of the manuscript. We are grateful to all members of the Spang group for insightful discussions and Marc Thommen and Laura Vrijbloed for their technical support. Figures were created with BioRender.

This study was supported by the Hilfe für krebskranke Kinder e.V., Frankfurt, Germany to L.M. Lueck, the Swiss National Science Foundation (310030_197779), and the University of Basel to A. Spang, and the clinician-scientist program Goethe University Frankfurt, Germany as well as the Dr. Rolf Schwiete Foundation to S. Bakhtiar. Open Access funding provided by Goethe-Universität Frankfurt am Main.

Author contributions: V. Szentgyörgyi: Conceptualization, Data curation, Formal analysis, Investigation, Methodology, Project administration, Validation, Visualization, Writing—original draft, Writing—review & editing, L.M. Lueck: Data curation, Formal analysis, Investigation, Methodology, Resources,

Validation, Visualization, Writing—review & editing, D. Overwijn: Formal analysis, Software, Visualization, Writing—review & editing, D. Ritz: Formal analysis, Resources, N. Zoeller: Resources, A. Schmidt: Methodology, Project administration, Resources, Supervision, M. Hondele: Resources, Supervision, Validation, Writing—review & editing, A. Spang: Conceptualization, Funding acquisition, Project administration, Supervision, Validation, Visualization, Writing—original draft, Writing—review & editing, S. Bakhtiar: Conceptualization, Funding acquisition, Investigation, Methodology, Supervision, Validation, Writing—original draft, Writing—review & editing.

Disclosures: The authors declare no competing interests exist.

Submitted: 3 February 2024

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

# Supplemental material

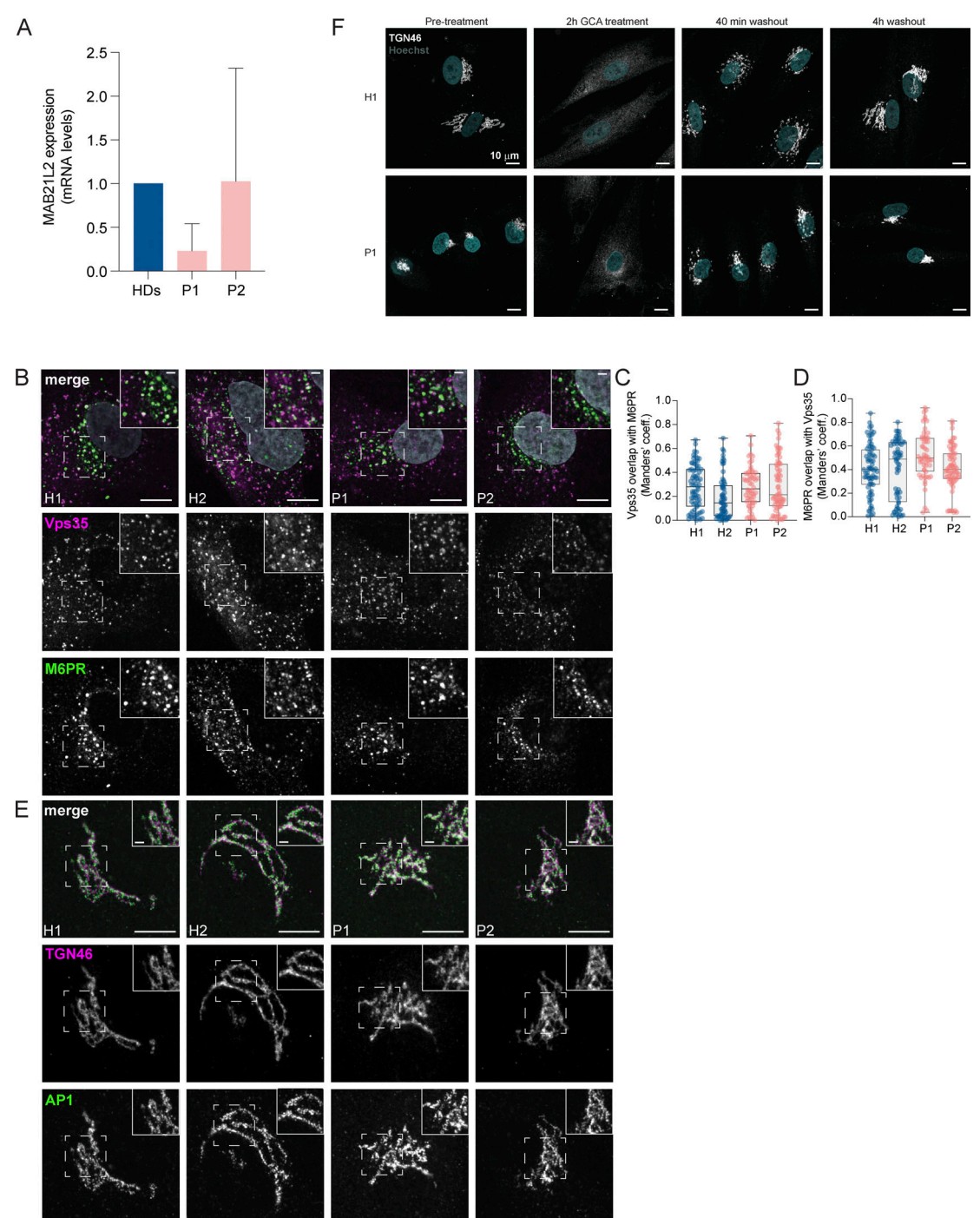

Figure S1. **LRBA does not regulate Golgi assembly. (A)** The nested MAB21L2 gene expression levels in LRBA deficient fibroblasts. MAB21L2 mRNA levels in two patient-derived and three HD fibroblast lines were determined by qRT-PCR. Mean and standard deviation are shown from $n = 3$ biological replicates. HDs versus P1 (P = 0.4805) and HDs versus P2 (P = 0.9990); one-way ANOVA using Tukey's multiple comparison. **(B)** Colocalization analysis of Vps35 and M6PR in two HDs and two patient-derived fibroblast cell lines. Representative confocal immunofluorescence images of single focal planes. Squares show the magnified area. Inlays are shown in the top right corner of the images. The labeling of the single channels represents the color of the channel on the merged image. Scale bar, 10 μm, inlays 2 μm. **(C and D)** Colocalization between Vps35 and M6PR was measured using Mander's colocalization index. Mean and minimum to maximum are shown, box ranges from the first (Q1–25th percentiles) to the third quartile (Q3–75th percentiles) of the distribution; H1 = 73 cells, H2 = 63 cells, P1 = 56 cells, P2 = 68 cells from $n = 3$ biological replicates. **(C)** Mander's coefficient of Vps35 overlap with M6PR. **(D)** Mander's coefficient of M6PR overlap with Vps35. **(E)** Immunofluorescence analysis of TGN46 and AP1 colocalization in two HDs and two patient-derived fibroblast lines. Representative confocal images of single focal planes. Squares show the magnified area. Inlays are shown in the top right corner of the images. The labeling of the single channels represents the color of the channel on the merged image. Scale bar, 10 μm, inlays 2 μm. **(F)** Golgi reassembly is not regulated by LRBA. H1 and P1 cells were grown on coverslips and treated with GCA for 2 h to vesiculate the Golgi. After 2 h, GCA was washed out and the cells were incubated with complete growth media for indicated time points for Golgi reassembly. Cells were then fixed and their Golgi was visualized by endogenous TGN46 staining and their nuclei by Hoechst staining. Representative maximum Z-projected images are shown.

**Szentgyörgyi et al.**

Arf1/3-dependent LRBA localization on Rab4 endosomes regulates lysosome function and size

**Journal of Cell Biology**  S2

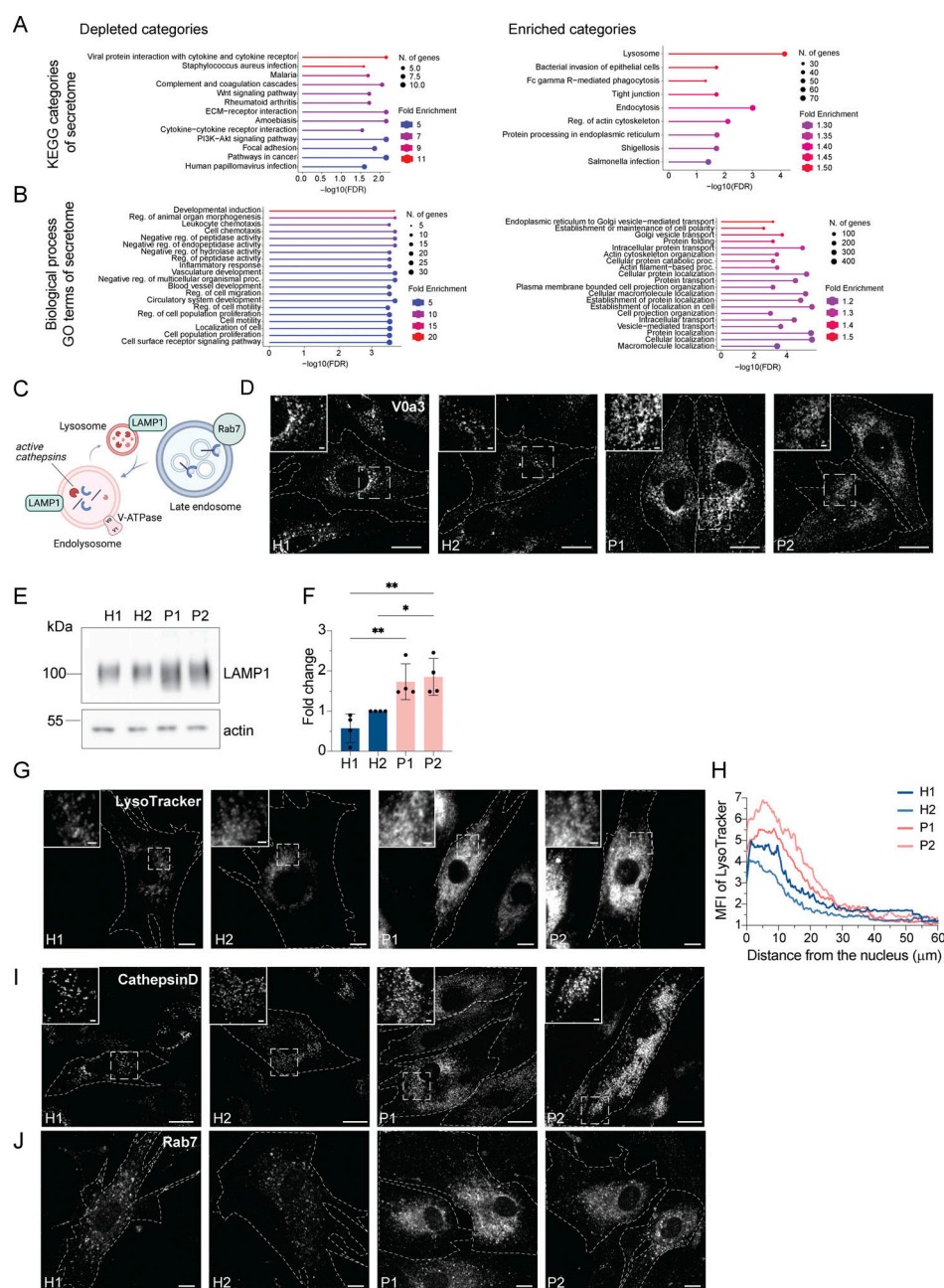

Figure S2. **Enlarged (endo)lysosomal structures are acidified and contain active cathepsin D. (A)** Gene ontology analysis of enriched and depleted KEGG categories in the patients' secretome. **(B)** Gene ontology analysis of enriched and depleted biological processes in the patients' secretome. **(C)** Scheme showing Rab7+ late endosome fusing with LAMP1+ lysosomes and becoming endolysosomes. Endolysosomes are acidified and contain active cathepsins in their lumen and vacuolar-ATPase and LAMP1 in their membrane. **(D)** Visualization of endolysosomes by immunostaining the V0a3 subunit of the lysosomal V-ATPase. Healthy and LRBA-deficient fibroblasts were fixed and stained with V0a3 antibody. Representative confocal images. Cell outlines are marked with a dashed line. Inlays are shown in the top left corner of the images. Scale bar, 10 μm, inlays 2 μm. **(E)** Immunoblot analysis of LAMP1 protein levels in two healthy and two LRBA-deficient patient-derived fibroblast lines. Actin was used as a loading control. **(F)** Quantification of LAMP1 levels based on immunoblots shown on panel E from $n$ = 4 biological replicates; one-way ANOVA using Tukey's multiple comparison; *P = 0.0279, **P = 0.0036 (H1 versus P1), **P = 0.0017 (H1 versus P2). **(G)** Accumulation of acidified endolysosomes in LRBA-deficient fibroblasts. Fibroblasts were seeded onto imaging chambers, stained with LysoTracker Green and imaged live at 37°C and 5% $CO_2$ atmosphere. Maximum Z-projection of wide-field images are shown. Cell outlines are marked with a dashed line. Inlays are shown in the top left corner of the images. Scale bar, 10 μm, inlays 2 μm. **(H)** Quantification of the LysoTracker Green intensity along the nucleus-cell periphery axis. A line ROI was drawn from the edge of the nucleus to the cell periphery and LysoTracker Green intensity was measured. Values were normalized to the maximum of each cell and averaged per experiment. The normalized mean of $n$ = 3 biological replicates is plotted along the axis; H1 = 30 cells, H2 = 30 cells, P1 = 30 cells, P2 = 30 cells were analyzed. **(I)** Cathepsin D is present in accumulating, enlarged (endo)lysosomes. Healthy and LRBA-deficient fibroblasts were fixed and stained with endogenous cathepsin D antibody. Representative confocal images from $n$ = 3 biological replicates. Cell outlines are marked with a dashed line. Inlays are shown in the top left corner of the images. Scale bar, 10 μm, inlays 2 μm. **(J)** Immunofluorescence analysis of Rab7+ late endosomes in two HDs and two patient-derived fibroblast lines. Representative confocal images. Cell outlines are marked with a dashed line. Scale bar, 10 μm. Source data are available for this figure: SourceData FS2.

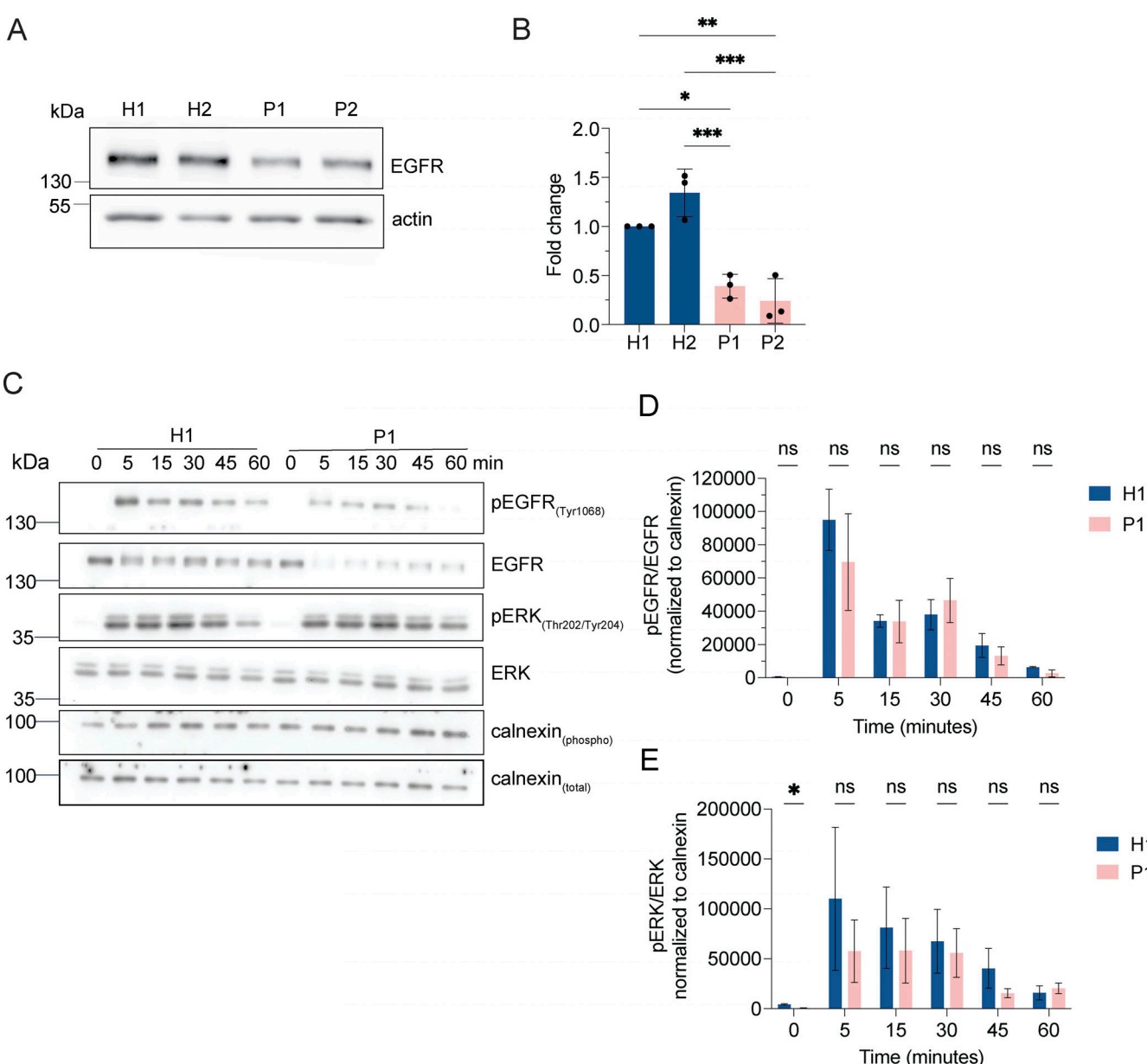

Figure S3. **LRBA does not regulate EGFR signaling attenuation. (A)** EGFR protein levels are reduced in LRBA-deficient fibroblasts. Immunoblot analysis of EGFR in two healthy and two LRBA-deficient patient-derived fibroblast lines. Actin was used as a loading control. The same blot has been re-probed with LRBA antibody and is shown in Fig. 1 C, therefore the loading control (actin) is identical on the two images. **(B)** Quantification of EGFR levels based on immunoblots shown on panel A from $n = 3$ biological replicates; one-way ANOVA using Tukey's multiple comparison; *P = 0.0124, **P = 0.0034, ***P = 0.0008 (H2 versus P1), ***P = 0.0003 (H2 versus P2). **(C)** Immunoblot analysis of EGFR signaling kinetics in H1 and P1 fibroblasts. Cells were serum-starved overnight in DMEM and then incubated with 2 µg/ml EGF in serum-free DMEM for the indicated time points. Cells were then rinsed with ice-cold PBS and lysed with M-PER lysis buffer supplemented with protease and phosphatase inhibitors. For addressing EGFR signaling, an antibody against EGFR and its phosphorylation site Tyr1068 was used. We also detected the downstream ERK phosphorylation with the Thr202/Tyr204 phosphorylation sites specific antibody. Calnexin was used as loading control. **(D)** Quantification of pEGFR levels and kinetics upon EGF stimulation based on immunoblots shown in A. pEGFR and EGFR levels were measured and normalized to calnexin loading controls (pEGFR_norm, EGFR_norm). Then pEGFR_norm values were normalized to EGFR_norm values and plotted over time. Two-way ANOVA using Šidák's multiple comparisons test. **(E)** Quantification of pERK levels and kinetics upon EGF stimulation based on immunoblots as shown in A. pERK and total ERK levels were measured and normalized to calnexin loading controls (pERK_norm, ERK_norm). Then pERK_norm values were normalized to ERK_norm values and plotted over time. Two-way ANOVA using Šidák's multiple comparisons test. *P = 0.0448. Source data are available for this figure: SourceData FS3.

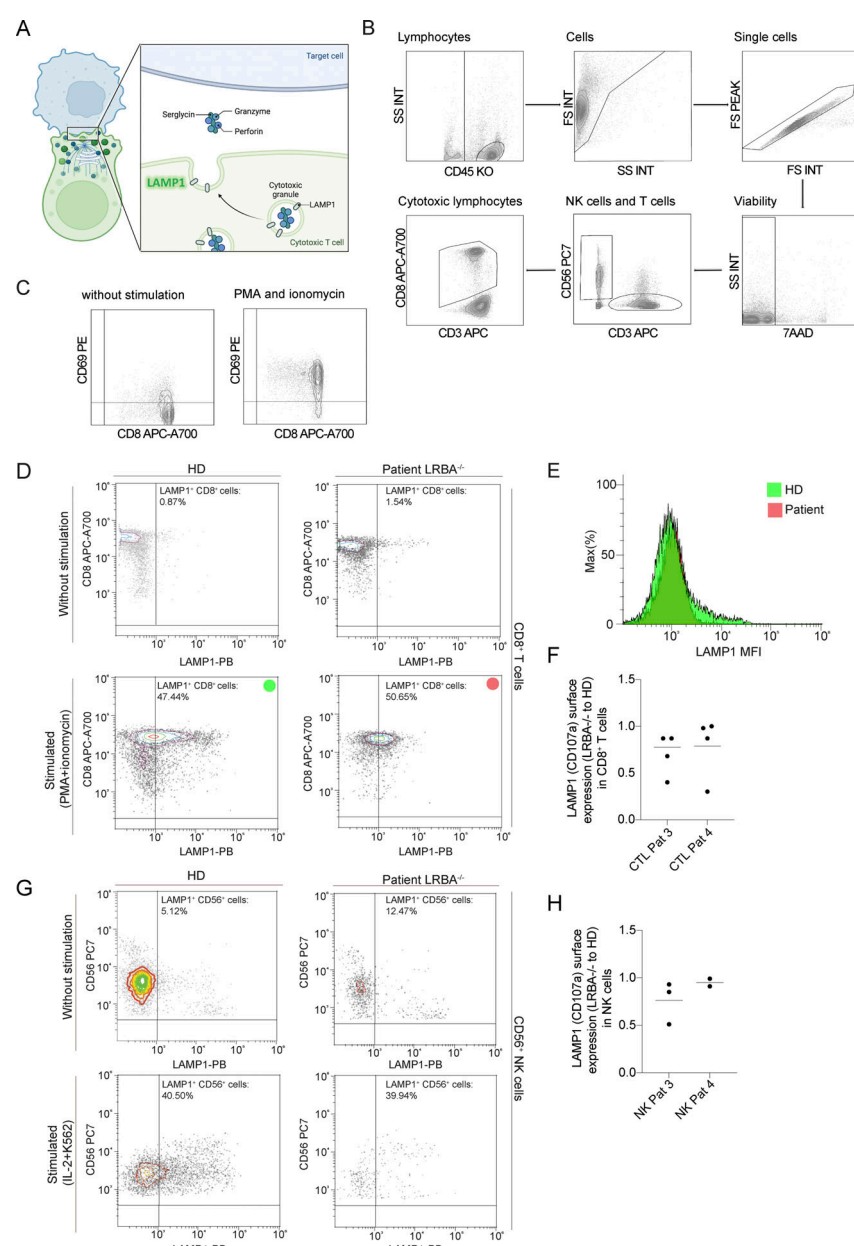

Figure S4. **Degranulation of CD8+ T cells and NK cells is unimpaired in LRBA deficiency. (A)** Scheme of immunological synapse formation and lytic/cytotoxic granule exocytosis in cytotoxic T cells. Upon target cell recognition an immune synapse is formed which induces a strong polarization in T cells. The microtubule-organizing center (MTOC) is trafficked to the immunological synapse bringing other organelles like the Golgi network, endosomes, and lytic granules to the synapse. Lytic granules are lysosome-related organelles containing canonical lysosomal proteins (LAMP1), granzyme and perforin. During degranulation, the lytic granules are fused with the plasma membrane. Upon release, perforin mediates the generation of pores in the plasma membrane of the target cell allowing granzymes to access the cytoplasm and induce apoptosis. The degranulation process exposes LAMP1 on the cell surface and can be used as a marker for degranulating cells. Scheme was created with Biorender. **(B)** Gating strategy of CD8+ T cells and NK cells in the degranulation assay. Gating strategy for cytotoxic lymphocytes and NK cells is shown on the sample of a HD. Leukocytes and lymphocytes are defined in an SSC/CD45 gate. Lymphocyte populations are visualized again in an SSC/FSC gate. Single cells are discriminated from doublets in an FS peak/FSC gate. Viable cells are gated as 7AAD negative cells in a viability gate. NK cells are defined as CD3- and CD56+ cells. Cytotoxic lymphocytes are defined as CD3+ and CD8+. SSC, side scatter, FSC, forward scatter. **(C)** CD69 was used as a common marker for lymphocyte activation upon stimulation. An example of CD69 gating in CD8+ cells from a HD is shown. **(D)** Flowcytometric analysis of LAMP1(CD107a) surface expression in CD8+ T cells of a healthy individual (HD) and an LRBA-deficient patient (LRBA−/−) in mononuclear cells (PMCS) isolated from peripheral blood samples. Upon stimulation with PMA+ ionomycin, there is a significant increase in LAMP1 expression. Dot blots show the CD8+LAMP1+ population marked with a green circle for the healthy and red circle for the affected individual. **(E)** The mean fluorescence intensity (MFI) was calculated for each sample, given as green (HD) and red (patient) histograms. **(F)** Patient-derived CD8+ T cells degranulate almost as efficiently as healthy cells. Degranulation is given by the ratio of LAMP1 surface expression for each LRBA deficient patient and the healthy control in each degranulation assay as these where available at different time points. **(G)** For the NK cells the stimulation included IL-2 and co-culture with K562 cells, resulting in a subpopulation of NK cells (%) expressing LAMP1 in a healthy donor and patient. Dot plots show the percentage of the CD56+LAMP1+ population (top right quadrants). **(H)** Patient-derived NK cells degranulate almost as efficiently as healthy cells. Degranulation is given by the ratio of LAMP1 surface expression for each LRBA-deficient patient and the healthy control in each degranulation assay as these were available at different time points.

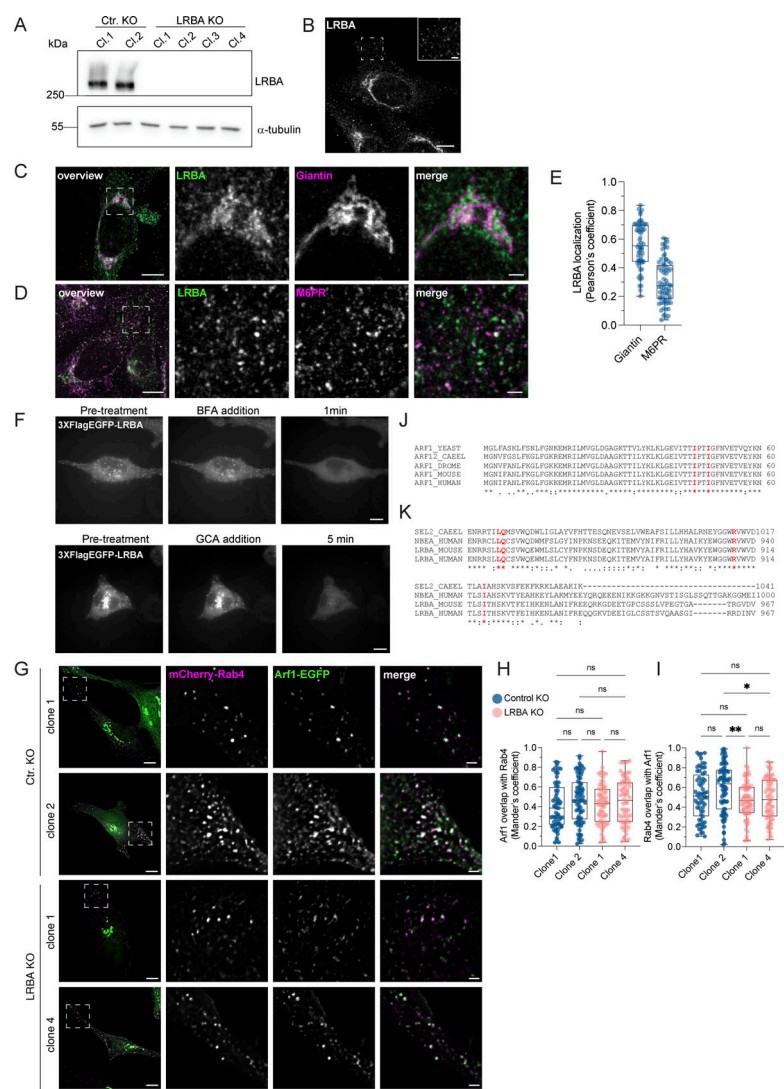

Figure S5. **LRBA is endogenously expressed in HeLa cells and recruited to endosomes by Arfs. (A)** Immunoblot analysis of LRBA presence in HeLa cells of two control and four LRBA KO clones using polyclonal LRBA antibody and α-tubulin as a loading control. **(B)** LRBA is localized at the perinuclear region and on vesicular structures in HeLa cells. Immunofluorescence analysis of endogenous LRBA in fixed HeLa cells. Scale bar, 10 μm, inlay 2 μm. **(C)** LRBA partly co-localizes with the cis-Golgi in HeLa cells. Immunofluorescence staining of endogenous LRBA and the cis-Golgi marker giantin in fixed HeLa cells. Squares show magnification of the perinuclear area. The labeling of the single channels represents the color of the channel on the merged image. Scale bar, 10 μm, inlays 2 μm. **(D)** LRBA does not colocalize with M6PR in HeLa cells. Immunofluorescence analysis of endogenous LRBA and endogenous M6PR colocalization in fixed HeLa cells. Squares show magnification of the perinuclear area. The labeling of the single channels represents the color of the channel on the merged image. Scale bar, 10 μm, inlays 2 μm. **(E)** Colocalization measurement of LRBA with giantin and M6PR. To measure LRBA colocalization with giantin one ROI at the perinuclear region was analyzed. To measure colocalization with M6PR, two ROIs per cell at the cell periphery were analyzed and the Pearson's coefficient was measured using the JACoP plugin in Fiji. Mean and minimum to maximum are shown, box ranges from the first (Q1–25th percentiles) to the third quartile (Q3–75th percentiles) of the distribution. All data points are shown. Giantin = 60 cells, M6PR = 35 cells. **(F)** LRBA puncta disperse upon treatment with ArfGEF inhibitors. Live-cell imaging of 3xFlagEGFP-LRBA upon BFA (top panels) and GCA (lower panels) treatment for indicated timepoints. Scale bar, 10 μm. **(G)** Arf1 is recruited onto Rab4+ endosomes in the absence of LRBA. Control KO and LRBA KO HeLa cells were transfected with mCherry-Rab4 and Arf1-EGFP and cells were imaged live using a wide-field microscope at 37°C, 5% $CO_2$ atmosphere. Deconvolved images of single stacks are shown. Squares show magnification of the perinuclear area. The labeling of the single channels represents the color of the channel on the merged image. Scale bar, 10 μm, inlays 2 μm. **(H and I)** Colocalization measurement of Arf1-EGFP and mCherry-Rab4 in control and LRBA KO HeLa cells. Two ROIs per cell were analyzed and Mander's coefficients were measured using the JACoP plugin in Fiji. Arf1 overlap with Rab4 (M1) is shown in H, Rab4 overlap with Arf1 (M2) is shown in I. All data points are shown. Ctr. KO clone1 = 32 cells, Ctr. KO clone2 = 39 cells, LRBA KO clone1 = 36 cells, LRBA KO clone 4 = 36 cells from $n$ = 3 biological replicates; **(H)** one-way ANOVA using Tukey's multiple comparison. **(I)** Kruskal–Wallis test using Dunn's multiple comparisons test, **P = 0.0071, *P = 0.0201. **(J)** The amino acids isoleucine 46 and 49 of Arf1 and Arf3 were predicted to interact with LRBA. Both amino acids are conserved across species. The amino acid sequences of the yeast, *C. elegans* (CAEEL), *Drosophila melanogaster* (DROME), mouse and human Arf1 and human Arf3 were aligned. Labels: (*) conserved sequence; (:) conservative mutation; (.) semiconservative mutation; (–) gap. Sequence alignments were performed using Clustal Omega. **(K)** The amino acids leucine 861, arginine 910, and isoleucine 918 of LRBA were predicted to interact with Arf1 and Arf3. All three amino acids are conserved across species. The amino acid sequences of the *C. elegans* SEL-2 (SEL2-CAEEL), the mouse and the human LRBA, and the human neurobeachin (NBEA) were aligned. Labels: (*) conserved sequence; (:) conservative mutation; (.) semi-conservative mutation; (-) gap. Sequence alignments were performed using Clustal Omega. Source data are available for this figure: SourceData FS5.

