## [Peer Review File · The Journal of Cell Biology]

Arf1-dependent LRBA recruitment to Rab4 endosomes is required for endolysosome homeostasis

Viktória Szentgyörgyi, Leon Lueck, Daan Overwijn, Danilo Ritz, Nadja Zoeller, Alexander Schmidt, Maria Hondele, Anne Spang, and Shahrzad Bakhtiar

Corresponding Author(s): Shahrzad Bakhtiar, Goethe University Frankfurt

Review Timeline:

Submission Date:	2024-02-03
Editorial Decision:	2024-03-04
Revision Received:	2024-07-15
Editorial Decision:	2024-07-25
Revision Received:	2024-08-05

Monitoring Editor: Jennifer Stow

Scientific Editor: Andrea Marat

Transaction Report:

DOI: <https://doi.org/10.1083/jcb.202401167>

March 4, 2024

Re: JCB manuscript #202401167

Dr. Shahrzad Bakhtiar
Goethe University Frankfurt
Theodor-Stern-Kai 7
Frankfurt 60529
Germany

Dear Dr. Bakhtiar,

Thank you for submitting your manuscript entitled "Endosomal LRBA regulates the endo-lysosomal pathway". The manuscript was assessed by expert reviewers, whose comments are appended to this letter. We invite you to submit a revision if you can address the reviewers' key concerns, as outlined here.

You will see that the reviewers appreciate that your manuscript shows the targeting of LRBA to Rab4 endosomes by Arfs 1 and 3, and novel endocytic enlargement associated with loss of LRBA in patient's cells. Overall, the experiments fall short of pinpointing mechanistic functions for LRBA in endolysosomal traffic or indeed explaining how the enlarged endosomes are linked to the immunodeficiency phenotype. Nevertheless, the implication of an endolysosomal role for LRBA is significant and the careful interrogation of endolysosomal morphology and functions here (although negative for LRBA) provide valuable information for the field in comparing the differential lysosomal phenotypes associated with loss of BEACH domain protein family members and with other lysosomal storage diseases. As reflected consistently by the reviewers' comments, while the manuscript currently provides descriptive evidence to back up its main findings, more quantitative data and further experimental data are needed for verification. Specifically, quantification of imaging is needed for localization experiments in Figures 6 and 7 and for discriminating Golgi and endosomal staining in patient cells and cell lines. Ideally, a rescue of the defects in patient cells by overexpression of WT LRBA should be included, if indeed cells are available; this could also be achieved by rescue of changes in the LRBA KO cell lines described in the manuscript. The LRBA interaction with Arfs1 and 3 predicted by in silico models needs to be verified experimentally by mutation or truncation of LRBA or by site specific mutation of Arfs, analyzed by IP or localization as evidence for interaction, or by use of the Arf binding mutants for ARF KO rescue experiments. It would indeed be of interest to examine CTLA-4 in the Arf1 and 3 mutants, given its association with LRBA, while this may be beyond the scope of the current study, the authors should include some speculation about CTLA-4 in the context of their findings. Finally, the title of the paper does not convey enough information to attract the relevant or interested readers, it could be enhanced by defining the LRBA abbreviation and/or by mention its association with patient immunodeficiency.

We would be happy to discuss a potential revision plan with you to ensure that we are all in agreement about the expectations for a resubmission to JCB. While you are revising your manuscript, please also attend to the following editorial points to help expedite the publication of your manuscript. Please direct any editorial questions to the journal office.

GENERAL GUIDELINES:

Text limits: Character count for an Article is < 40,000, not including spaces. Count includes title page, abstract, introduction, results, discussion, and acknowledgments. Count does not include materials and methods, figure legends, references, tables, or supplemental legends.

Figures: Articles may have up to 10 main text figures. Figures must be prepared according to the policies outlined in our Instructions to Authors, under Data Presentation, <https://jcb.rupress.org/site/misc/ifora.xhtml>. All figures in accepted manuscripts will be screened prior to publication.

IMPORTANT: It is JCB policy that if requested, original data images must be made available. Failure to provide original images upon request will result in unavoidable delays in publication. Please ensure that you have access to all original microscopy and blot data images before submitting your revision.

Supplemental information: There are strict limits on the allowable amount of supplemental data. Articles may have up to 5 supplemental figures. Up to 10 supplemental videos or flash animations are allowed. A summary of all supplemental material should appear at the end of the Materials and methods section.

Please note that JCB now requires authors to submit Source Data used to generate figures containing gels and Western blots with all revised manuscripts. This Source Data consists of fully uncropped and unprocessed images for each gel/blot displayed in the main and supplemental figures. Since your paper includes cropped gel and/or blot images, please be sure to provide one Source Data file for each figure that contains gels and/or blots along with your revised manuscript files. File names for Source

Data figures should be alphanumeric without any spaces or special characters (i.e., SourceDataF#, where F# refers to the associated main figure number or SourceDataFS# for those associated with Supplementary figures). The lanes of the gels/blots should be labeled as they are in the associated figure, the place where cropping was applied should be marked (with a box), and molecular weight/size standards should be labeled wherever possible.

The typical timeframe for revisions is three to four months. While most universities and institutes have reopened labs and allowed researchers to begin working at nearly pre-pandemic levels, we at JCB realize that the lingering effects of the COVID-19 pandemic may still be impacting some aspects of your work, including the acquisition of equipment and reagents. Therefore, if you anticipate any difficulties in meeting this aforementioned revision time limit, please contact us and we can work with you to find an appropriate time frame for resubmission. Please note that papers are generally considered through only one revision cycle, so any revised manuscript will likely be either accepted or rejected.

Thank you for this interesting contribution to Journal of Cell Biology. You can contact us at the journal office with any questions at cellbio@rockefeller.edu.

Sincerely,

Jennifer Stow, PhD
Monitoring Editor

Andrea L. Marat, PhD
Senior Scientific Editor

Journal of Cell Biology

Reviewer #1 (Comments to the Authors (Required)):

This is a manuscript that focuses on understanding the function of LRBA in regulating endolysosomal traffic. Mutations in LRBA is known to lead to severe childhood immune dysregulation, yet how LRBA functions remains to be determined. It was proposed that LRBA mutations results in increased degradation of CTLA-4, presumably due to issues in endolysosomal traffic and/or maturation. However, how LRBA regulates CTLA-4 mechanistically is still unclear. Since this manuscript attempts to answer this question, it is potentially very interesting and has high significance. Unfortunately the manuscript comes way short of this goal. While it has some interesting findings, such as involvement of Arf1 and Arf6 in LRBA localization to Rab4 endosomes, the manuscript for most part is quite descriptive and correlative. Additional experiments (some pretty straight forward and easy to do) needs to be included to strengthen the study. For example, (a) mutational analysis of LRBA need to be performed to test in silico findings, (b) is CTLA-4 traffic affected in Arf1 and Arf3 KO cells?, (c) does LRBA affect Arf1 and Arf6 function? These are just few questions that I have after reading this manuscript. I obviously do not expect authors to address all of them, but some mechanistic insights would dramatically strength the manuscript. Additionally, it is still not clear to me how LRBA localization on Rab4-endosomes affect lysosomal trafficking. Authors looked at several aspects of endolysosomal function and none of them seem to be affected (except enlargement). That rises the questions about how LRBA really function. Does it affect specifically endolysosomal traffic of CTLA-4? Many of these questions need to answered before manuscript is ready for publication. Some more specific comments are listed below.

- 1) Figure 6. Colocalization (or lack of) between LRBA and various endocytic markers (shown in mages) need to be quantified.
- 2) Figure 7B. Colocalization of LRBA and Rab4 in wt and KO cells need to shown and quantified.
- 3) Figure 8. The sites predicted to mediate Arf1 and Arf3 binding in silico, need to be tested in cells (IPs and localization).

Reviewer #2 (Comments to the Authors (Required)):

In this manuscript, Szentgyorgyi et al. address the cellular function of the Lipopolysaccharide Responsive Beige-like Anchor protein (LRBA). Patients have been identified with mutations at amino acids 816 and 2288 in LRBA, both of which lead to severe childhood immune dysfunction, which has been linked to impaired trafficking of the immune checkpoint receptor, CTLA-4, via Rab11. The authors do a series of experiments on LRBA, which may provide some new insight on its function. They maintain (in the abstract) that they do not find colocalization with Rab11, but instead observe colocalization with Rab4-containing endosomes. They provide data that suggests that patient cells with mutant LRBA have essentially no LRBA expression, and this in turn leads to morphological changes at the TGN. However, the altered morphology has little or no effect on Golgi function, including M6PR trafficking and biosynthetic transport (although secretion was not tested). At the same time, the authors observed that endolysosomes are enlarged and clustered in the perinuclear region of the cell. However, these enlarged endolysosomes are also functional, and capable of EGFR degradation. Moreover, the degranulation of lytic granules (which are lysosome-related organelles), is similarly unaffected in patients, which apparently differentiates between LRBA and LYST, a large BEACH domain protein that causes Chediak-Higashi disease. Additional studies suggest that Arf1 and Arf3 together recruit LRBA to Rab4-containing endosomes (but are not essential for Golgi recruitment, where the majority of LRBA is localized). Finally, the authors show AlphaFold models and propose an interaction between LRBA and Arf1/3-but this is not tested experimentally.

This is a study that uses fibroblasts derived from patients with mutations in LRBA to address the biological function of this protein. While there is some limited new insight into the complex function of this very large protein, and the study primarily illustrates various morphological phenomena, and it provides little new mechanistic information. A few of the key weaknesses of this study include the limited "positive data" for the function of LRBA, the failure to mechanistically connect between Arf1/3 and LRBA, limited insight as to how mutant/non-expressed LRBA leads to enlarged endolysosome, and how these enlarged structures impact the immune defects in patients.

Specific comments:

- 1) The authors show loss of mRNA and protein expression for the two patient cells. Loss of protein expression might be explained by degradation of a misfolded mutant protein. Is this indeed the case? If so, what is the explanation for decreased mRNA levels? It seems that these are important questions that are not addressed. On the other hand, there is a lot of data showing morphological change that has little effect on function.
- 2) Golgi morphology is often affected by a variety of conditions and genetic manipulations. Given that there is no detected change in Golgi function, these data are not highly consequential, and should be included as supplemental rather than primary data.
- 3) Golgi secretion was not tested.
- 4) The authors contend that Rab4 colocalizes with LRBA, whereas Rab11 does not. This reviewer does not agree with this interpretation. First, there is no quantification of the data shown in Fig. 6. Second, from the single set of images displayed for Rab4 and Rab11, there is little discernible difference in the overlap with LRBA. It is entirely unclear on what basis this interpretation is made. Evidence for select recruitment to Rab4 endosomes is not compelling, and this is one of the main conclusions drawn from the study.
- 5) Fig. 6 entirely lacks quantification, but also uses over-expression of the endosomal markers, which can lead to misinterpretations due to high levels of expression. Why are endogenous endosomal markers not used? There are excellent antibodies available as markers of endosomes.
- 6) There is no proposed rationale for why, when the vast majority of LRBA is found at the Golgi, that there are such significant effects at endosomes.
- 7) There are no essential controls demonstrating the knockout of Arf1, Arf3 and the double knockout. There is a reference to a previous paper, but cells can evolve and one needs to see the knockout in this manuscript (Fig. 7).
- 8) There are no "rescue" experiments to demonstrate that the effects visualized are not due to off-target effects or clonal selection of cell lines.
- 9) If Arf1/3 KO is insufficient to impair LRBA localization to the Golgi and the authors propose that other Arfs may be responsible/compensating, then this should be tested.
- 10) The authors show AlphaFold models suggesting that Arfs might interact with LRBA. This is entirely speculative, and yet determining if there is binding, how, and whether the binding is required is one of the mechanistic connections that would

contribute to current understand of LRBA function. Why do the authors not test binding?

11) The authors note that "MAB21L2 transcripts are still present in both patients, although with somewhat reduced levels as compared to controls". However, this does not appear to be an accurate depiction: Sup. Fig. 1 shows one patient with much lower levels than control, while the other is essentially the same, but with a huge standard deviation. These data likely need additional repeats to verify if indeed there are statistical differences before making conclusions.

12) Data from Fig. S2 need to be quantified-including the immunoblots and fluorescence images.

13) There are no statistical tests applied in Fig. S3 to address whether the differences observed are significant (B and C).

14) In Fig. S5, the authors claim that in HeLa cells, LRBA juxtaposes with the Golgi but does not colocalize with M6PR. There is no quantification. This reviewer can identify, even in the area chosen by the authors for an insert, several instances of what appears to be colocalization between M6PR and LRBA. Conclusions must be based on quantified data.

15) In Fig. S5F-the authors show homologous residues across species in Arf1/3 predicted to interact with LRBA, but do not show the interaction.

Reviewer #3 (Comments to the Authors (Required)):

Szentygöryi et al. have used a range of cell biology techniques to characterise patient mutations in LRBA, a Golgi and endosome localised protein. They carefully characterise the phenotype of the patient cells and the effects of LOF LRBA mutants on the sub-cellular organisation. They see clear and striking phenotypes of enlarged endosomes, and they usefully and sensibly include negative phenotypes such as on EGF uptake. The endosomal phenotype is curious as LRBA appears mostly localised to the Golgi; however, the authors identify a population at the endosome. The endosomal population is demonstrated to be recruited by Arf1 and 3, and an alphafold model of the binding interface is presented. Overall, the manuscript presents a set of very well-performed experiments, appropriately interpreted with proper controls in most cases. The data is important, and understanding the molecular cell biology of diseases such as LRBA mutations is valuable and allows for a better understanding of disease coupled with the ability to make novel insights into fundamental biological mechanisms. There are a number of controls and additional experiments I detail below that I feel will strengthen the manuscript and some of its claims.

In figures 1 and 2 the authors use TGN46 to measure the TGN shape and volume. They refer to this as the TGN, however, this can be high risk in situations where membrane trafficking is defective as the trafficking of TGN46 could result in a different localisation. In Figure S1B, the authors do address this but performing a GCA treatment and subsequent washout, however I have several issues with this experiment, considering the strength of evidence needed. Firstly, the time course of 4 hours is enough to return to steady state- as the authors show already. Secondly, I see a difference at the 40-minute time point. Thirdly, none of this data is quantified. I also note that the LAMP1 glycosylation was affected, and as the authors point out in the discussion, this could be interpreted as a defect in Golgi function. I finally note that the TGN looked unaffected in EM, although it is hard to tell from the presented data. Ultimately to prove Golgi function is unaffected a lot more evidence is required.

Although there are many advantages to working with patient cells, one of the problems is that they are not isogenic with the controls. Accordingly, one additional control I would expect is to have overexpression of the WT protein cDNA to demonstrate that the phenotypes recover. It may be a bit much to ask for the authors to repeat every experiment with an additional control, however I would expect this for at least a few key experiments and observations.

The data that Arf1 and LRBA colocalise at the endosome is convincing (although not quantified); I do not find the localisation at the Golgi convincing. The authors claim a colocalisation there, but I see two different localisations.

LRBA has a clear endosomal population in HeLa cells; as I said, at least a pool of this appears to coincide with Arf1. I note that 1) arf3 is not assessed. More importantly, 2) no endosomal population was visible in the healthy donor cells in 1D. The authors do discuss this in the discussion, suggesting that the endosomes are more clustered in the healthy donor cells, however this is an important issue as it really points to the key phenotypes in the cells. I think localising other arf1/3 endosomal markers and showing them in endosomes that are proximal to the Golgi would be an important way to conclude this point.

Generally, the quantification and biological repeats are of the highest quality, unusually so, and I commend the authors for their rigour. Saying that there are a number of important quantifications missing. Co-localisations in figure 6 and 7. If the authors find colocalisation quantification a problem, at least the number of experimental repeats should be presented in the legend so that an experimental mistake for a key observation can be ruled out.

The arf1/3 KO is really a key result in this paper. However, no evidence of a KO at the RNA or protein level is presented. In addition, to rule out off-target effects of the RNA, I would expect a recovery by over-expression of the arf1 and independently

arf3. If the author's hypothesis is correct, either should recover.

I find the evidence presented in Figure 8 compelling but weak. A PAE plot is absolutely essential when presenting AF data, without that the confidence in the binding model is not assessable. I think that in an ideal world, binding data by pulldown or ITC would be a large improvement (with, of course, the mutant as well). The authors could use the alpha fold to design a fragment of LRBA if it is hard to purify. A KO of LRBA and recovery with WT or mutant will also suffice if this is technically challenging. Again I understand LRBA is a very large protein, so if this is difficult perhaps a fragment of the binding region is enough to localise to endosomes. Finally, an alternative experiment would be to mutate the Arf's in the recovery of the arf KO and show that mutating that interface prevents recruitment, this is slightly less elegant as it is less direct. I am sure the authors may be able to think of other experiments, but the key point is that some kind of experimental validation of the binding is necessary, if this is not possible I wonder if it should be included at all.

A very minor point is that *trans* and *cis* should be in italics (when referring to Golgi stacks, for example)

We wish to thank the reviewers for their insightful comments.

Reviewer #1 (Comments to the Authors (Required)):

This is a manuscript that focuses on understanding the function of LRBA in regulating endolysosomal traffic. Mutations in LRBA is known to lead to severe childhood immune dysregulation, yet how LRBA functions remains to be determined. It was proposed that LRBA mutations results in increased degradation of CTLA-4, presumably due to issues in endolysosomal traffic and/or maturation. However, how LRBA regulates CTLA-4 mechanistically is still unclear. Since this manuscript attempts to answer this question, it is potentially very interesting and has high significance. Unfortunately the manuscript comes way short of this goal. While it has some interesting findings, such as involvement of Arf1 and Arf6 in LRBA localization to Rab4 endosomes, the manuscript for most part is quite descriptive and correlative. Additional experiments (some pretty straight forward and easy to do) needs to be included to strengthen the study. For example, (a) mutational analysis of LRBA need to be performed to test in silico findings, (b) is CTLA-4 traffic affected in Arf1 and Arf3 KO cells?, (c) does LRBA affect Arf1 and Arf6 function? These are just few questions that I have after reading this manuscript. I obviously do not expect authors to address all of them, but some mechanistic insights would dramatically strength the manuscript. Additionally, it is still not clear to me how LRBA localization on Rab4-endosomes affect lysosomal trafficking. Authors looked at several aspects of endolysosomal function and none of them seem to be affected (except enlargement). That rises the questions about how LRBA really function. Does it affect specifically endolysosomal traffic of CTLA-4? Many of these questions need to answered before manuscript is ready for publication. Some more specific comments are listed below.

We thank reviewer 1 for the comments. There seems to be a misunderstanding about the focus of our study. The main goal was not to understand how LRBA regulate CTLA-4 trafficking but rather to learn about LRBA deficiency globally as LRBA expression is not limited to the immune system. CTLA-4 is not expressed in the patients' fibroblasts or in HeLa cells. Therefore, CTLA-4 was not the focus of our research. To make this clearer, we determined the secretome and the surface expressed proteins in fibroblasts from patients versus healthy donors. The surface proteome revealed the downregulation of proteins such as transferrin receptor, Apolipoprotein E and EGFR in patients' fibroblasts, indicating that LRBA is also involved in the transport of a subset of proteins beside CTLA-4 to the plasma membrane. These data were confirmed in the secretome data. In addition, we observed a strong increase of secreted proteins from patients' fibroblasts. This fraction was enriched in proteins involved in lysosomal function and positioning, possibly reflecting secretion of lysosomes and/or exosomes. These data are now included into the manuscript.

Point a) we aimed to confirm the binding site of Arf1 in LRBA. LRBA is a huge protein with 2863 amino acids and an apparent molecular weight of 319 kDa. So far, we have been unable to introduce full-length LRBA into cells. The HeLa cell line that expresses 3xFlagEGFP-LRBA was generated by the lab of Anne Simonsen using Flp-INTM recombinase mediated integration system. Given this difficulty, we could not perform the mutational analysis. We obtained LRBA fragments published in Lo et al Science 2015. Some of them do not express well and some of them do not seem to fold correctly. Therefore, we were unable to test the predicted binding site in LRBA and do the mutational analysis. In contrast, Arf1 expresses well and can be mutated. We mutated the residues in Arf1 (Ile46Ser, Ile49Ser) that would mediate the interaction with LRBA and showed that expression of this construct in Arf1/Arf3 KO

cells did not support recruitment of LRBA to endosomes. These data are now included into the manuscript (see also comment #3).

b) As already pointed out above, CTLA-4 is not expressed in HeLa cells. It has been previously reported that the localization of exogenously expressed CTLA-4 was only marginally affected by LRBA knockdown in HeLa cells (Janman et al., Immunology, 2021). Nevertheless, we tested the localization of CTLA-4-GFP (Lo et al., Science, 2015) in ARF1+3 dKO cells. Already in the parental cell line, very little, if any, of CTLA-4-GFP was observed at the plasma membrane, and most of it accumulated in large intracellular structures. Over all there was very little difference between the parental cell line and the ARF1+3 dKO. If there was any, we would not be able to distinguish between a defect in recycling or exit from the TGN. However, as it stands, we assume that the GFP-tagged CTLA-4 is not fully functional or not trafficked in a similar way than in immune cells. Having said this, even in Janman et al, CTLA-4 is not localized at the plasma membrane in their immunostainings in HeLa cells. Given the complication with the interpretation of the data and that Janman et al. already showed that loss of LRBA does only mildly affect CTLA-4 localization in HeLa cells, we decided not to include the data into the manuscript but rather adding pictures below for your information.

c) While we cannot exclude a feedback loop in which LRBA would influence Arf1 and/or Arf3 (we assume this is what the reviewer meant, since we don't have any data on Arf6), we consider it unlikely at this point. We rather think of LRBA as an effector of Arf1 on endosomes. Nevertheless, we tested the reviewer's hypothesis and detected Arf1 and Rab4 in cells lacking LRBA. We found no changes in the level of co-localization. These data are included in the manuscript.

1) Figure 6. Colocalization (or lack of) between LRBA and various endocytic markers (shown in mages) need to be quantified.

We provided the quantification.

2) Figure 7B. Colocalization of LRBA and Rab4 in wt and KO cells need to shown and quantified.

We tested two different Rab4 antibodies. Neither of them worked for immunofluorescence. Therefore, we could not perform the experiment. The co-localization between LRBA and mCherry-Rab4 is shown in Figure 6B and quantified in panel C.

3) Figure 8. The sites predicted to mediate Arf1 and Arf3 binding in silico, need to be tested in cells (IPs and localization).

As pointed out above, we have technical difficulties to express LRBA or LRBA mutants in tissue culture cells and are therefore unable to perform the suggested experiments. We did however introduce mutation in Arf1 in the predicted binding site. This construct did abolish LRBA recruitment to endosomes (Figure 8 I and J) and failed to co-immunoprecipitate with Arf1 (Figure 9 A and B). Since we cannot experimentally verify the Arf1 binding site in LRBA, we make it very clear that this is only a prediction.

Reviewer #2 (Comments to the Authors (Required)):

In this manuscript, Szentgyorgyi et al. address the cellular function of the Lipopolysaccharide Responsive Beige-like Anchor protein (LRBA). Patients have been identified with mutations at amino acids 816 and 2288 in LRBA, both of which lead to severe childhood immune dysfunction, which has been linked to impaired trafficking of the immune checkpoint receptor, CTLA-4, via Rab11. The authors do a series of experiments on LRBA, which may provide some new insight on its function. They maintain (in the abstract) that they do not find colocalization with Rab11, but instead observe colocalization with Rab4-containing endosomes. They provide data that suggests that patient cells with mutant LRBA have essentially no LRBA expression, and this in turn leads to morphological changes at the TGN. However, the altered morphology has little or no effect on Golgi function, including M6PR trafficking and biosynthetic transport (although secretion was not tested). At the same time, the authors observed that endolysosomes are enlarged and clustered in the perinuclear region of the cell. However, these enlarged endolysosomes are also functional, and capable of EGFR degradation. Moreover, the degranulation of lytic granules (which are lysosome-related organelles), is similarly unaffected in patients, which apparently differentiates between LRBA and LYST, a large BEACH domain protein that causes Chediak-Higashi disease. Additional studies suggest that Arf1 and Arf3 together recruit LRBA to Rab4-containing endosomes (but are not essential for Golgi recruitment, where the majority of LRBA is localized). Finally, the authors show AlphaFold models and propose an interaction between LRBA and Arf1/3-but this is not tested experimentally.

This is a study that uses fibroblasts derived from patients with mutations in LRBA to address the biological function of this protein. While there is some limited new insight into the complex function of this very large protein, and the study primarily illustrates various morphological phenomena, and it provides little new mechanistic information. A few of the key weaknesses of this study include the limited "positive data" for the function of LRBA, the failure to mechanistically connect between Arf1/3 and LRBA, limited insight as to how mutant/non-expressed LRBA leads to enlarged endolysosome, and how these enlarged structures impact the immune defects in patients.

We thank the reviewer for his/her assessment.

Specific comments:

1) The authors show loss of mRNA and protein expression for the two patient cells. Loss of protein expression might be explained by degradation of a misfolded mutant protein. Is this indeed the case? If so, what is the explanation for decreased mRNA levels? It seems that these are important questions that are not addressed. On the other hand, there is a lot of data showing morphological change that has little effect on function.

We agree with the reviewer that it is interesting that in the patients' cells also the LRBA mRNA levels are down. We assume that this is due to nonsense-mediated mRNA decay. We decided not to follow up the mRNA decay, because the main problem of the cells is the lack LRBA protein, whether this is regulated on the mRNA or on the protein level or both is an interesting question for the future. We are currently more interested in the cellular consequences. We respectfully disagree that we do not show effects related to function. We show that the Arf1/3-dependent recruitment of LRBA onto Rab4 endosomes is required to maintain endolysosome/lysosome homeostasis. Moreover, we provide strong evidence that the localization of the bulk of protein may not always reflect the place of its main function. We bolstered up our findings during the revisions and performed surface biotinylation followed by LC/MS and secretome analyses, which allowed us to further establish that the main defect in LRBA patients' fibroblasts comes from the endolysosomal system and not from the Golgi apparatus.

2) Golgi morphology is often affected by a variety of conditions and genetic manipulations. Given that there is no detected change in Golgi function, these data are not highly consequential, and should be included as supplemental rather than primary data.

We would like to keep the Golgi morphology data as a main figure because the bulk of LRBA is present on the Golgi. We also showed in the last version of the manuscript that LAMP1 is underglycosylated. In addition, we determined the surface proteome and the secretome (Figure 3 D-G, Figure S2 A and B). See also point 3.

3) Golgi secretion was not tested.

To test secretion and recycling at the same time, we determined the surface proteome and the secretome in cells from patients and healthy donors (Figure 3 D-G, Figure S2 A and B). These data show that less extracellular matrix proteins are secreted and some plasma membrane proteins such as EGFR, transferrin receptor and Apolipoprotein E are less abundant at the plasma membrane. Moreover, there are only few factors that are secreted more efficiently from cells of healthy donors compared to patients' cells. These data are now included into the manuscript. In addition, LAMP1 is also underglycosylated. These data are consistent with our finding that Golgi function is probably only slightly impaired and that traffic from the Golgi to the plasma membrane is not drastically altered.

4) The authors contend that Rab4 colocalizes with LRBA, whereas Rab11 does not. This reviewer does not agree with this interpretation. First, there is no quantification of the data shown in Fig. 6. Second, from the single set of images displayed for Rab4 and Rab11, there is little discernible difference in the overlap with LRBA. It is entirely unclear on what basis this interpretation is made. Evidence for select recruitment to Rab4 endosomes is not compelling, and this is one of the main conclusions drawn from the study.

We apologize for the omission. We now include the quantification of the localization data in Figure 6. The Pearson's coefficient is higher for Rab4 than for Rab11 which is in agreement with another report (Wilson et al. 2023, JCS) showing that LRBA is strongly enriched on Rab4a compartments as compared to either Rab11a or Rab25 determined by proximity labeling and proteomic analysis. We agree with the reviewer that LRBA co-localizes more with Rab11 endosomes than with Rab5 or Rab7 structures. We amended the text to better reflect the LRBA co-localization pattern with endosomal markers.

5) Fig. 6 entirely lacks quantification, but also uses over-expression of the endosomal markers, which can lead to misinterpretations due to high levels of expression. Why are endogenous endosomal markers not used? There are excellent antibodies available as markers of endosomes.

As pointed above, we included the quantification into Fig. 6. As for the use of endogenous endosomal markers. While antibodies against Rab proteins work well in Western Blot, they are not very specific in IF experiments, and hence people use fluorescently tagged proteins for intracellular localization experiments. We also used LAMP1 as a marker, as is commonly used in the field for endolysosomes and lysosomes. We could have used EEA1 for early endosomes, but then EEA1 does not label all early endosomes. Finally, to our knowledge there are no alternative generally accepted markers for Rab11 or Rab4 compartments. So, in a way, we are stuck with the Rabs, and since there are no functional antibodies, we use the tagged versions.

6) There is no proposed rationale for why, when the vast majority of LRBA is found at the Golgi, that there are such significant effects at endosomes.

We spent quite some efforts to elucidate the function of LRBA on the Golgi. We report that there are mild phenotypes but nothing detrimental in the LRBA-deficient cells. One possibility is that there are other BEACH domain containing proteins that can in part compensate for loss of LRBA at the Golgi but not in endosomal transport. For example, neurobeachin, which is probably the closest homolog, is expressed in skin fibroblasts and may partially compensate. Another possibility is that LRBA is essential at the Golgi, but not under the growth conditions we tested, but perhaps after stimulation of some sort or under stress. Finally, it is also possible that LRBA in the Golgi represents a reservoir for the endosomal pathway and the distribution between endosomal and Golgi fraction might be dependent on the environment and growth conditions. None of the possibilities are mutually exclusive, and one can surely think of other possibilities. At this point, this is pure speculation and hence we refrained from discussing this point in detail.

7) There are no essential controls demonstrating the knockout of Arf1, Arf3 and the double knockout. There is a reference to a previous paper, but cells can evolve and one needs to see the knockout in this manuscript (Fig. 7).

As per request, we reconfirmed the knockout and these data are included in the manuscript (Fig. 7 C).

8) There are no "rescue" experiments to demonstrate that the effects visualized are not due to off-target effects or clonal selection of cell lines.

We include rescue experiments for the Arf1+3 dKO cell lines. Unfortunately, we were unable to re-express LRBA in the patients' fibroblasts even using transposon-mediated insertion. The Arf1+3 dKO rescue experiments are now included in the manuscript (Fig. 7 F and G).

9) If Arf1/3 KO is insufficient to impair LRBA localization to the Golgi and the authors propose that other Arfs may be responsible/compensating, then this should be tested.

The binding site of LRBA in Arf1 and 3 and the surrounding region are absolutely conserved between Arf1, 3, 4 and 5, but different in Arf6 (Fig. 8 E). Thus, there is no reason to believe that Arf4 and Arf5 would not bind to LRBA. The knockout of either Arf1+Arf4 or Arf4+Arf5 is lethal (Pennauer et al., JCB 2022) and thus knockout of Arf1,3,4,5 cannot be established. Therefore, we cannot easily test the compensation. It is, however, worthwhile to mention that Arf4 alone can compensate for the loss of Arf1, 3 and 5.

10) The authors show AlphaFold models suggesting that Arfs might interact with LRBA. This is entirely speculative, and yet determining if there is binding, how, and whether the binding is required is one of the mechanistic connections that would contribute to current understand of LRBA function. Why do the authors not test binding?

We agree with the reviewer that the predictions from the LRBA-Arf1 complex model need to be tested. We introduced mutants into Arf1 which according to the model should disrupt the Arf1-LRBA interaction. This appears indeed to be the case. When we transfect the mutated Arf1(I46S,I49S) into Arf1+3KO cells, LRBA is not found on endosomes, while expression of WT Arf1 rescues LRBA binding to endosomes. These data are now included into the manuscript (Figure 8 I and J).

We were also successful to pulldown endogenous LRBA with Arf1 (or the constitutively active form) and this interaction was reduced when Arf1(I46S,I49S) was used for the pulldown (Figure 9 A and B).

We also obtained LRBA fragments generated by Lo et al., (Science 2015) that cover the entire LRBA protein. The expression of some of the fragments is very poor, including fragment #2, which should contain the putative Arf1 binding site. The poor expression level questions whether this construct can be properly folded. In addition, at least the C-terminal construct is not properly folded as it runs as a high molecular weight smear. This C-terminal construct contains only 6 blades of a 7-bladed WD40 repeat, providing a likely explanation for the strange behavior. Nevertheless, we used the constructs for IPs, which were unfortunately inconclusive. Given the different expression levels, we adjusted the LRBA fragment concentration in each pull-down. This analysis revealed that binding to construct 6 was due to the highest expression level. At similar levels, it actually bound to Arf1 at background levels. Other groups did similar pull-downs without adjusting the protein concentration, which might lead to misleading results. Therefore, we do not trust these constructs and decided not to use them for our analyses.

Given the large size of LRBA and that it cannot be easily transfected into cells, we were unable test mutants in the potential Arf1 binding site and determine their effect.

While we have supporting data for the binding site of LRBA in Arf1, we could not test the Arf1 binding site in LRBA. We make this also clear in the manuscript.

11) The authors note that "MAB21L2 transcripts are still present in both patients, although with somewhat reduced levels as compared to controls". However, this does not appear to be an accurate depiction: Sup. Fig. 1 shows one patient with much lower levels than control, while the other is essentially the same, but with a huge standard deviation. These data likely need additional repeats to verify if indeed there are statistical differences before making conclusions.

We change the wording about the MAB21L2 transcript levels in patient cells. The only motivation to do provide these data was to show that the observed phenotype was not related to MAB21L2. MAB21L2 expression levels in the fibroblasts are close to the detection limit, and hence error bars are high. Since MAB21L2 transcript levels are detected at least in patient #2 to a similar level than in healthy donor cells and since we observe the enlarged endolysosome phenotype in both patients' cell lines, there is no correlation between MAB21L2 expression levels and the enlarged endolysosome phenotype. According to the human protein atlas, MAB21L2 is not detected in skin fibroblasts, and we detect small amounts, which goes along the same lines.

12) Data from Fig. S2 need to be quantified-including the immunoblots and fluorescence images.

We provide the quantification of the western blots. As for the fluorescent images in Fig. S2, we use three additional lysosomal markers (V0a3, LysoTracker, Cathepsin D), in addition to the three markers from main Fig. 2 (LAMP1, Magic Red and Rab7) to show spread of endolysosomes and lysosomes in patient cells compared to healthy donors. All six markers look similar and support this conclusion. We did quantify the Magic Red signal in the main Fig. 2 and additionally the LysoTracker signal in Fig. S2. We do not understand what difference the reviewer would expect, given this obvious phenotype being very similar across six markers. Together with other data presented in the manuscript, these data also allow us to conclude that these endolysosomes are active.

13) There are no statistical tests applied in Fig. S3 to address whether the differences observed are significant (B and C).

We apologize for this omission. The statistical test is now included.

14) In Fig. S5, the authors claim that in HeLa cells, LRBA juxtaposes with the Golgi but does not colocalize with M6PR. There is no quantification. This reviewer can identify, even in the area chosen by the authors for an insert, several instances of what appears to be colocalization between M6PR and LRBA. Conclusions must be based on quantified data.

We included the quantification. The Pearson's coefficient is much higher for the co-localization with giantin than with M6PR.

15) In Fig. S5F-the authors show homologous residues across species in Arf1/3 predicted to interact with LRBA, but do not show the interaction.

We mutated the relevant residues in Arf1 and showed that expression of the mutant Arf1(I46S,I49S) does not restore LRBA endosomal localization in Arf1+3 dKO cells, while wild-type Arf1 does under same conditions (Figure 8 I and J). The mutated Arf1(I46S,I49S) is still recruited to Rab4 endosomes (Figure 8 F,G and H). Thus, our data are consistent with the notion that LRBA is binding to Arf1 at the predicted site. Since this region is identical between Arf1-5, we assume that all four Arfs will bind LRBA.

Reviewer #3 (Comments to the Authors (Required)):

Szentgyörgyi et al. have used a range of cell biology techniques to characterise patient mutations in LRBA, a Golgi

and endosome localised protein. They carefully characterise the phenotype of the patient cells and the effects of LOF LRBA mutants on the sub-cellular organisation. They see clear and striking phenotypes of enlarged endosomes, and they usefully and sensibly include negative phenotypes such as on EGF uptake. The endosomal phenotype is curious as LRBA appears mostly localised to the Golgi; however, the authors identify a population at the endosome. The endosomal population is demonstrated to be recruited by Arf1 and 3, and an alphafold model of the binding interface is presented. Overall, the manuscript presents a set of very well-performed experiments, appropriately interpreted with proper controls in most cases. The data is important, and understanding the molecular cell biology of diseases such as LRBA mutations is valuable and allows for a better understanding of disease coupled with the ability to make novel insights into fundamental biological mechanisms. There are a number of controls and additional experiments I detail below that I feel will strengthen the manuscript and some of its claims.

We thank the reviewer for the positive assessment of our data!

In figures 1 and 2 the authors use TGN46 to measure the TGN shape and volume. They refer to this as the TGN, however, this can be high risk in situations where membrane trafficking is defective as the trafficking of TGN46 could result in a different localisation. In Figure S1B, the authors do address this but performing a GCA treatment and subsequent washout, however I have several issues with this experiment, considering the strength of evidence needed. Firstly, the time course of 4 hours is enough to return to steady state- as the authors show already. Secondly, I see a difference at the 40-minute time point. Thirdly, none of this data is quantified. I also note that the LAMP1 glycosylation was affected, and as the authors point out in the discussion, this could be interpreted as a defect in Golgi function. I finally note that the TGN looked unaffected in EM, although it is hard to tell from the presented data. Ultimately to prove Golgi function is unaffected a lot more evidence is required.

We agree with the reviewer that we cannot exclude that there are minor defects in Golgi function. We actually never tried to convey this message. In fact, we did mention that LAMP1 glycosylation is affected and that there was less EGFR at the plasma membrane. On the other hand, we did not observe any gross defects in M6PR and structural changes on the ultrastructural level. To address Golgi function better, we determined the secretome and the surface proteome from healthy donor and patients' fibroblasts (Figure 3 D-G, Figure S2 A and B). Gratifyingly, we detected less EGFR and TfR at the plasma membrane as well as a number of other proteins in the samples from patients' fibroblasts. Yet again, we did not observe drastic remodeling of the surface proteome. The strongest reduction in the secretome was from proteins related to the extracellular matrix, which might also be an indication that glycosylation in the Golgi is not fully operational.

Thus, we maintain that Golgi function is only mildly affected.

Surprisingly, patients' fibroblasts secrete more lysosome and endosome related proteins, indicating an increase of lysosome fusion with the plasma membrane and potentially exosome release from these cells.

Although there are many advantages to working with patient cells, one of the problems is that they are not isogenic with the controls. Accordingly, one additional control I would expect is to have overexpression of the WT protein cDNA to demonstrate that the phenotypes recover. It may be a bit much to ask for the authors to repeat every experiment with an additional control, however I would expect this for at least a few key experiments and observations.

We agree with the reviewer. We aimed to rescue the patients' fibroblasts by expressing WT LRBA. The problem is that LRBA is a huge protein with over 3000 amino acids and is therefore hard to transfect into cells. We used a

sleeping beauty transposon construct (15 kb), which was kindly provided to us by Anne Simonsen. We tried very hard to get the rescue plasmid into the fibroblasts, but were unsuccessful at this point. Therefore, we cannot provide the data for the rescue of the patients' fibroblasts. We did not attempt to use lentiviral transfection because the size the LRBA construct is much bigger than what can be packaged into lentiviruses (8-10 kb).

The data that Arf1 and LRBA colocalise at the endosome is convincing (although not quantified); I do not find the localisation at the Golgi convincing. The authors claim a colocalisation there, but I see two different localisations.

We quantified the LRBA-Arf1 localization to endosomes and Golgi. These data are now included into the manuscript. The Pearson's coefficient is very high, also in the Golgi region. We agree that the merge of the LRBA and Arf1 localization at the Golgi may not have been optimally adjusted and thus could give the impression that the co-localization is not very high. From gray scale images, however, the high degree of co-localization is apparent.

LRBA has a clear endosomal population in HeLa cells; as I said, at least a pool of this appears to coincide with Arf1. I note that 1) arf3 is not assessed. More importantly, 2) no endosomal population was visible in the healthy donor cells in 1D. The authors do discuss this in the discussion, suggesting that the endosomes are more clustered in the healthy donor cells, however this is an important issue as it really points to the key phenotypes in the cells. I think localising other arf1/3 endosomal markers and showing them in endosomes that are proximal to the Golgi would be an important way to conclude this point.

We apologize for the poorly adjusted image. Endosomal LRBA is also present on healthy donor cells and these data are now included (Figure 1 D). We agree that we did not assess Arf3. We need to delete both Arf1 and Arf3 to eliminate endosomal LRBA, indicating that Arf3 can at least substitute for Arf1 in the recruitment of LRBA to endosomes. Moreover, the binding sites for LRBA in Arf1 and Arf3 are highly conserved.

We also aimed to bolster up our findings with Arf3-mCherry, which is available on Addgene (#79420). However, Arf3-mCherry is largely cytoplasmic with a faint staining at the Golgi. This staining is consistent with previously published data (Chun et al., MBoC 2008, Fig. 2, Makyio et al., EMBO 2012, Fig. 1, Manolea et al, MBoC 2010, Suppl. Fig. 2). Therefore, we think that Arf3-mCherry cannot be recruited to the membrane efficiently and may not be fully functional. Hence, we decide against using this construct for analyses.

Generally, the quantification and biological repeats are of the highest quality, unusually so, and I commend the authors for their rigour. Saying that there are a number of important quantifications missing. Co-localisations in figure 6 and 7. If the authors find colocalisation quantification a problem, at least the number of experimental repeats should be presented in the legend so that an experimental mistake for a key observation can be ruled out.

Thank you very much for the praise! Indeed, we always strive to provide the highest quality of data and be rigorous in our science. We provide now also the quantifications for the data shown in Fig. 6 and 7. We avoid publishing experiments that were not performed at least three times independently. The number of cells analyzed will also be indicated in the legends of Figure 6 and Figure 7B.

The arf1/3 KO is really a key result in this paper. However, no evidence of a KO at the RNA or protein level is presented. In addition, to rule out off-target effects of the RNA, I would expect a recovery by over-expression of the arf1 and independently arf3. If the author's hypothesis is correct, either should recover.

We reconfirmed the ARF1+3 dKO again (Figure 7 C). We initially obtained the cell line from the lab of Martin Spiess (Pennauer et al., JCB 2022). Moreover, re-expression of Arf1-GFP rescued the localization of LRBA to endosomes in ARF1+3 dKO cells (Figure 7 F and G). These data are now included into the manuscript.

I find the evidence presented in Figure 8 compelling but weak. A PAE plot is absolutely essential when presenting AF data, without that the confidence in the binding model is not assessable. I think that in an ideal world, binding data by pull-down or ITC would be a large improvement (with, of course, the mutant as well). The authors could use the alpha fold to design a fragment of LRBA if it is hard to purify. A KO of LRBA and recovery with WT or mutant will also suffice if this is technically challenging. Again I understand LRBA is a very large protein, so if this is difficult perhaps a fragment of the binding region is enough to localise to endosomes. Finally, an alternative experiment would be to mutate the Arf's in the recovery of the arf KO and show that mutating that interface prevents recruitment, this is slightly less elegant as it is less direct. I am sure the authors may be able to think of other experiments, but the key point is that some kind of experimental validation of the binding is necessary, if this is not possible I wonder if it should be included at all.

We included the PAE plot and the iPTM score. Even though we aimed to confirm the Arf1 binding site in LRBA, we were already unable to express WT LRBA in cells, and hence could not test the mutant proteins. The reason for this problem is the large size of LRBA. However, we could test the binding site of LRBA in Arf1. When we mutated the predicted residues in Arf1 (I46S, I49S), we failed to recruit LRBA to endosomes, and pull-down of endogenous LRBA was decreased to a similar level that when the dominant inactive version was used, while the constitutively active mutant co-immunoprecipitated with LRBA. These data indicate that at least this part of the model might be correct. We make it clear that the predicted binding site of Arf1 in LRBA is only a model and has not been experimentally verified.

A very minor point is that *trans* and *cis* should be in italics (when referring to Golgi stacks, for example)

Done.

July 25, 2024

RE: JCB Manuscript #202401167R

Dr. Shahrzad Bakhtiar
Goethe University Frankfurt
Theodor-Stern-Kai 7
Frankfurt 60529
Germany

Dear Dr. Bakhtiar:

Thank you for submitting your revised manuscript entitled "Arf1-dependent LRBA recruitment to Rab4 endosomes is required for endolysosome homeostasis". We would be happy to publish your paper in JCB pending final revisions necessary to meet our formatting guidelines (see details below).

A. MANUSCRIPT ORGANIZATION AND FORMATTING:

- 1) Text limits: Character count for Articles is < 40,000, not including spaces. Count includes abstract, introduction, results, discussion, and acknowledgments. Count does not include title page, figure legends, materials and methods, references, tables, or supplemental legends.
- 2) Figures limits: Articles may have up to 10 main text figures.
- 3) Figure formatting: Scale bars must be present on all microscopy images, * including all inset magnifications (you may alternatively indicate the diameter of the inset). Molecular weight or nucleic acid size markers must be included on all gel electrophoresis.
- 4) Statistical analysis: Error bars on graphic representations of numerical data must be clearly described in the figure legend. The number of independent data points (n) represented in a graph must be indicated in the legend. Statistical methods should be explained in full in the materials and methods. For figures presenting pooled data the statistical measure should be defined in the figure legends. Please also be sure to indicate the statistical tests used in each of your experiments (either in the figure legend itself or in a separate methods section) as well as the parameters of the test (for example, if you ran a t-test, please indicate if it was one- or two-sided, etc.). Also, if you used parametric tests, please indicate if the data distribution was tested for normality (and if so, how). If not, you must state something to the effect that "Data distribution was assumed to be normal but this was not formally tested."
- 5) Abstract and title: The abstract should be no longer than 160 words and should communicate the significance of the paper for a general audience. The title should be less than 100 characters including spaces. Make the title concise but accessible to a general readership.
- 6) Materials and methods: Should be comprehensive and not simply reference a previous publication for details on how an experiment was performed. Please provide full descriptions in the text for readers who may not have access to referenced manuscripts.
- 7) All antibodies, cell lines, animals, and tools used in the manuscript should be described in full, including accession numbers for materials available in a public repository such as the Resource Identification Portal. Please be sure to provide the sequences for all of your primers/oligos and RNAi constructs in the materials and methods. You must also indicate in the methods the source, species, and catalog numbers (where appropriate) for all of your antibodies. Please also indicate the acquisition and quantification methods for immunoblotting/western blots.
- 8) Microscope image acquisition: The following information must be provided about the acquisition and processing of images:
 - a. Make and model of microscope
 - b. Type, magnification, and numerical aperture of the objective lenses
 - c. Temperature
 - d. Imaging medium

- e. Fluorochromes
- f. Camera make and model
- g. Acquisition software
- h. Any software used for image processing subsequent to data acquisition. Please include details and types of operations involved (e.g., type of deconvolution, 3D reconstitutions, surface or volume rendering, gamma adjustments, etc.).

10) Supplemental materials: There are strict limits on the allowable amount of supplemental data. Articles may have up to 5 supplemental figures.. Please also note that tables, like figures, should be provided as individual, editable files. A summary of all supplemental material should appear at the end of the Materials and methods section.

13) ORCID IDs: ORCID IDs are unique identifiers allowing researchers to create a record of their various scholarly contributions in a single place. Please note that ORCID IDs are now *required* for all authors. At resubmission of your final files, please be sure to provide your ORCID ID and those of all co-authors.

Please note that JCB now requires authors to submit Source Data used to generate figures containing gels and Western blots with all revised manuscripts. This Source Data consists of fully uncropped and unprocessed images for each gel/blot displayed in the main and supplemental figures. Since your paper includes cropped gel and/or blot images, please be sure to provide one Source Data file for each figure that contains gels and/or blots along with your revised manuscript files. File names for Source Data figures should be alphanumeric without any spaces or special characters (i.e., SourceDataF#, where F# refers to the associated main figure number or SourceDataFS# for those associated with Supplementary figures). The lanes of the gels/blots should be labeled as they are in the associated figure, the place where cropping was applied should be marked (with a box), and molecular weight/size standards should be labeled wherever possible.

Journal of Cell Biology now requires a data availability statement for all research article submissions. These statements will be published in the article directly above the Acknowledgments. The statement should address all data underlying the research presented in the manuscript. Please visit the JCB instructions for authors for guidelines and examples of statements at (<https://rupress.org/jcb/pages/editorial-policies#data-availability-statement>).

B. FINAL FILES:

****It is JCB policy that if requested, original data images must be made available to the editors. Failure to provide original images upon request will result in unavoidable delays in publication. Please ensure that you have access to all original data images prior to final submission.****

****The license to publish form must be signed before your manuscript can be sent to production. A link to the electronic license to publish form will be sent to the corresponding author only. Please take a moment to check your funder requirements before choosing the appropriate license.****

Thank you for your attention to these final processing requirements. Please revise and format the manuscript and upload materials within 7 days. If you need an extension for whatever reason, please let us know and we can work with you to determine a suitable revision period.

Thank you for this interesting contribution, we look forward to publishing your paper in Journal of Cell Biology.

Sincerely,

Jennifer Stow, PhD
Monitoring Editor

Andrea L. Marat, PhD
Deputy Editor

Journal of Cell Biology

Reviewer #3 (Comments to the Authors (Required)):

The authors have addressed all of the comments in a satisfactory manner. There are a set of points that they could not address due to technical reasons (fibroblast rescue, LRBA overexpression); however, in all cases, the reasons are legitimate, and the resulting limitations are made clear in the manuscript. Accordingly, I commend the authors for their work and recommend the manuscript be accepted.